# A combinatorial synthetic strategy for developing genome-editing protein-delivery agents targeting mouse retina

Jianye Zhang[1,14], Rafał Hołubowicz[1,2,14], Roman Smidak[1,14], Yulun Hu[3], Samuel W. Du[1,4], Jiin H. Felgner[5], Grazyna Palczewska[1,6], Carolline Rodrigues Menezes[1,4], Eleonora Risaliti[1,4], Zhiqian Dong[1], Xiuli Ma[1], Mojtaba H. Shayegan[3], Paul Z. Chen[7,8,9,10], Li Xing[11], Maria Hołubowicz[1], Bowen Li[3], David R. Liu[7,8,9], Philip L. Felgner[5], Gregory P. Tochtrop[3] ✉ & Krzysztof Palczewski[1,4,12,13] ✉

CRISPR/Cas9-based gene-editing technologies offer promise for treating inherited retinal diseases (IRDs), however safe and efficient ocular delivery of precision editors remains challenging. To address this challenge, we report a class of Coomassie brilliant blue (CBB)-derived lipidoids that bind and deliver proteins. Subretinal injection of Cre complexed with these lipidoids into mT/mG mice leads to robust recombination in the retinal pigment epithelium and photoreceptors. We employ the CBB-lipidoid platform to deliver adenine base editor (ABE) ribonucleoproteins (RNP). Incorporating CBB lipidoids into liposomes improves delivery efficiency. CBB11 stands out for facilitating precise in vivo ABE-mediated gene editing. Delivery of liposome-CBB11-RNP complexes results in a 120-fold increase in base editing compared to RNP alone and restores the scotopic ERG b-wave response in the rd12 mouse model. These results demonstrate the potential of CBB-augmented, liposome-RNP systems for therapeutic gene editing in the eye, paving the way for single-dose precision medicines to treat IRDs.

The future of medicine is increasingly being shaped by advances in genome editing, which promises to revolutionize the treatment of genetic diseases, cancers, and various other conditions. Building on the initial discoveries of the CRISPR-Cas9 system,[1] the transformative technologies of base editing[2,3] and prime editing[4] offer solutions for nearly all disease-causing mutations, in addition to opening up new avenues for enhancing genetic traits and studying complex biological systems. However, realizing the therapeutic potential of genome editing requires safe, efficient delivery of these gene editors to specific cells and tissues.

[1]Gavin Herbert Eye Institute – Brunson Center for Translational Vision Research, Department of Ophthalmology, University of California, Irvine, Irvine, CA, USA. [2]Department of Biochemistry, Molecular Biology and Biotechnology, Faculty of Chemistry, Wroclaw University of Science and Technology, Wroclaw, Poland. [3]Department of Chemistry, College of Arts and Sciences, Case Western Reserve University, Cleveland, OH, USA. [4]Department of Physiology and Biophysics, University of California, Irvine, Irvine, CA, USA. [5]Adeline Yen Mah Vaccine Center, Department of Physiology and Biophysics, University of California, Irvine, California, USA. [6]Polgenix Inc., Department of Medical Devices, Cleveland, OH, USA. [7]Merkin Institute for Transformative Technologies in Healthcare, Broad Institute of MIT and Harvard, Cambridge, MA, USA. [8]Department of Chemistry and Chemical Biology, Harvard University, Cambridge, MA, USA. [9]Howard Hughes Medical Institute, Harvard University, Cambridge, MA, USA. [10]David H. Koch Institute for Integrative Cancer Research, Massachusetts Institute of Technology, Cambridge, MA, USA. [11]UC Irvine Materials Research Institute, University of California, Irvine, Irvine, CA, USA. [12]Department of Chemistry, University of California, Irvine, Irvine, CA, USA. [13]Department of Molecular Biology and Biochemistry, University of California, Irvine, Irvine, CA, USA. [14]These authors contributed equally: Jianye Zhang, Rafał Hołubowicz, Roman Smidak. ✉e-mail: gpt6@case.edu; kpalczew@uci.edu

Currently, three primary modalities are utilized for delivering editors to target cells in vivo: viral vectors, non-viral mRNA formulations, and direct protein-delivery systems. Viral vectors, such as adeno-associated viruses (AAVs), are widely used due to their versatility and high efficiency in delivering genetic material into cells; however, they carry risks of immunogenicity and potential genomic integration[5]. Furthermore, these vectors can only package approximately 5 kb of genetic material, which is insufficient to accommodate large transgenes within a single AAV genome, but splitting the cargo into multiple AAVs limits the transduction efficiency[6–8]. Direct delivery of mRNA via non-viral vectors, including lipid nanoparticles (LNPs) and polymer-based systems, offers a safer alternative by avoiding many of the limitations of AAVs. There is currently rapid development of mRNA-delivery approaches to enable delivery to a variety of tissues,[9–11] but efficient genome editing in this context has remained challenging in tissues other than liver[12,13]. Direct delivery into cells of pre-assembled editor proteins and guide-RNAs as ribonucleoprotein (RNP) complexes represents an emerging approach that conceptually provides transient and controlled activity, reducing the risk of off-target effects and immunogenicity[14–16]. Furthermore, protein delivery systems can be engineered to enhance specificity toward the tissue of interest, and thus improve the overall precision of genome-editing therapies[17].

Despite the promise of tissue-specific delivery of pre-assembled, therapeutic Cas9 RNPs, including base editors (BEs) or prime editors (PEs), their large size, overall negative charge, and lack of a specific protein-transduction mechanism pose challenges that must be overcome. Various systems have been explored for the delivery of BE- and PE-RNPs, including liposomes,[18] micelles,[19] polymer-based systems,[20] dendrimers,[21] and nanoparticles[22,23]. Lipid-based systems offer notable advantages. The size and composition of lipid-based vehicles can be easily tailored to carry editors and RNA, and these vehicles often rely on electrostatic interactions between positively charged lipids and negatively charged nucleic acids. As there are no naturally occurring positively charged bilayer-forming lipids, researchers have leveraged design principles from liposome research to synthesize cationic-lipid molecules. The first commercial cationic liposome-based transfection product, Lipofectin, was introduced in 1988 to deliver plasmids into cells in vitro[24]. Since then, further development has resulted in numerous ionizable/permanently ionized lipids with multiple functionalities like Lipofectamine and MC3,[25] and delivery of large RNPs has been reported using lecithin-based liposomes[26] and ionizable LNPs[27]. Critically, several LNP formulations have obtained approval from the US Food and Drug Administration (FDA), most notably the branched lipid compositions (including SM102 and ALC-0315), which were approved for delivery of the mRNA COVID-19 vaccines[28]. Both in the case of mRNA and RNPs, negative charges on the RNA nucleotides provide a driving force to bind the ionizable lipids that facilitate the encapsulation of the cargo.

We improve the existing genome-editing toolkit by developing a safe and efficient protein delivery system. To achieve this, we design a methodology based on Coomassie Brilliant Blue (CBB) lipidoids, whereby the CBB moiety strongly binds proteins, and the lipid moiety interacts with the cell membrane to enable uptake of the protein cargo by the cell. We rapidly synthesize libraries of CBB derivatives via click chemistry and sulfonamide conjugation. We evaluate the ability of these CBB derivatives to enhance the delivery of genome-editor proteins (Cre) and RNPs (adenine base editors, ABEs) in vitro and in vivo. After initial success, we further refine our approach by incorporating the CBB lipidoid into liposomes to achieve efficient editing in an in vivo inherited retinal disease (IRD) mouse model, *rd12*. Our approach is distinct from LNPs, as we form the CBB liposomes without protein and use them as neutral, aqueous suspension to complex the cargo in neutral pH and without organic solvents. Our CBB lipidoids engage in CBB-protein interactions, and thus can be used for the delivery of all types of proteins, including recombinases (Cre, Bxb1), genome-editing

proteins (zinc finger nucleases, TALENs), and genome-editing RNPs (Cas9, ABE, PE, and others). We maximize the potential of CBB liposomes for RNP delivery by incorporating ionizable lipids to achieve synergistic complexing of the RNPs via CBB-protein and ionizable lipid-guide RNA interactions. Incorporating CBB into liposomes enhances RNP-mediated genome editing, allowing to reach clinically significant recovery of visual function and editing in the genome. This outcome affirms that specialized delivery agents, including our CBB lipidoids, are essential for achieving the RNP-mediated editing efficiency necessary for further clinical translation.

## Results

### Design and synthesis of coomassie-derived lipidoids

Lipidoids have been developed as key components of lipid-based particles to facilitate membrane penetration, cargo release and endosomal escape via receptor mediated endocytosis, interactions with the lipid bilayer, and phase transitions in the acidifying endosome[29]. However, they do not offer specific interactions with proteins. To enable the interactions between the lipids and proteins, we incorporated a protein-binding domain into lipidoids, as shown in Fig. 1. Our first-generation CBB compounds began with commercially available CBB G-250 (CBB22), which was functionalized through the coupling of amides or esters of N-methyl-1,3-diaminopropane (K1) or PEG$_3$ (K3) to CBB, utilizing phosphorus trichloride to form the amino-sulfonamide or sulfonic ester (see Supplementary Fig. 1 and Synthesis of CBB intermediates and lipidoids in Supplementary Information). This intermediate was then coupled to diverse lipid and non-lipid tails of various lengths and degrees of saturation. For example, CBB1 contained a cholesteryl moiety, and CBB3 contained docosahexaenoic acid (DHA). CBB5 and CBB16 were isolated as byproducts during silica gel chromatographic purification of CBB3 and CBB15. CBB1, CBB3, CBB6, and CBB13 were further methylated to produce the lipidoids CBB2, CBB4, CBB7, and CBB14 with the positively charged linker K2. CBB21 was also coupled to the C-terminus of a cell-penetrating peptide (CPP5, sequence KLPVM), to yield CBB18 and CBB19. Upon long-term storage in DMSO, we noted oxidation of CBB11, resulting in CBBox11. Mass spectrometry of separated impurities indicated that CBBox11 comprised two isomers, likely representing hydroxylation events on the polyunsaturated fatty acid or the tertiary amine (Supplementary Fig. 2).

We also developed an alternative pathway to generate protein-binding lipid derivatives of CBB. We reasoned that since our first-generation CBB derivatives eliminated the sulfonic-acid functionality of CBB, retention of this group would enhance protein-CBB interactions, as CBB G-250 binds to proteins through hydrophobic interactions with aromatic residues (Phe, Tyr, Trp) and electrostatic interactions between the anionic sulfonate and basic amino acids (His, Lys, Arg)[30]. We created a CBB derivative, CBBZ0, with a propargyl ether functionality that can be linked by click chemistry to lipid/lipidoid azides. The construction of the click library began with a family of bifunctional linkers, to which lipidoids were joined via Michael addition reactions with acrylates, as illustrated in Supplementary Fig. 3. The alcohol functionality of the bifunctional linker was then converted to an azide via mesylation and displacement with sodium azide. Finally, the CBBZ compounds were assembled via click reactions, creating the CBB-triazole library consisting of compounds CBBZ1-10.

### In vitro delivery of Cre by coomassie-derived lipidoids

First, we used a small genome-editing protein, Cre recombinase, to assess the CBB lipidoid-mediated intracellular protein delivery. We used a HEK293-loxP-GFP-RFP color-switching reporter cell line (CS) that constitutively expresses GFP, while an upstream stop codon prevents expression of RFP. Cre recombinase activity in the nucleus triggers the excision of the floxed GFP-stop sequence, and RFP expression replaces GFP (Fig. 2a)[15]. Nine compounds (CBB2, 7, 11, 12, 13, 14, 18,19,

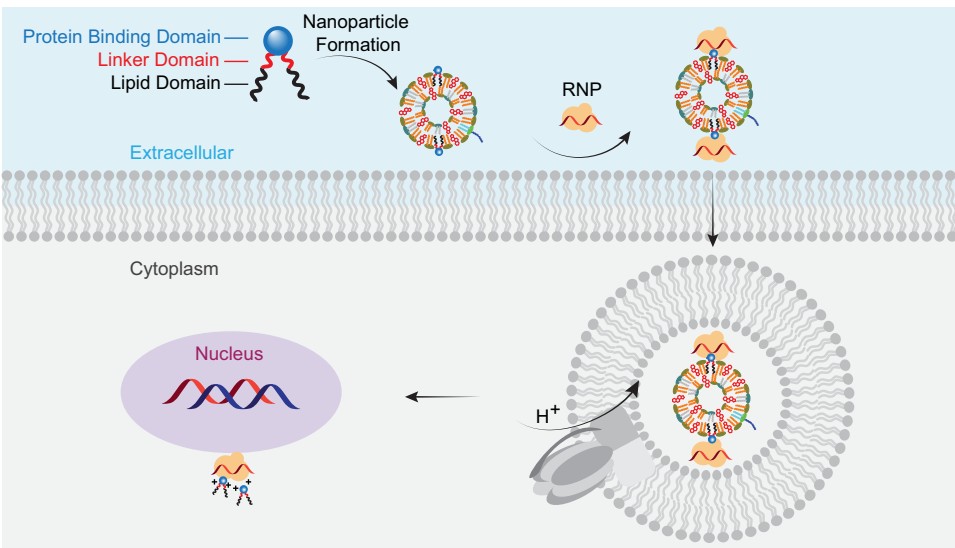

**Fig. 1 | Construction of complexes, and proposed mechanism(s) underlying intracellular delivery of genome-editing ribonucleoproteins (RNPs) mediated by Coomassie Brilliant Blue (CBB) liposomes.** The fundamental design principles for a CBB-containing delivery agent include: (1) the CBB protein-binding domain; (2) an ionizable-linker domain; and (3) a lipid or lipidoid domain. The CBB compounds are integrated into liposomes comprising ionizable lipid (orange), phospholipid (grey), cholesterol (red) and PEG lipid (light blue), and bind RNP on the liposome surface in an aqueous, neutral environment. The RNP-liposome complex is absorbed by the cells via the endocytosis pathway. Subsequent protonation of the ionizable CBB delivery agents and ionizable lipids releases the tagged proteins from the particles, enabling their intracellular transport to the nucleus to edit the targeted DNA sequence.

Z1) were selected for the titration of the [CBB] to [Enzyme] ratio. The CBB lipidoids were dissolved in DMSO (1 volume) and complexed at room temperature with Cre recombinase (19 volumes) at a 2:1, 4:1, 8:1 or 16:1 molar ratio before delivery (Supplementary Fig. 4a). The enzymatic activity of cytosolic Cre was evaluated by fluorescence microscopy according to the extent of GFP to RFP conversion, which indicated that Cre recombinase intracellular delivery was dependent on the CBB-lipidoids concentration, and the dose-response relationship showed that the effect was generally saturated at a ratio 8:1 for most compounds. Accordingly, in the overall screening of the CBB lipidoids, the [CBB]/[Enzyme] ratio was set to 8:1, Lipofectamine 3000 (LF) served as a positive-control vehicle, and free Cre protein was used alongside non-treated cells as a negative control. Compounds were considered positive hits if the ratio of red-to-green (RFP/GFP) quantified by flow cytometry (CBB compounds) or fluorescence microscopy (CBBZ compounds) was two-fold higher than in the cells treated with free Cre (Fig. 2b, panel "C"). Six compounds from the amide coupled library (CBB6, CBB11, CBB14, CBB18, CBB19, and CBB21) demonstrated the ability to enhance delivery of Cre (Fig. 2b, c). The CBBZ approach was tested with C12 and C16 aliphatic chains present in CBB12 and CBB6, respectively, but it did not improve the ability of CBB to deliver Cre compared to amide-linked CBB compounds (Fig. 2d, C12: Z1, Z2, Z6, Z10; C16: Z3, Z7, Z8). Although SM102 enabled efficient delivery of ABE-RNPs in our previous study[15], neither of the CBBZ compounds, in which the branched 9-heptadecyl lipid moiety of SM102 or the entire SM102 moiety were attached to CBB via a triazole ring, displayed protein-delivery ability (Fig. 2d, Z4, Z5, Z9). We noted the formation of a substantial blue precipitate immediately following the complexation of Cre with any of the CBBZ compounds, more pronounced than with amide-linked CBB compounds. This precipitation may have resulted from stronger protein binding of the CBBZ compounds than the CBB compounds, due to conservation of the CBB sulfonic acids in the CBBZ compounds, followed by colloidal aggregation. Despite this outcome, the robust click-based synthetic methods described here may hold significant promise for expanding the chemical space around lipidoids with protein-binding domains in the future. The Cre recombinase and CBB compounds were tolerated well by the CS cells, as shown by flow cytometric exclusion of DAPI (Supplementary Fig. 5a).

The highest GFP-to-RFP conversion, $58.2 \pm 2.8\%$, was achieved for Cre complexed with CBB14, which contained butyrate tails. Methylation of the ionizable tertiary-amine linker K1 to the positively ionized quaternary ammonium K2 was important for the protein trafficking activity of CBB14, as CBB13, a precursor with the same butyrate tails but with the K1 linker, did not consistently deliver Cre activity into the reporter cells. Alkylation of other CBB compounds that had lipid chains containing cholesterol (CBB1), DHA (CBB3), or palmitate (CBB6), to CBB2, CBB4, and CBB7, respectively, did not improve in vitro delivery of Cre. Notably, CBB21, a CBB G-250 amide without lipid moieties, delivered Cre recombinase into reporter cells and triggered a pronounced GFP-to-RFP conversion. Although we previously found that delivery of Cre recombinase fused to CPP5 was negligible in vitro and in vivo[15], here, delivery of a CPP5-CBB conjugate (CBB19) was robust, producing up to $22.4 \pm 0.6\%$ GFP-to-RFP conversion. CBB18, a combination of CPP5 and TFA, further increased the color conversion to $49.0 \pm 3.6\%$, suggesting synergistic effects between cell-penetrating peptides and CBB. The leading CBB compounds for Cre recombinase delivery, CBB14, 18, 19 and 21, may have a greater positive charge under neutral aqueous conditions, and have less hydrophobic lipid moieties. CBB11-Cre complex consisted of homogenous particles whose size was approximately 1440 nm, whereas CBB14-Cre consisted of two molecular species whose diameters were approximately 167 and 2130 nm (Supplementary Fig. 6). While both CBB compounds aggregated with Cre into large complexes, the presence of smaller species in CBB14-Cre may explain the enhanced Cre delivery activity. Overall positive charge and lower hydrophobicity may be beneficial for Cre delivery, despite the pronounced aggregation. Lower hydrophobicity may increase long-term colloidal stability of the complexes. In our hands, highly hydrophobic CBB1 and CBB2, that were previously proposed as general protein-delivery vehicles via an endocytosis-independent tagging-penetrating mechanism[31], did not deliver Cre into the reporter cells.

## In vivo delivery of Cre

After in vitro nomination of CBB lead compounds, Cre recombinase was delivered with the in vitro hits CBB6, CBB14, CBB11, and CBBox11, or Cre recombinase alone, into the eyes of mT/mG Cre reporter mice via subretinal injection. Analogous to the reporter cell line described

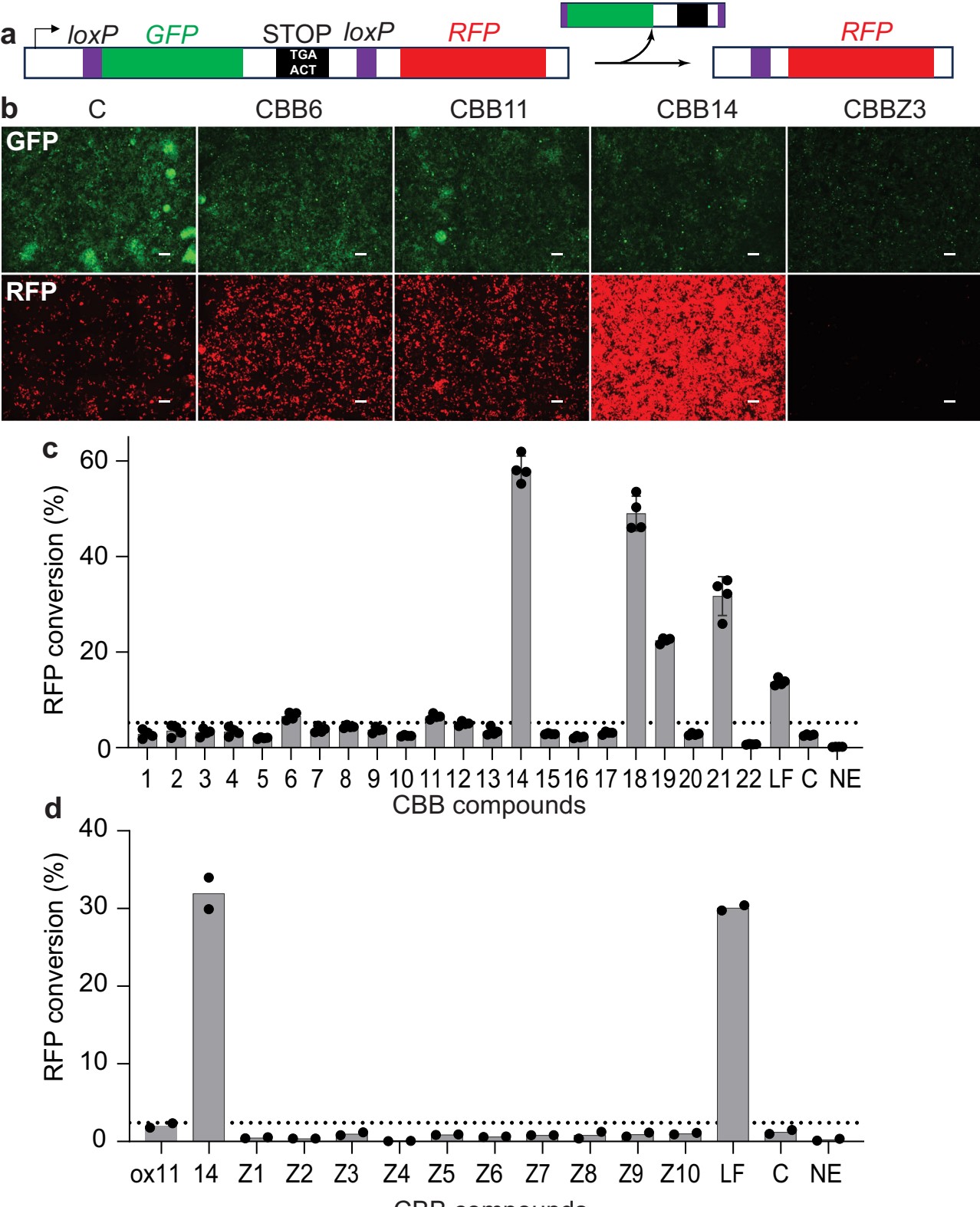

above, the mT/mG mouse is a global fluorescence-reporter strain for monitoring Cre recombinase activity, but with the inverse color change. The *Rosa26* locus in mT/mG mice constitutively expresses membrane-localized tdTomato; however, an upstream polyadenylation signal stops the expression of eGFP (Fig. 3a). Upon excision of the floxed tdTomato cassette by Cre recombinase, eGFP expression replaces tdTomato. This red-to-green color switching in mouse tissues,

including the retina, can be non-invasively monitored by two-photon microscopy of intact eyes. In control mouse eyes, both the retinal pigment epithelium (RPE) and the photoreceptors exhibited uniform tdTomato expression (Fig. 3b). The injection of 20 μM Cre recombinase (850 ng in 1 μl) in buffer with 7.25% DMSO resulted in minimal eGFP expression in the RPE and photoreceptor layers. Injection of 20 μM Cre complexed with 160 μM (208 ng) CBB14 increased eGFP

**Fig. 2 | CBB compounds facilitate delivery of Cre recombinase in vitro into color-switching Cre reporter cells (CS cells). a** Schematic representation of the color-switch Cre reporter. A gene encoding GFP with a terminal stop codon is floxed, allowing constitutive expression of GFP until Cre is introduced. Excision of the GFP-STOP by Cre results in the expression of RFP. **b** Representative fluorescence microscopic images of CS cells treated with Cre complexed with CBB lipidoids. The increase in RFP expression, accompanied by diminishing GFP expression, reflects Cre activity in the cells. C, Cre without CBB. Scale bar, 200 μm. **c**, **d** Quantification of GFP-to-RFP conversion in CS cells treated with the complexes of Cre recombinase with CBB compounds (**c**) by flow cytometry or with CBBZ compounds (**d**) by fluorescence-microscopic images only. [Cre]/[CBB(Z)] = 1:8; Cre delivery mediated by Lipofectamine 3000 (LF) as a positive control is also shown. C, Cre recombinase only. NE, no enzyme. The CBB compounds were considered positive hits if the ratio of red-to-green (RFP/GFP) with flow cytometry (**c**) or with fluorescence intensities of microscopy images only (**d**) was more than twice that of C (the dashed line). Data are presented as (**c**) mean ± SD; $n = 4$ replicates and (**d**) mean, $n = 2$ replicates. Source data are provided as a Source Data file.

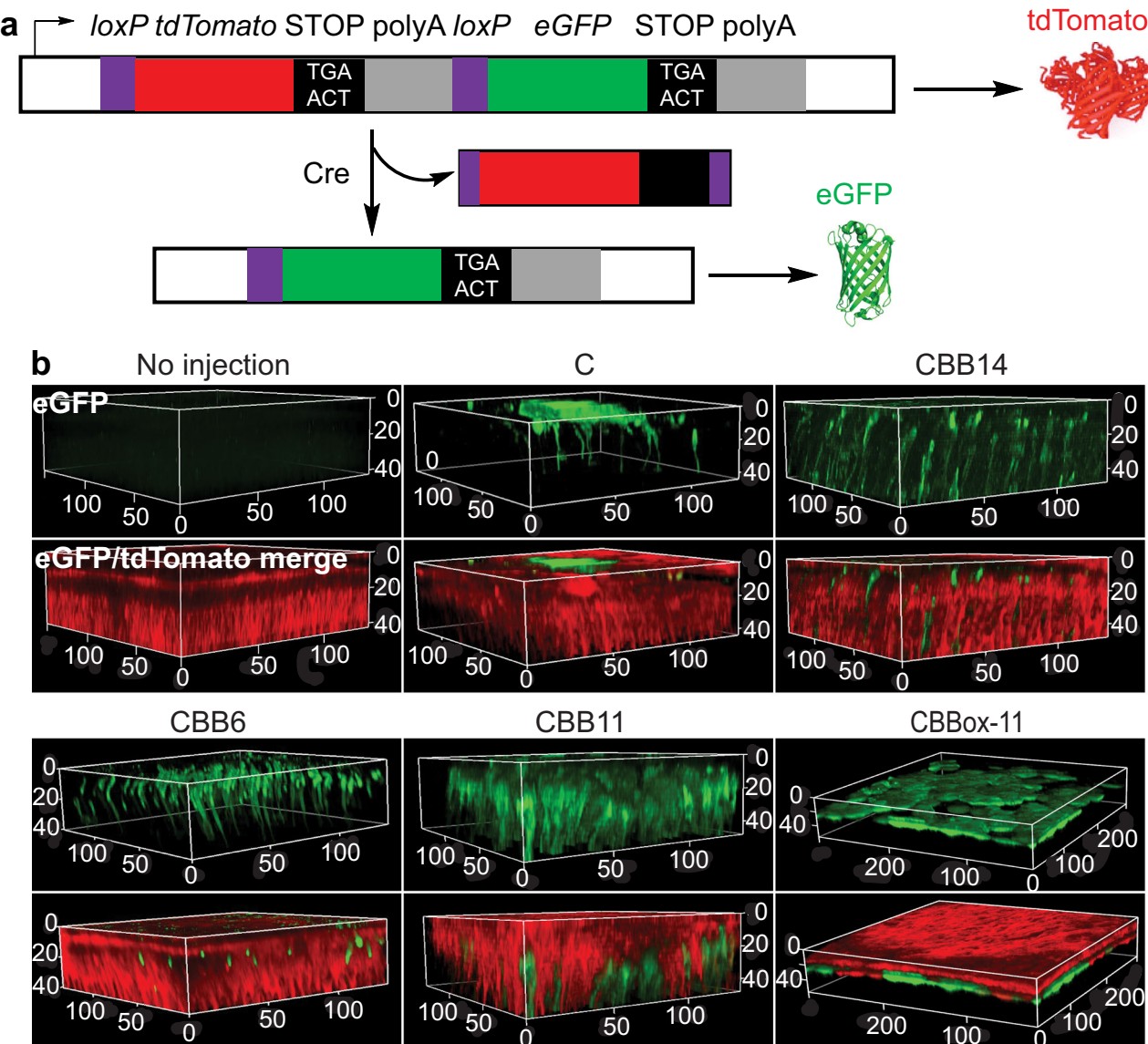

**Fig. 3 | Cell-specific delivery of Cre recombinase mediated by CBB lipidoids in the mT/mG mouse retina. a** Schematic representation of the genetic construct of the mT/mG Cre-reporter-mouse model. The delivery of Cre recombinase to the nucleus triggers excision of the floxed tdTomato and STOP-polyA cassette, enabling the expression of eGFP. **b** Representative images indicating the switch in expression from tdTomato to eGFP, one week after subretinal injection of Cre recombinase complexed with CBB lipidoids. The RPE layer is orientated toward the top. C, Cre recombinase only. The scale is provided in micrometers. Images are representative of at least 3 eyes.

expression in the RPE, producing up to 90% red-to-green color switching (Fig. 3b and Supplementary Fig. 7). CBB6, CBB11, and CBB14 also mediated delivery of Cre into photoreceptors, with CBB14 achieving up to 20% tdTomato-to-eGFP conversion in photoreceptors (Fig. 3b and Supplementary Fig. 7). Notably, we observed that in vitro, the GFP-to-RFP conversion mediated by CBB11 was moderate, while in vivo CBB11 enhanced Cre-delivery-triggered eGFP expression in both outer and inner segments of the photoreceptors, with up to 30% red-to-green conversion. Although in vitro delivery of Cre recombinase mediated by CBBox11, an oxidation product of CBB11, was

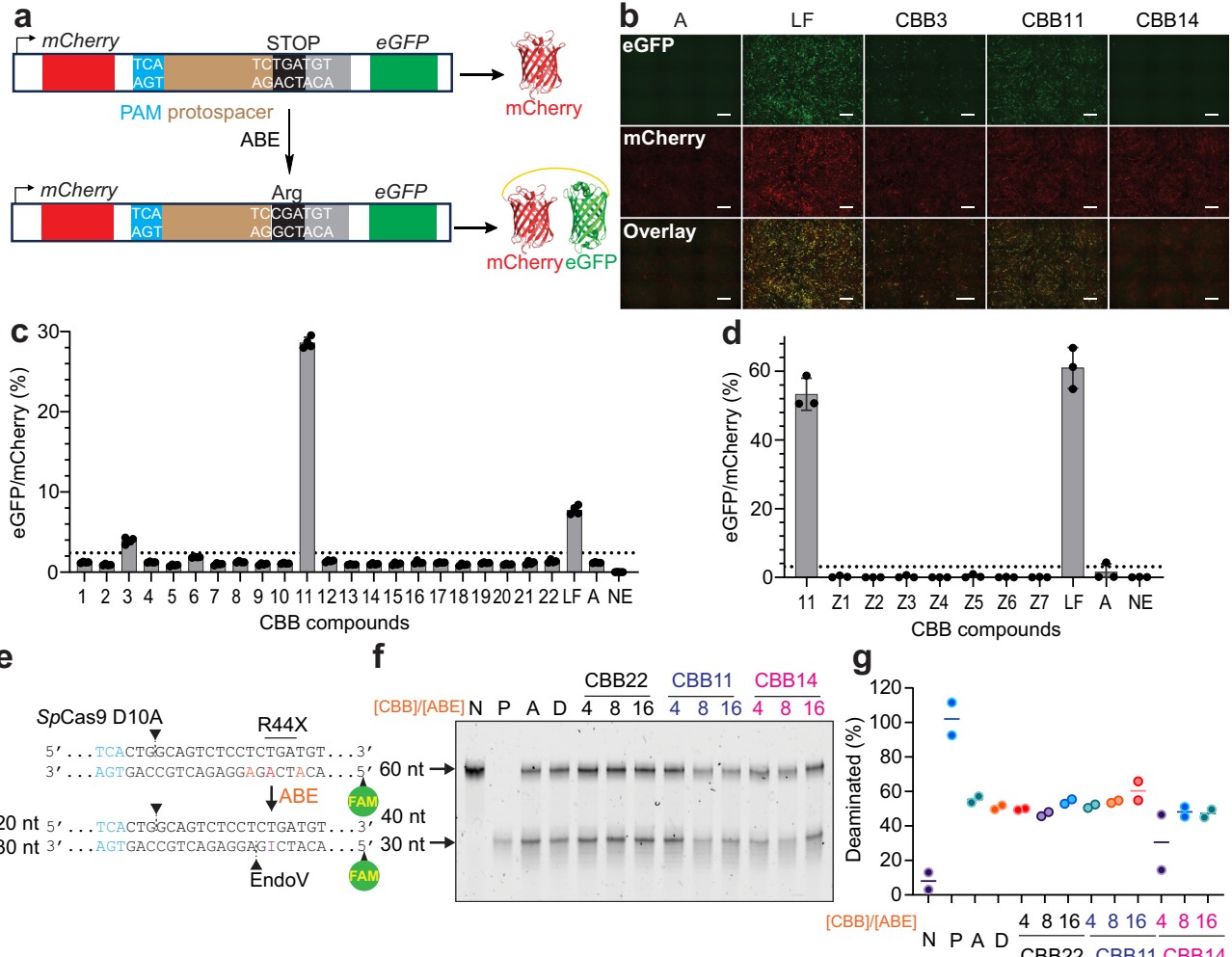

**Fig. 4 | Delivery of ABE-RNPs to *rd12*-reporter cells mediated by CBB lipidoids.**
**a** Schematic diagram of the color-switching construct of the *rd12* reporter-cell line. An intervening sequence from the *Rpe65-rd12* genomic sequence containing the mutation (c.130 C > T; p.R44X) was inserted into a gene expressing both mCherry and eGFP. Expression of eGFP in the reporter cells is prevented by a STOP codon, which can be restored via proper action of ABE. **b** Representative fluorescence-microscopic images of *rd12*-reporter cells 48 h after adding the RNP-CBB complex. Scale bar, 400 μm. **c**, **d** Quantitative assessment of ABE-RNP delivery into *rd12*-reporter cells, mediated by lipidoid complexes of ABE8e with CBB compounds (**c**) by flow cytometry; or CBBZ compounds (**d**) by fluorescence-microscopic imaging only. [ABE8e]/[CBB(Z)] = 1:8. A, ABE8e-RNP only. NE, no enzyme. Quantification of ABE8e RNP delivery mediated by Lipofectamine 3000 (LF) is included as a positive control. The ratio (eGFP/mCherry) reflects the relative editing efficiency. The CBB compounds were considered positive hits if the ratio of green-to-red (eGFP/mCherry) was more than twice that of A (the dashed line). Data are presented as mean ± SD, *n* = 4 replicates (**c**) and 3 replicates (**d**). **e** Schematic diagram of the ABE-activity assay. EndoV = endonuclease V. Blue, SpCas9 PAM; red, target base; orange, bystander base; arrowhead, nick site; FAM, fluorescein. **f** Representative urea-PAGE-gel image of products of ABE deamination cleaved by EndoV. The gel was imaged with a fluorescein filter. N, no enzyme; P, positive control: 60 bp DNA with inosine in the middle; A, ABE-RNP in TAS; D, ABE-RNP in TAS with 10% DMSO. Numbers indicate molar excess of the CBB compound *versus* ABE. **g** Fluorescence quantification of substrate and products of deamination and EndoV cleavage. Two independent samples per data point were assayed in parallel. Source data are provided as a Source Data file.

marginal, it boosted in vivo eGFP expression exclusively in RPE cells to a level comparable to that mediated by CBB14.

## In vitro delivery of ABE-RNP
After confirming the ability of CBB to deliver Cre in vitro and in vivo, we next investigated the potential for CBB-lipidoids to transfer CRISPR genome editors. We used the adenine base editor (ABE) ABE8e as a model. To this end, we used our NIH/3T3 fluorescent reporter *rd12* cell line[32] that contains *mCherry* and *eGFP* separated by a 204-bp *Rpe65* cDNA fragment containing the nonsense *rd12* mutation (c.130 C > T; p.R44X) (Fig. 4a). In this cell line, eGFP expression is suppressed by a stop codon, which can be corrected to an Arg codon by adenine base editing. After the editing, both mCherry and eGFP are expressed, appearing as yellow in merged fluorescent microscopic images.

ABE8e RNP was prepared as previously described[32] and applied to the reporter cell lines with or without delivery vehicles. When complexed with Lipofectamine 3000, ABE-RNPs successfully induced eGFP expression, whereas ABE-RNPs incubated with DMSO alone enabled only marginal conversion of the reporter cells (Fig. 4b). We titrated the ABE RNP with selected CBB compounds (CBB2, 7, 11, 12, 13, 14, 18,19, Z1 and found that higher concentrations of CBB compounds may better facilitate RNP delivery (Supplementary Fig. 4b). However, at higher concentrations we also observed pronounced precipitation of CBB compounds and decided to maintain the CBB:RNP ratio of 8:1 to prevent complications such as clogging of flow cytometric fluidics or injection syringes. We then screened our CBB compounds for ABE RNP delivery ability and found that among CBB1-22 and CBBZ1-Z7, consistent eGFP expression in reporter cells was only achieved by CBB3 (3.9 ± 0.4%) and CBB11 (28.6 ± 0.7%) (Fig. 4b–d). The complexing of

ABE-RNPs with CBB lipidoids with 10% DMSO also did not affect the deaminase activity (Fig. 4e–g)[15]. The common feature of CBB3 and CBB11 are unsaturated lipid domains, DHA (CBB3) and linoleate (CBB11), that contain multiple unsaturated C-C bonds. This may increase the fluidity of the lipid layer and promote fusion of the colloids with the cell membrane. Similarly to Cre, the flow cytometric DAPI exclusion assay showed that the ABE RNP-CBB complexes were tolerated by the rd12 reporter cells (Supplementary Fig. 5b).

We found that the CBB-RNP complexes have poor colloidal stability. Precipitation occurred at high concentrations or upon prolonged incubation, and was associated with a decrease in delivery efficiency. The colloidal instability of CBB11 could have hindered the uptake of ABE-RNPs by the cells or interfered with editing. To improve the distribution of CBB11 in an aqueous environment, we formulated liposomes (Lipo1/3/5) containing, in molar ratio, 5% CBB11, 50% ionizable lipids (either SM102, ALC-0315, or MC3), 10% 1,2-distearoyl-sn-glycero-3-phosphocholine (DSPC), 33.5% cholesterol, and 1.5% 1,2-dimyristoyl-rac-glycero-3-methoxypolyethylene glycol-2000 (DMG-PEG 2000), and a control liposome (Lipo2/4/6) without CBB11 (Fig. 5a). The composition of Lipo2 (50% SM102, 10% DSPC, 38.5% cholesterol, and 1.5% DMG-PEG 2000) resembled the lipid nanoparticles described in our previous work[32]. We noted an improvement of colloidal properties of CBB11 after incorporating it into the liposomes. Upon buffer exchange, free CBB11 was absorbed by a size-exclusion chromatography resin, whereas liposome-bound CBB11 was recovered in the eluate (Supplementary Fig. 8). Dynamic light-scattering measurements indicated that the two groups of liposomes were highly monodisperse (polydispersity index = 0.16/0.19/0.17, and 0.19/0.20/0.15 for Lipo1/3/5 and Lipo2/4/6, respectively), with an average hydrodynamic diameter of 140-155 nm and 155-161 nm, respectively. In contrast, the particle size of CBB11 in the TAS buffer ranged from 400-500 nm (Fig. 5b and Supplementary Fig. 9a–d). We also did not observe precipitation of ABE-RNP with either Lipo1/3/5 or Lipo2/4/6. The cryo-TEM studies on Lipo1 revealed that it consists of unilamellar vesicles with an aqueous core, and that the ABE-RNP is prone to be recruited to the surface of the liposome (Fig. 5e and Supplementary Fig. 10). Furthermore, the presence of ABE-RNP dramatically changed the zeta potential on the Lipo1 surface (a net decrease of 10.16 mV), but marginally affected the zeta potential on the Lipo2 surface (a net decrease of 0.64 mV) (Fig. 5d). No ABE-RNP was observed to be incorporated into Lipo1. The ABE-RNP complexed with Lipo1 remained accessible to Proteinase K (Supplementary Fig. 11). Thus, the protein binding occurred on the surface of Lipo1, and was intensified by CBB11. As expected, this enhanced liposome-protein interaction boosted intracellular protein delivery. Compared to our positive control Lipofectamine 3000, Lipo1 and Lipo2 increased intracellular Cre delivery to CS reporter cells 2.4- and 2.2-fold, respectively, indicated by increased GFP to RFP conversion visualized with fluorescence microscopy (Fig. 5c). At 40 nM RNP, Lipo1 increased the efficiency of ABE8e-RNP delivery into the rd12-reporter cells, as monitored by eGFP fluorescence, 10.5-, 2.9-, and 1.3-fold when compared to CBB11, Lipo2, and LF, respectively (Fig. 5f–i). The RNP delivery enhancement mediated by CBB11 did not depend on the ionizable lipid, as we observed similar enhancement with liposomes made with ALC-0315 and MC3 (Fig. 5g, h and Supplementary Fig. 9e), and with liposomes formulated without ionizable lipids (Supplementary Fig. 12). With 25 nM RNP, the eGFP fluorescence relative to mCherry in reporter cells treated with Lipo3 was 4.4-fold higher than with Lipo4, and the eGFP expression in cells treated with Lipo5 was 31.9-fold higher than with Lipo6 under the same conditions (Fig. 5g, h). We also investigated the delivery of ABE-RNPs into TIGER HEK cells, where functional tdTomato expression is induced by precise A to G editing of the mutated tdTomato at position 7 of the protospacer (Supplementary Fig. 13a)[33]. With 25 nM ABE8e RNP, Lipo1 induced approximately 2-fold higher red fluorescence than our positive control, Lipofectamine 3000. Higher doses of ABE8e led to

decreased red fluorescence, originating from bystander editing at of neighboring adenine A9 that leads to the expression of tdTomato with a missense mutation that decreases its fluorescence (Supplementary Fig. 13a, b). The use of a more precise ABE8e N108Q[34] led to enhanced rescue of tdTomato fluorescence at a broader range of concentrations (Supplementary Fig. 13c). The incorporation of CBB and the ionizable lipid in the liposomes leads to the creation of a broadly applicable macromolecular delivery system optimized for use with ABE RNP. Separately, CBB and ionizable lipids specifically engage the ABE protein and guide RNA, respectively, as shown by the delivery activity of ionizable-lipid-only and CBB-only liposomes built on a scaffold of phospholipid-cholesterol-PEG liposome. Use of CBB and ionizable lipids leads to synergistic engagement of ABE RNP, as demonstrated by the increased potency of Lipo1/3/5 versus Lipo2/4/6. The dispersion of the CBB-RNP complexes likely is achieved by the embedding of the lipid chains of the CBB lipidoids within the liposomes.

## Protein delivery by CBB-liposomes in vivo

Based on the promising results with Lipo1 and Lipo2 in vitro, we assessed liposome-mediated delivery of Cre recombinase to the retina. Lipo1 and Lipo2 were mixed with Cre recombinase to a final concentration of 3.2 mM total lipids and 20 μM Cre. In the Lipo1-Cre complex, the ratio of CBB11 to RNP was set at 8:1. The resulting mixtures were delivered to the eyes of mT/mG-reporter mice via subretinal injections, and tdTomato to eGFP conversion was monitored by two-photon microscopy (Supplementary Fig. S7)[32]. Both Lipo1 and Lipo2-selectively delivered Cre into RPE cells, triggering color change with similar efficiency. In contrast to free CBB11, Lipo1 that contained 5% CBB11 did not deliver Cre to photoreceptors, but to RPE (Fig. 3b and Supplementary Fig. 7). A possible explanation of this change in CBB11-delivery specificity is that liposome endocytosis was mediated by receptors such as low-density lipoprotein receptor-related protein 1 (LRP1) whose expression levels in RPE are much higher than in photoreceptors[35].

Finally, we investigated the non-viral delivery of ABE-RNPs via our liposome formulations for therapeutic genome editing. To assess the delivery of ABE8e-RNPs, we utilized the rd12 mouse model of Leber congenital amaurosis. These mice harbor a nonsense mutation in Rpe65, which effectively eliminates RPE65 expression within the RPE. Without the enzyme to regenerate the visual chromophore 11-cis-retinal, the visual cycle is non-functional, and the electroretinography (ERG) waveform representing the photoresponse to dim-light stimulus is negligible. Effective editing of the Rpe65 gene restores functional RPE65 expression, the visual cycle, and in turn, induces the ERG response[36-38].

To assess the editing capabilities of ABE8e-RNPs in conjunction with our CBB lipidoids, we measured restoration of functional vision mediated by several delivery vehicles. Two weeks after subretinal injections of 20 μM ABE-RNPs in these vehicles, rd12 mice were dark-adapted overnight, and their scotopic ERG responses were recorded (Fig. 6a). We did not observe an ERG-flash response in untreated rd12 mice, while mice treated with ABE-RNPs in TAS buffer or in TAS with 7.25% DMSO regained recordable ERG responses. The scotopic b-wave amplitudes in response to a 0.5 cd s m$^{-2}$ stimulus were $45.8 \pm 12.9\ \mu V$ and $50.6 \pm 33.9\ \mu V$, respectively. CBB11 and CBB14 complexed to ABE8e performed marginally better than untagged RNPs ($60.8 \pm 14.9\ \mu V$ and $58.5 \pm 37.4\ \mu V$, respectively). These results indicate that our observations in vitro with ABE-RNPs with CBB11, or in vivo with Cre recombinase delivery by CBB14, are not directly translatable to ABE-RNP delivery in vivo. This disconnect could originate from the previously observed propensity of CBB-RNP complexes to aggregate, which, together with the intrinsic propensity of ABE to dimerize[15,39,40] might exacerbate the CBB-induced oligomerization and complicate RNP delivery to a larger extent than observed for Cre. Consistent with in vitro observations for the CBB-RNP complexes, the liposome

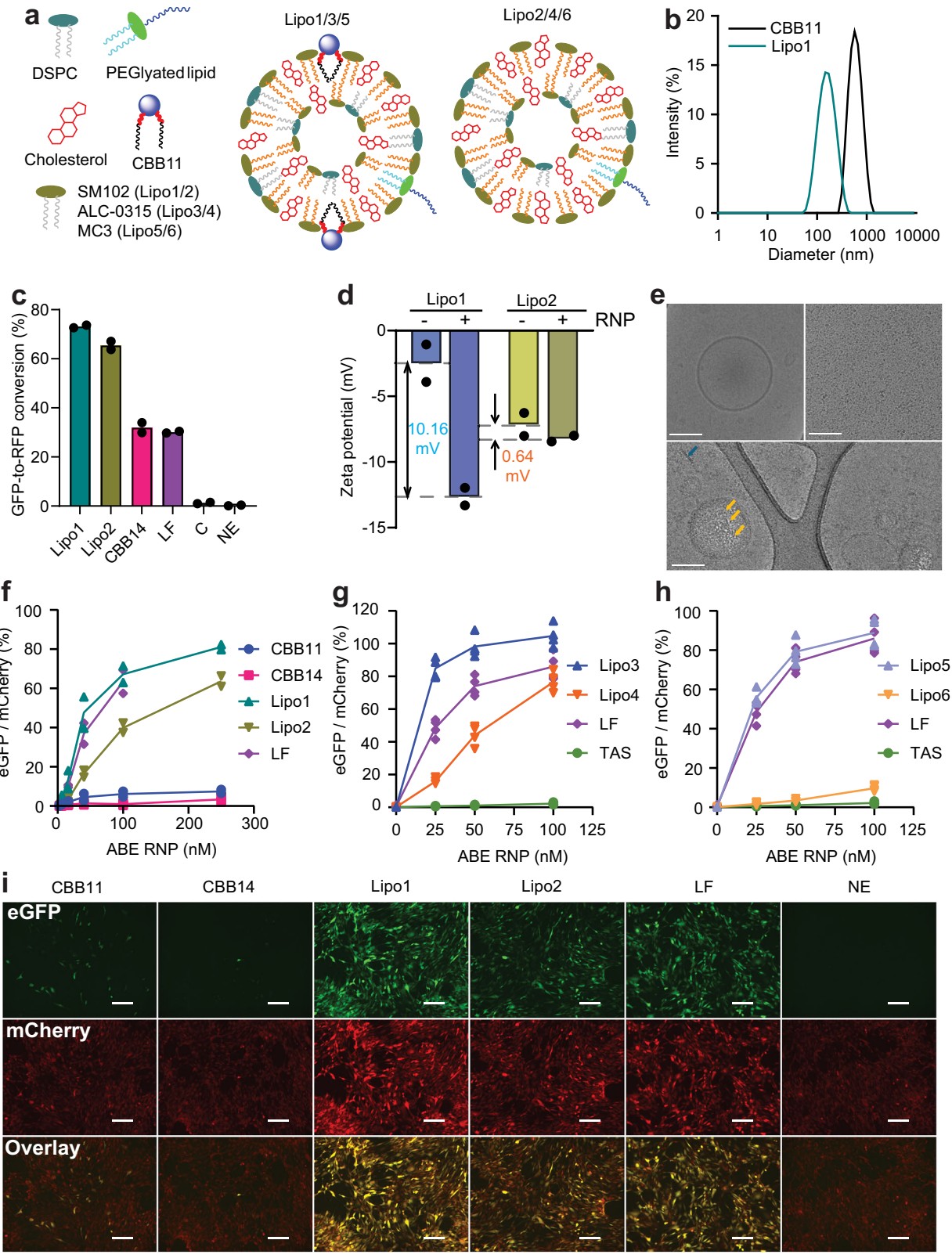

formulations outperformed the non-liposomal CBB formulations. The restoration of ERG responses was robust, especially when CBB11-containing Lipo1 was used. The b-wave amplitudes were measured at $227 \pm 69\,\mu V$ for Lipo1, and $95.3 \pm 62.5\,\mu V$ for Lipo2. Notably, the best example of Lipo1-mediated editing achieved a b-wave amplitude of $353\,\mu V$, estimated to be 76% of the ERG response of wild-type mice. These results support the proposed synergy between nucleic acid

engagement by the liposome and protein engagement by the CBB that enables the therapeutic effect of our RNP formulation.

We further analyzed the outcomes of the phenotypic rescue by performing retinoid analysis of eyes treated with ABE-RNP complexed with Lipo1 or Lipo2. After 24 h of dark adaptation, 11-*cis*-retinal levels reached approximately 40 and 16 pmol/eye for Lipo1 and Lipo2, respectively, estimated to be 10.5 % and 4.2% of the level for wild-type

**Fig. 5 | Liposome-facilitated delivery of Cre- and ABE-RNPs to reporter cells.**
**a** Pictorial depiction of the liposome formulation. **b** The size distribution of the CBB11-alone (black) and CBB11-doped liposomes (Lipo1, dark green), in TAS buffer. Average of 3 replicates. **c** Fluorescence microscopic quantification of the recombination activity triggered by Cre-recombinase delivery, mediated by Lipo1 (dark green) or Lipo2 (dark yellow). CBB14 (red) and Lipofectamine 3000 (LF, purple) served as positive controls. C, Cre recombinase only; NE, no enzyme. Data are presented as the mean of 2 replicates. **d** The zeta potentials of Lipo1 (blue) and Lipo2 (yellow) in the absence (-, light color) or the presence (+, dark color) of ABE-RNP. The ratio of ABE-RNP to CBB11 is 1:8. Means of two biological replicates. **e** Representative images of cryo-TEM of Lipo1 (upper right), ABE-RNP only (upper left) and Lipo1 with ABE-RNP (lower). The blue arrows point to ABE-RNP aggregates, and the dark points labeled with the yellow arrow are assumed to be the protein on the liposome surface. Scale bar, 100 nm. *N*, 8 replicates. **f** Titration of in vitro

delivery of ABE8e-RNPs mediated by CBB11 (blue), CBB14 (red), and Lipo1 (dark green) - 5 h after the RNP was complexed with the CBB compounds and liposomes. The ratio [CBB]/[RNP] = 8:1. The total lipid amounts of Lipo 2 (dark yellow) and Lipofectamine 3000 (LF, purple, as control) were comparable to those for Lipo1. Data are presented as mean ± SD; *n* = 2 replicates. **g, h** Titration of in vitro delivery of ABE8e-RNPs mediated by Lipo3 (dark blue)/Lipo4 (orange) (**g**); and Lipo5 (light purple)/ Lipo6 (yellow) (**h**), respectively, after allowing 15 min for complex formation in each case. The ionizable lipid concentration was always 80-fold greater than the concentration of ABE-RNP. ABE-RNP in TAS buffer (TAS, green) served as a negative control. Data are presented as mean ± SD; *n* = 4 replicates. **i** Fluorescence-microscopic images of *rd12*-reporter cells treated with liposome-RNP complex. The RNP concentration was 100 nM. Representative of 2 biological replicates. Scale bar, 200 µm. Source data are provided as a Source Data file.

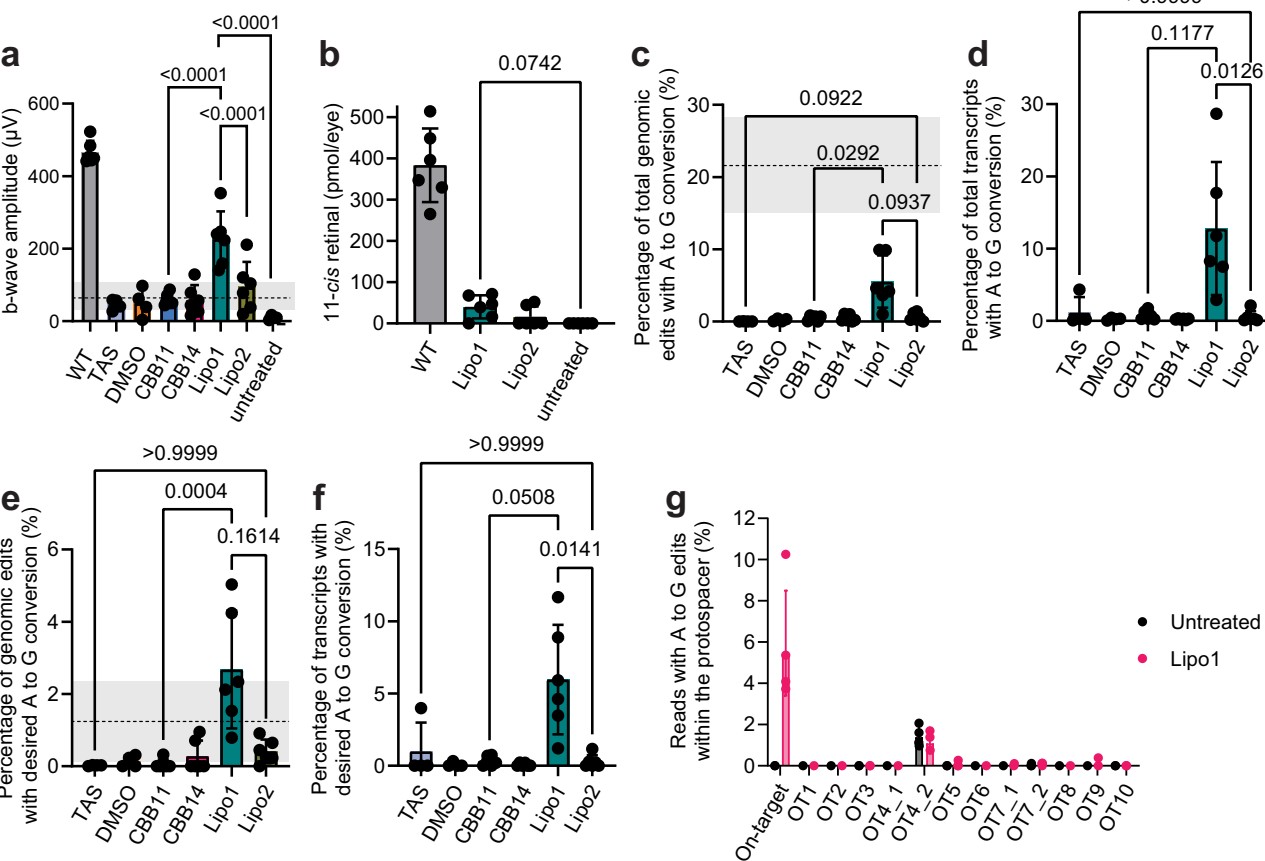

**Fig. 6 | ABE-RNP delivery mediated by CBB-containing liposomes in *rd12* mice.**
Four-week-old *rd12* mice were treated via subretinal injection with 20 µM ABE8e-RNPs with indicated reagents, with 8-fold molar excess of CBB and 80-fold molar excess of SM-102, where applicable. **a** B-wave amplitudes of ERG responses of *rd12* mice recorded two weeks after treatments. Data are presented as mean ± S.D., *n* = 4 eyes (TAS, DMSO, untreated) or 6 eyes (WT, CBB11, CBB14, Lipo1, Lipo2); one-way ANOVA with Šídák's multiple comparisons test. **b** Quantification of the visual chromophore 11-*cis* retinal in the eyes from ABE-treated *rd12* mice obtained after a 24 h dark adaptation. Kruskal-Wallis test with Dunn's multiple comparisons test, *n* = 6 eyes. **c**–**f** Quantification of (**c**) total DNA editing, (**d**) total cDNA editing, (**e**)

precise DNA editing and (**f**) precise cDNA editing in the retinal pigment epithelium isolated from the treated *rd12* mice. Mean ± S.D., *n* = 4 (TAS, DMSO) and 6 eyes (CBB11, CBB14, Lipo1, Lipo2); Kruskal-Wallis test with Dunn's multiple comparisons test. **g** Off-target adenine genome-editing in *rd12* mouse RPE after delivery of ABE-RNPs by Lipo1. *N* = 4 isolates picked randomly from samples analyzed in panels (**c**, **d**). The data for ERG responses (**a**) and genomic edits (**c**, **d**) were compared with the published data for the *rd12* mice treated with eVLP-ABE-RNPs.[47] The dashed line and the gray band represent the average and the range of the corresponding ERG (**a**) or genomic editing (**c**, **d**) values in *rd12* mice treated with eVLP-ABE-RNPs, respectively. Source data are provided as a Source Data file.

mice (Fig. 6b). We also demonstrated precise editing of the *rd12* mutation in the DNA and transcripts isolated from RPE samples obtained from the mice whose ERG measurements are reported in Fig. 6a. Next-generation sequencing revealed up to 9.9% total adenine editing in *Rpe65* genomic amplicons (Fig. 6c), which was correlated with up to 28.8% sequence editing of the *Rpe65* transcripts (Fig. 6d) in

eyes treated with Lipo1. Strikingly, although ABE8e has the propensity to deaminate multiple adenosines within its activity window even upon transient delivery via eVLPs,[38] in this case, when delivered in complex with Lipo1, we recorded precise A-to-G corrections by ABE8e up to 5.0% in gDNA sequences (Figs. 6e) and 11.7% in transcripts (Fig. 6f). Precisely edited gDNA sequences in mice injected with ABE-RNPs

delivered by Lipo1 were 127-, 20-, 50-, 9.7-, and 6.5-fold higher than those with TAS, DMSO, CBB11, CBB14, and Lipo2, respectively. These findings indicate that the benefits of augmenting liposomes with CBB11 were corroborated by improved visual response and increased editing of the genome.

We also compared the ERG responses and genomic editing of *rd12* mice treated with Lipo1-ABE8e-RNPs to the analogous measurements for mice treated with ABE8e-eVLP described previously (Fig. 6a, c, e)[5]. The ABE8e-RNP carried by eVLPs achieved $21.6 \pm 6.6\%$ total genomic editing, outperforming lentiviral delivery ($11.6 \pm 4.5\%$). Although the genome editing by Lipo1-ABE8e-RNP in this study ($5.5 \pm 3.3\%$) is less efficient, both the precise genomic correction rate ($2.7 \pm 1.5\%$) and ERG response ($227 \pm 69\ \mu V$) in *rd12* mice treated with Lipo1-ABE8e-RNPs are almost 2.2- and 2.3-fold greater, respectively, than those in mice treated with eVLP-ABE8e-RNPs. We attribute this increase of vision-function recovery in *rd12* mice to a more transient gene editing in RPE cells by the Lipo1-ABE8e-RNP system, effecting more precise correction. Furthermore, off-target editing analysis by CIRCLE-seq revealed A to G substitutions at background levels in the *rd12* mice treated with Lipo1-ABE8e-RNPs, comparable to those observed in untreated animals (Fig. 6g).

To evaluate the safety of the ABE-RNP delivery to the retina mediated by Lipo1, we also administrated ABE-RNP-Lipo1 complex to WT mice (Supplementary Fig. 14) via subretinal injection. Since the oxidation of CBB11 might induce epoxides in the liposome, only freshly formulated Lipo1 was applied to the WT mice. Different batches of Lipo1 showed similar morphology and comparable ABE-RNP delivery efficiency to *rd12* reporter cell lines (Supplementary Fig. 14a, b). In vivo retinal imaging by optical coherence tomography (OCT) indicated no substantial alterations to the retinal structure of WT mice administrated the complex of RNP ($20\ \mu M$) and Lipo1. We noticed the presence of hyperreflective spots visible in the vitreous, indicating possible inflammation[41], which could be ameliorated by lowering the RNP concentration (Supplementary Fig. 14c). The ERG response and the generation of the visual chromophore, 11-*cis* retinal, of WT mice treated with Lipo1 and RNP ($10\ \mu M$) were also comparable to those of untreated WT mice (Supplementary Fig. 14d, e). In parallel experiments in which *rd12* mice were treated with the same batch of Lipo1 and RNP complex, subretinal injections of 1 µl of Lipo1 with 10 µM RNP achieved optimal ERG response and visual chromophore recovery (Supplementary Fig. 14d, e), comparable to those of *rd12* mice treated with Lipo1-ABE-RNP ($20\ \mu M$) (Fig. 6). These results suggested that optimal vision-function rescue requires a balance between the biological tolerance of the cells and the editing efficiency of the gene editors.

## Discussion

In this work, we addressed the challenge of efficient genome-editor delivery by generating two families of CBB-derived lipidoids, each with distinct synthetic routes that build on existing knowledge while enabling improved functionality. Notably, CBB is already FDA-approved for ocular surgery (marketed as ILM Blue and Brilliant Peel), which may help mitigate safety concerns and bolster the potential translatability of our reagents. In one approach, we employed sulfonic amidation of CBB (Supplementary Fig. 2), introducing ionizable linkers and varying acyl chains. This strategy expanded on earlier cholesterol-CBB conjugation methods that, in our hands, revealed discrepancies in reported mass spectrometry and NMR data, emphasizing the need for thorough validation and characterization[31]. The resulting lipidoids differed by chain length, degree of saturation, and linker charges, and related byproducts were also isolated (*e.g.*, CBB5, CBB16, CBBox11), highlighting the importance of meticulous purification and characterization to ensure a uniform product profile (see Supplementary Information). Meanwhile, our second approach (Supplementary Fig. 3) relied on an alkyne-bearing

CBB derivative suitable for copper-catalyzed azide-alkyne cycloaddition (click reaction). By preserving the dye's native sulfonic acids, we generated triazole-linked CBBZ derivatives that could be easily tailored for diverse payloads. Although these triazole-based lipidoids were less efficient in our preliminary gene-editing assays, their modular click-based design offers significant advantages for rapid library expansion and fine-tuning of protein-binding affinities. Taken together, the two synthetic routes provide complementary paths for systematically exploring CBB-based carriers to create custom-designed lipidoids capable of improving the delivery of a wide range of BEs and PEs in future clinical and research applications.

Several additional key observations are pertinent to ongoing studies and future development of this platform. First, Cre and ABE delivered under similar conditions displayed markedly different uptake efficiencies, likely reflecting the distinct sizes, charges, and structural features of these proteins. As mentioned above, due to their physical nature as lipidoids, the CBB-lipid conjugates are prone to form colloids in neutral aqueous environments. The complexing of colloids with proteins was also modulated by the distinct sizes, charges, and structural features of these proteins. The CBB lipidoids facilitated the cellular uptake of a small positively charged protein, Cre, which led to DNA recombination in the reporter CS cells and in mT/mG mice. Remarkable Cre-recombinase delivery mediated by CBB14 was achieved in vitro (Fig. 2c) and in vivo (Fig. 3b). The colloidal stability and protein-delivery efficiency of CBB compounds dropped dramatically when they were bound to the large, aggregation-prone and negatively charged ABE-RNP (Fig. 4c). Modulating the membrane permeability of the CBB lipidoids with polyunsaturated fatty acids, as in the case of CBB11, could moderately bolster the ABE-RNP uptake by *rd12* reporter cells. The CBB-RNP complex contained 7.25% DMSO, which was well tolerated by ABE and, as such, may be further explored as a vehicle for protein delivery. Secondly, the efficiency of CBB-lipidoid–mediated delivery can vary widely among different cell types, both in vitro and in vivo, suggesting that the interaction of these compounds with cellular membranes and the subsequent processing pathways are context-specific. Even subtle alterations in the CBB core or lipid-chain structure can dramatically shift delivery targets (Fig. 3b). For instance, tdTomato-to-eGFP conversion in the mT/mG mice treated with Cre recombinase complexed with CBB6 or CBB11 occurred dominantly in photoreceptors (Fig. 3b), but a mild oxidation event in CBB11 switched the tissue specificity of subretinally injected Cre recombinase from photoreceptors to the RPE. The lipid tail length of these two CBB lipidoids ranges from C16 to C18. Coincidentally, the dominant fatty acid moieties in rat and human retina are stearoyl (C18) and palmitoyl (C16). This correlation implies that the Cre delivery mediated by CBB6 and CBB11 in the retina might utilize existing lipid metabolism pathways[42]. Thirdly, in vivo experiments diverged to some extent from in vitro results; for example, some lipidoids that performed modestly in cell culture were significantly more potent in subretinal injections, underscoring the complexity of tissue-level parameters like local microenvironment or differences in membrane composition. It is possible that increasing the search space through high-throughput in vivo liposome screening would provide insight and help deconvolute these possibilities[43]. Two-photon imaging showed that CBB-lipidoids were well tolerated in the eye, an important consideration given the sensitivity of retinal tissues. While CRISPR-Cas9 nuclease RNPs could be viewed as a logical intermediate between Cre and base editors, we prioritized ABE for in vivo studies because in comparison with Cas9 nuclease, ABE has the advantage of catalyzing single-base transitions without creating double strand breaks, with equivalent potential for RNP delivery as Cas9[44]. The molecular interactions that we engage to deliver ABE RNP are applicable to other RNPs, and thus, we envision that our CBB liposomes are applicable to other CRISPR nucleases and precise editors as well.

Several of the CBB-lipidoids substantially enhanced the delivery of both Cre recombinase and ABE proteins to mammalian cells in vitro and in vivo. Particularly striking was the high efficacy observed when these compounds were formulated into ionizable cationic liposomal particles to improve the colloidal stabilities of the CBB-protein complexes. We designed our CBB liposomes with a goal of maintaining the enzymatic activity of ABE RNP by minimizing denaturation. This imposed a requirement to avoid acidic pH and ethanol, both of which are crucial for achieving efficient encapsulation of nucleic acids and RNPs within the LNPs. As a first step, we prepared lipid films containing all the lipid constituents: liposome scaffold of DSPC, cholesterol, and DMG-PEG 2000, ionizable lipid (SM102, ALC-0315 or MC3) to bind guide RNA, and CBB11 to bind protein. Next, instead of dissolving the film in ethanol and complexing the RNP in a microfluidic system, we formed the liposomes in our mild dispersant, TAS buffer, and made liposome-RNP complexes by gentle mixing in the pH neutral, aqueous environment of TAS. This procedure, which resembles traditional transfection workflows, led to adsorption of the RNP on the liposome surface. This enzyme-focused approach led to striking editing efficiencies and clinically relevant restoration of vision in mice. For instance, Lipo1 containing 5% CBB11 achieved up to 9.9% genomic and 28.8% transcript-level edits of the *Rpe65* mutation in *rd12* mice, a correction sufficient to restore vision to an estimated 76% of the wild-type ERG response (Fig. 6). These results surpass those reported for virus (AAV, LV) or virus-like particle (VLP) delivery systems[37,38,45–47]. The magnitude of ERG response reported here for ABE8e-RNP complexed with CBB liposomes is almost twice those reported for ABE8e-eVLP,[38] which more directly correlated the precise genomic edits by ABE8e-Lipo1 complex (2.15-fold of ABE8e-eVLP) than the total genomic editing efficiency (25% of ABE8e-eVLP). On the other hand, the increase in concentration of ABE8e-Lipo1 complex from 25 nM to 50 nM caused a steep drop of red fluorescence in TIGER HEK cells. We speculate that the redundant cytosolic ABE8e-RNPs increased bystander editing at A9, which in turn resulted in expression of mutated tdTomato with intrinsically lower fluorescence (Supplementary Fig. 13). Expression of the more-fluorescent native tdTomato in the HEK cells, to the level comparable to that effected by 25 nM ABE8e-RNP-Lipo1, could not be achieved by administering the more precise editor ABE8e N108Q-RNP-Lipo1. These data re-emphasize that the balance between the efficiency and the precision of gene editing is critical for optimal rescue of vision function in *rd12* mice and other models of disease.

Liposomes and LNPs represent the most advanced drug-delivery systems for protein intracellular delivery. In LNPs, the medicine is encapsulated in the lipid particles requiring high lipid to protein ratio. In our previous work, ABE-RNP encapsulated in SM102-LNP (lipid:RNP ratio 6.56:1 by weight) was administered to *rd12* mice via subretinal injection, resulting in 0.30% precise genomic correction. In this study, we developed a family of lipidoids incorporating a CBB protein-binding domain, which enabled us to decorate the corona layer of the nanoparticle with gene-editing RNPs. Cryo-TEM images indicated that a majority of the RNPs were associated with the liposome-corona layer, except the protein aggregates. With similar amounts of lipids, the concentration of ABE-RNP subretinally injected into the same model mice was elevated 10-fold to 20 μM (lipid: RNP ratio 0.45: 1 by weight). Correspondingly, the precise A to G genomic edits were boosted almost 9-fold to 2.67% (Fig. 6d). Noticeably, the proportion of only desired genomic edits to the total corrections triggered by CBB-liposomal delivery increased to 48%, which is also much higher than that mediated by VLP delivery or LV-viral delivery. It should be noted that the RNP on the liposome surface is more vulnerable to proteolytic degradation and immune detection compared to RNP encapsulated within the LNP (Supplementary Fig. 11). On the other hand, RNP on the liposome surface does not need to break through the lipid layers of lipid particles before entering the nucleus. In general, the delivery of ABE-RNPs mediated by CBB-liposomes is a more transient mode of

delivery of gene-editing tools, consistent with decreased bystander editing[15,38]. The CBB-supplemented liposomes bolstered the editing efficiency in *rd12* mice at least 6-fold compared to control liposomes. We attribute the enhanced editing to increased RNP recruitment to the liposome surface, and speculate that the thermodynamics of this interaction could minimize RNP exchange behavior between the local environment and the liposomes. We envision that delivery efficiency could be further enhanced by modulating the CBB structure and tuning the liposome formulation. Moreover, CBB-supplemented liposomes could be retargeted in a cell-specific manner by adding a ligand to the surface of the liposomes; this approach could enhance specific interactions with cell-specific receptors to achieve high-affinity interactions and specificity. The cargo of CBB-liposomes could also be partially replaced by nucleic acids, steroids, or fluorescent markers due to the broad binding ability of the CBB moiety.

Despite these advances, important limitations remain to be addressed. First, the safety profile of the liposome-protein complex should be further investigated. In this study, ABE-RNP or Lipo1 were relatively well tolerated by the mouse retina; however, an induced immune response in the vitreous was observed in mice subretinally injected with high concentrations of Lipo1 with and without RNP (Supplementary Fig. 14c). Even though the immune response could be mitigated by lowering the dose, there was a corresponding decrease in gene-editing efficiency. A combination of genomic therapy with immune-suppression thereby should be investigated in future studies. Secondly, although CBB-lipidoids showed low toxicity in ocular tissues, long-term investigations are essential to determine whether repeated dosing, extended release or prolonged exposure are feasible in clinical scenarios. Thirdly, the inherently hydrophobic and self-aggregating nature of many lipidoid conjugates complicates both the synthesis and the treatment efficacy. Larger compound libraries, guided by the principles and lead structures presented here, will be necessary to systematically optimize properties such as solubility, membrane fusion, and endosomal escape. In addition, transduction of photoreceptors remains a major challenge, as these cells are typically less phagocytic than RPE cells. Our preliminary assessment of Cre delivery by CBB hints that further optimization of CBBs could increase photoreceptor delivery. Excitingly, our Cre formulations transduced the photoreceptors in mT/mG mice, and a follow-up study in models of photoreceptor degeneration suitable for ABE or PE will clarify the potential of the CBB lipidoids and liposomes to deliver larger RNP cargo to the photoreceptors. The chemical properties of the ligands presented on the surface of the particles and the colloidal stability of CBB-RNP complexes may determine whether the nanoparticles will be absorbed by the photoreceptor cells. Lastly, an additional practical determinant is particle size. Dynamic-light-scattering measurements indicated that CBB11/Cre and CBB14/Cre form ~ 1.4–1.5 μm assemblies, with a significant fraction of CBB14-Cre forming smaller, 160-nm particles, whereas CBB11 alone forms ~ 400–500 nm colloids; importantly, CBB11-doped liposomes (Lipo1–6) are ~ 140–160 nm and highly monodisperse. Micrometer-scale complexes are unlikely to diffuse broadly within the neural retina or traverse ocular barriers from systemic delivery, whereas systemic delivery of the smaller, narrowly distributed liposomal formulations would be effective. Notably, despite their large hydrodynamic diameters, CBB11 and CBB14 still delivered Cre via subretinal injection. Future work will focus on compositional adjustments to the liposome core or the lipid tails to maximize uptake by the photoreceptors.

Taken together, these findings highlight the potential utility of CBB-based lipidoid platforms in gene-editor delivery. By coupling a promiscuous protein-binding domain to membrane-penetrating lipid moieties, and by formulating these conjugates into stable liposomes, we achieved robust Cre and ABE delivery both in vitro and in vivo. CBB derivatives are approved to be administered intraocularly as an adjunct for retinal surgery, and CBB derivatives have also been well

profiled toxicologically, leading us to believe that our CBB lipidoids could be similarly applied translationally for retinal genetic surgery. Moving forward, our modular click-based synthetic framework will facilitate efficient incorporation of different lipids or lipidoid-building blocks, enabling systematic optimization of physicochemical features for targeting diverse tissues or cargoes. Our results lay the groundwork for further exploration of CBB-lipidoid liposomes as a class of gene-editor vectors, with the potential to advance precise, efficient genome-editing therapies for inherited retinal diseases, and beyond. Through further chemical optimization, comprehensive toxicity studies, and expanded application to diverse targets and tissues, this platform has the potential to surmount critical delivery barriers, thereby accelerating the clinical translation of genome editing for vision preservation and restoration. Moreover, our innovative approach extends beyond ocular applications, holding promises to significantly impact the broader fields of gene therapy and personalized medicine by providing a versatile and efficient delivery system for a wide range of therapeutic agents.

## Methods

### Animals

The C57BL/6 J ("WT", JAX 000664), B6(A)-Rpe65 rd12/J (rd12) (JAX 005379), and B6.129(Cg)-Gt(ROSA)26Sortm4(ACTB-tdTomato,-EGFP) Luo/J (mTmG) (JAX 007676) mice were purchased from Jackson Laboratories (Bar Harbor, ME, USA). mT/mG mice were crossed with BALB/cJ albino mice (JAX 000651) to establish an albino mT/mG line used in this study. The mice of both sexes between 4–13 weeks of age with a body weight of 15–25 grams were used for the experiments (see Source data). The mice were housed in the vivarium at the University of California, Irvine, where they were maintained on a normal mouse chow diet and a 12 h light ( <10 lux)/12 h dark cycle. The temperature ranged from 75-76 °F, humidity from 30-40%, and animals were given food and water ad libitum. No sex-based stratification or sex-based analysis was performed in this study. Both male and female mice were used. The research applies to both sexes. The functional readout is electroretinography (ERG), and there are no documented differences in ERG responses between male and female mice[48]. We did not find literature evidence of the influence of sex on genome editing in mice. Here, the cohorts are too small to conduct sex-based analysis.

### Mammalian cell culture

HEK293-loxP-GFP-RFP cells (referred to as "CS cells" GenTarget Inc., San Diego, CA, USA, SC018-Bsd), reporter-rd12 cells (generated previously in Palczewski laboratory)[15] and TIGER HEK cells (generated previously in Palczewski laboratory)[33] were maintained in DMEM/F-12 medium with GlutaMAX supplement (Thermo Fisher, Waltham, MA, USA, 10565018), or in DMEM with glutamine (Thermo Fisher, 11965092), both supplemented with 10% fetal bovine serum (FBS) (Genesee Scientific, San Diego, CA, USA, 25-514H) and optional 100 U ml$^{-1}$ penicillin-streptomycin (Thermo Fisher, 15140122) (complete medium) in a humidified incubator at 37 °C, 5% $CO_2$.

The cells were rapidly thawed from liquid nitrogen storage and passaged at least twice before use in experiments. For protein-delivery experiments, the cells were seeded 24 h prior to the experiment in 24-, 48-, and 96-well plates to reach 50–70% confluence (approximately 100,000, 50,000 and 25,000 cells per well, respectively, for the CS cell line; and 50,000, 24,000 and 10,000 cells per well, respectively, for rd12 reporter cells).

### General experimental details for chemical synthesis

All reactions were performed in oven-dried glassware, under an inert atmosphere with exclusion of moisture from reagents and solvents. Reagents were used as supplied. Liquid chromatography was performed using flash chromatography on silica gel (230-400 mesh), on a CombiFlash NextGen 300+ system, using eluting solvent (reported as a

V:V-ratio mixture). Analytical thin layer chromatography (TLC) was performed on 0.25 mm glass-backed Silicycle 60 F254 plates. Visualization of the developed chromatogram was accomplished with UV light (254 nm). Detailed procedures and analytical data are in the Supplementary Information.

### Mass spectrometry

The Coomassie compound solution in methanol (Fisher Scientific) was diluted to approximately 5 μM and centrifuged at 17,000 g for 15 min. Five microliters of the supernatant were injected into a Vanquish HPLC system coupled with a Q-Exactive Mass spectrometer (Thermo Fisher Scientific, Waltham, MA). The separation was carried out on a BEH C18 Colum (1.7 μm, 2.1 × 50 mm, Waters, Milford, MA) using a mobile phase consisting of 0.1% formic acid (Thermo Fisher) in water (A) and 0.1% formic acid and 50% 2-propanol (Thermo Fisher) in acetonitrile (B) at a flow rate of 300 μl min$^{-1}$ with a gradient of 30% B at 0-1 min, 30-100% B at 1–5 min and 100% B at 5–8.5 min. The normalized collision energy (NCE) was set at 40. The data was extracted using Xcalibur 1.4 (Thermo Fisher Scientific, Waltham, MA) and processed with FreeStyle 1.8 P2 (Thermo Fisher Scientific) and GraphPad Prism 10 (GraphPad, San Diego, CA).

### Nuclear magnetic resonance

$^1$H and $^{13}$C-NMR spectra were recorded on a Bruker Ascend Avance III HD™ spectrometer operating at 400 MHz, 500 MHz, or 600 MHz. Chemical shifts reported in $\delta$ units (part per million (ppm)) with reference to the residual solvent peak $CD_3OD$ (MilliporeSigma) ($\delta$ 4.87, 3.33) for $^1$H and ($\delta$ 49.15) for $^{13}$C spectra. Bruker TopSpin 4.4.1 software was used to collect the data, and Mnova 14.3.1 was used to analyze the results. NMR data are presented in the following order: chemical shift, peak multiplicity (s = singlet, bs = broad singlet, d = doublet, t = triplet, q = quartet, dd = doublet of doublets, m = multiplet), coupling constant (in Hz).

### Expression and purification of Cre recombinase

Plasmid encoding Cre recombinase for expression in *Escherichia coli* was a gift from Niels Geijsen (Addgene plasmid # 62730; http://n2t.net/addgene:62730; RRID:Addgene_62730). The plasmid sequences were viewed using Snapgene 8.2.1 software. The Cre plasmid was transformed into *Escherichia coli* BL21star (DE3) bacteria (Thermo Fisher Scientific, C601003) and plated on LB agar (MilliporeSigma) plates, with 50 μg ml$^{-1}$ kanamycin (Goldbio, St. Louis, MO, USA, K-120-SL25); selected clones were grown in Terrific Broth (Thermo Fisher Scientific, 22711-022) containing 50 μg ml$^{-1}$ kanamycin, overnight at 37 °C with mixing at 190 rpm. The production cultures were inoculated with overnight cultures and incubated at 37 °C with mixing at 190 rpm. After $A_{600nm}$ of the culture reached 0.5, protein expression was induced with 0.5 mM IPTG (Goldbio, I2481C25), and incubation was continued at 20 °C for 16 h. The cells were harvested by centrifugation at 7000 × g at 4 °C for 10 min, and the pellets were stored at − 80 °C until used further for protein purification. Cre recombinase was kept on ice or refrigerated at 4–8 °C throughout the purification procedure. The cell pellet from a 1.5-liter culture was thawed in a room-temperature water bath, resuspended in Cre-lysis buffer (50 mM Na phosphate (Fisher Scientific) pH 7.4, 1 M NaCl (MilliporeSigma), 1 complete EDTA-free protease inhibitor cocktail tablet (MilliporeSigma, St. Louis, MO, USA, COEDTAF-RO) per 50 ml buffer), and lysed by sonication (125-W pulses (5 sec on / 5 sec off) for 10 min total) or by French press (3 passes at up to 15,000 psi). The lysate was centrifuged at 48,500 × g for 15 min, and the supernatant was incubated with 1 ml of a suspension of Ni-Sepharose High Performance beads (Cytiva, Uppsala, Sweden, 17526801) in a rotating mixer for 1 h. The resin was centrifuged at 500 × g for 5 min, washed with 40 ml of the Cre-wash buffer (25 mM Na phosphate pH 7.4, 500 mM NaCl), centrifuged again, and then packed in a Tricorn 5/50 (Cytiva) column connected to a Bio-

Rad DuoFlow system (Bio-Rad, Hercules, CA, USA) with Bio-Rad Bio-Logic DuoFlow 5.30 software, which was perfused at 0.5 ml min⁻¹. The resin was washed with Cre-wash buffer (20 ml, or until a stable absorbance baseline (280 nm) was observed); then, the proteins were eluted with 30 ml of a continuous gradient of the Cre-elution buffer (25 mM Na phosphate, pH 7.4, 500 mM NaCl, and 0–500 mM imidazole (Fisher Scientific)). The fractions containing Cre recombinase that were identified by SDS-PAGE were concentrated and subjected to size-exclusion chromatography on a Superdex 200 Increase 10/300 GL column (Cytiva, 28990944), or a HiLoad 16/600 Superdex 200 pg column (28989335), with P500G buffer (Na phosphate 20 mM pH 7.4, NaCl 500 mM, glycerol (Fisher Scientific) 20% (v/v)) as the mobile phase. Fractions containing pure Cre recombinase were concentrated, snap frozen in liquid nitrogen, and stored at −80 °C.

### Expression and purification of ABE

*E. coli* BL21star (DE3) bacteria (Thermo Fisher Scientific, C601003) were transformed with the ABE8e *Sp*Cas9 NG (referred to as ABE8e) expression plasmid described previously,[15] and grown overnight on Luria broth (LB)-agar plates containing 25 μg ml⁻¹ kanamycin; single clones were used to inoculate starter cultures in Terrific broth (TB, Thermo Fisher) containing 25 μg ml⁻¹ kanamycin, and grown overnight. One and a half liters of TB containing 25 μg ml⁻¹ kanamycin was inoculated with 10 ml of starter culture, and the culture was grown at 37 °C with mixing at 190 rpm, until it reached an $A_{600nm}$ of 1.5. Then, the culture was cooled in an ice-water slurry, and protein expression was induced with 0.8% (w/v) rhamnose (Goldbio, R-105-250). The proteins were expressed at 17 °C with mixing of the culture at 190 rpm for 16–24 h. The bacteria were harvested by centrifugation at 5000 × g, 10 min, 4 °C, and the pellets were stored at −80 °C.

All protein purification steps were conducted in a cold room (~4-8 °C), or on ice. The thawed bacterial pellet from a 1.5-liter culture was homogenized with a 40 ml Dounce homogenizer in lysis buffer (100 mM Bis-Tris Propane (Millipore Sigma), pH 8.0, 1 M NaCl, 20% glycerol (v/v), 5 mM TCEP (Hampton Research), 1 cOmplete Ultra EDTA-Free protease inhibitor tablet per 40 ml of the buffer), and lysed by sonication with a Qsonica Q125 sonicator (125 W) with a 1/8-inch microtip at 100% amplitude for a total of 20 min (intermittent pulses: 5 sec on; 5 sec off). The lysate was clarified at 4 °C by centrifugation at 48,500 × g for 10 min. ABE protein was captured on a 3 ml TALON metal-affinity resin (Takara Bio, San Jose, CA, USA, 635502). The resin was then washed with 100 ml of the lysis buffer without inhibitors, and with 10 ml of ABE wash buffer (100 mM Bis-Tris Propane, pH 8.0, 500 mM NaCl, 20% glycerol (v/v), 1 mM TCEP). The proteins were eluted with the wash buffer supplemented with 150 mM imidazole.

In a second step, ABE was purified by immunoaffinity chromatography using a 1D4 resin (CNBr-activated Sepharose 4B(Cytiva, 17043001), with immobilized 5–10 mg ml⁻¹ 1D4 antibody purified in house). The 1D4 resin (4 ml) was packed in a DWK Life Sciences Kimble™ Kontes™ FlexColumn™. The column was equilibrated with ABE wash buffer. ABE was loaded by gravity flow. The column was washed with at least 40 ml of the wash buffer at 0.5 ml min⁻¹, and then ABE was eluted with 1 mg ml⁻¹ 1D4 peptide in the wash buffer at approximately 1 ml h⁻¹. Fractions containing ABE were pooled, concentrated, and further purified to remove contaminating nucleic acids and aggregates by size exclusion chromatography on a Superdex 200 Increase 10/300 GL column, or a Superdex 200 16/60 Prep Grade column. The protein was eluted with ABE storage buffer (HEPES 10 mM (Goldbio), pH 7.0, 500 mM NaCl, 20% glycerol). The fractions containing pure ABE were concentrated using Amicon Ultra centrifugal filters with 30-kDa MWCO membranes (Merck, UFC903024), sterilized by passage through 0.22-μm filters, quantified using UV absorption at 280 nm, divided into aliquots, and snap frozen in a chilled metal block for storage at -80 °C. Protein purity was assessed using SDS-PAGE in hand-cast Tris-glycine-SDS discontinuous gels with 4% acrylamide in the

stacking gel (pH 6.8) and 10% acrylamide in the resolving gel (pH 8.8), with 2.7% cross-linker (acrylamide:bis-acrylamide ratio of 37.5:1, Bio-Rad 1610158). The samples were mixed with 4x-concentrated Laemmli sample buffer (Bio-Rad 1610747) and supplemented with 50 mM dithiothreitol (DTT, MilliporeSigma D9779), denatured at 70 °C for 10 min and centrifuged before applying the supernatant on to the gel. The protein concentration was quantified using UV absorption spectroscopy by measuring absorbance at 280 nm with a Nanodrop ND-1000 spectrophotometer (software version 3.8.1). Separation of contaminating nucleic acids was followed by monitoring the $A_{260nm}/A_{280nm}$ ratio; a ratio < 0.60 was used as an indication of protein free from nucleic acids. A typical $A_{260\ nm}/A_{280\ nm}$ value for purified ABE was 0.55. Extinction coefficients of the constructs were estimated using the ProtParam tool (https://web.expasy.org/protparam/).

### Delivery of Cre recombinase in vitro

The CS cells were plated on 48-well treated tissue-culture plates in complete medium with 10% FBS. For each sample, Cre recombinase was first complexed for 15 min at room temperature with CBB compound in 12.5 μl of OptiMEM medium (Thermo Fisher, 31985070), at final concentrations of 10 μM of Cre enzyme and 20–160 μM of CBB compound. The mixture was then diluted with OptiMEM medium (Thermo Fisher, 31985070) to a final volume of 250 μl and applied to cells. After incubation for 3 h at 37 °C, 5% CO₂, the medium was exchanged with DMEM medium (Thermo Fisher, 10566016) containing 10 % FBS, and the cells were incubated for an additional 24 h. For imaging, the medium was changed to FluoroBrite DMEM (Thermo Fisher, A1896701), and the cells were imaged using a Keyence BZ-X810 microscope with optical filters for GFP and Texas Red. Keyence BZ-X800 Viewer 1.1.1.3 was used to collect the images, and Keyence BZ-X800 Analyzer 1.1.1.2 and XnView 2.51.4 were used to analyze the images. Adobe Illustrator 28.4.1 was used to prepare the figures.

### Flow cytometry

Cells were washed with PBS (Thermo Fisher, 10010023), detached with 0.05% trypsin (Thermo Fisher Scientific, 25300054), transferred to a 96-well round-bottom plate, and centrifuged at 180 × g for 5 min at room temperature. Centrifuged cells were washed with a FACS buffer (PBS with 2% FBS, 100 U ml⁻¹ penicillin-streptomycin (Fisher Scientific, 15-140-122)), centrifuged again and resuspended in the FACS buffer with 1 μg ml⁻¹ DAPI (Thermo Fisher, 62248). The cells were analyzed using a Novocyte Quanteon (Agilent) flow cytometer with Pacific Blue (445/45 nm), FITC (530/30 nm), and PE (586/20 nm) optical filters, with NovoExpress 1.6.2 software. Cells were gated on forward and side scatter, and single cells, and analyzed for viability via DAPI exclusion. The gating strategies are outlined in Supplementary Fig. 15.

### ABE activity assay

Synthetic 60-bp-long DNA oligonucleotides (MilliporeSigma, see Supplementary Data 1) labelled with fluorescein on the strand undergoing deamination were annealed at a 1:1 ratio in 10 mM Tris pH 8.0, 50 mM NaCl and 1 mM EDTA (MilliporeSigma) by incubation at 95 °C for 5 min and subsequent slow cooling to 20 °C. End-modified sgRNA (Supplementary Data 1) was ordered from IDT Technologies, with 2′-O-methyl groups on the first three and last three nucleotides, and the first and last three phosphodiester bonds were replaced with phosphorothioate bonds. The guide RNAs were dissolved in nuclease-free water at 37 °C for 15 min at 500 rpm in a thermomixer and folded by incubation at ~75 °C for 5 min, followed by slow cooling. Prepared nucleic acids were quantified using UV absorption spectroscopy by measuring absorbance at 260 nm. Accordingly, an $A_{260nm}$ value of 1.0 corresponded to a DNA concentration of 50 μg ml⁻¹ or an RNA concentration of 40 μg ml⁻¹. DNA was stored at −20 °C and RNA at −80 °C. For the ABE enzymatic assay, ABE ribonucleoprotein was assembled by incubation with 1.5-fold molar excess of sgRNA in a reaction buffer

(20 mM Bis-Tris propane, pH 7.5, 100 mM KCl (Fisher Scientific), 2.5 mM $MgSO_4$ (Fisher Scientific), 2 mM DTT, 5% (v/v) glycerol) for 15 min at room temperature. Additional 10% (w/v) sucrose (MilliporeSigma) was used when RNPs were assembled above 4 μM. Then, ABE was diluted to 1 μM with the reaction buffer, preheated at 37 °C and added with 15 ng of DNA substrate (0.02 μM final). The deamination was conducted for 10 - 60 min for ABE8e in the absence/presence of 4- to 16-fold of CBB22, 11, 14. The 20 μl reactions were quenched by addition of 30 μl of water preheated to 95 °C and incubated for 2 min at 95 °C. After cooling, the mixtures were treated with 1 μl of RNase A (20 mg ml$^{-1}$, MilliporeSigma) and 1 μl of proteinase K (20 mg ml$^{-1}$, Viagen Biotech 501-PK) for 15 min at room temperature; then DNA products were purified using an Oligo Clean and Concentrator kit (Zymo Research, D4061). Purified DNA was nicked with 5 units of Endonuclease V (EndoV, NEB) for 2-3 h at 37 °C, after which the reaction was quenched by addition of TriTrack DNA-loading dye (1 × final) (Thermo Fisher, R1161) and incubation at 95 °C for 2 min. The cleavage products were analyzed by denaturing polyacrylamide gel electrophoresis with urea (Urea-PAGE) in Bio-Rad MiniProtean continuous hand cast 15% acrylamide gels in Tris-borate-EDTA (TBE; Bio-Rad, 1610770) with 7 M urea and 5% crosslinker (acrylamide:bis-acrylamide ratio of 19:1; Bio-Rad,1610144). The voltage was controlled to maintain at least 42 °C in the electrophoresis chamber. Imaging was done using the ChemiDoc MP system (Bio-Rad) and analyzed with ImageLab 6.1.0 software (Bio-Rad).

### Delivery of ABE in vitro

ABE-RNP was assembled by incubation of up to 20 μM ABE8e with 1.1-fold molar excess of chemically modified, heat-refolded, synthetic sgRNA (IDT and Genscript, see Supplementary Data 1) in OptiMEM medium with 10% (w/v) sucrose for at least 15 min at room temperature. The sgRNA had stabilizing modifications as described[49]. Next, ABE-RNP was diluted in TAS buffer (20.1 mM Tris, 4.3 mM sodium acetate (Fisher Scientific), pH 7.3, sucrose 10% (v/v)) and added to CBB-lipidoids at 1:2 to 1:16 ratios of enzyme:compound, 5%/7.25% (v/v) final DMSO (MilliporeSigma, D2650-100ML), 10 μM final RNP, and incubated for a further 15 min (Figs. 4 or 5 h) (Fig. 5f, i) at room temperature. The mixtures were applied on the cells in a 48-well plate that had been washed with OptiMEM, at a concentration of 0.1 μM ABE in OptiMEM, without added sucrose. After 3 h, the medium was exchanged with 250 μl of complete medium.

To assemble complexes of ABE-RNP with liposomes (Lipo1-Lipo8), 10 μM ABE-RNP was diluted with TAS buffer and added to concentrated liposomes to achieve 1 μM ABE-RNP with 8-fold excess of CBB-11 (Lipo1/3/5/7) or equivalent amount of lipids (Lipo2/4/6/8) and incubated for 15 min at room temperature. Next, ABE-RNP complexed with liposomes was diluted ten-fold with preheated complete cell culture medium, serially diluted and applied onto the cells in 96-well plate, 100 μl per well.

To assemble complexes of ABE-RNP with Lipofectamine 3000 (LF, Fisher Scientific L3000001), 0.5 μl of LF per well of a 48-well plate, or 0.2 μl of LF per well of a 96-well plate, were diluted to 12.5 μl and 5 μl per well, respectively, with OptiMEM containing 10% (w/v) sucrose; the diluted ABE-RNPs were then added to the diluted lipids and incubated for 15 min at room temperature. A lipoplex of ABE RNP with Lipofectamine in a volume of 25 μl was added to the cells with 225 μl fresh complete medium in a 48-well plate, or 10 μl was added to the cells with 90 μl medium in a 96-well plate.

In all assays, cell culture medium was replaced 24 hours, and the analyses were done 48 h after applying ABE. The cells were imaged using a Keyence BZ-X810 microscope with optical filters for GFP and Texas Red after a medium change to Fluorobrite DMEM with 1% FBS, equilibrated for at least 15 min in $CO_2$ incubator. Activities of ABE in the *rd12* reporter cell line were estimated by comparing ratios of intensities of GFP and mCherry. Activities of ABE in the TIGER HEK reporter cell line were estimated by calculating the intensity of tdTomato in areas covered with cells using ImageJ/Fiji 1.54i.

### Subretinal injections and delivery of Cre recombinase in vivo

Cre proteins complexed with CBB lipidoids (as above) were diluted in OptiMEM to 20 μM. First, the mice were anesthetized by an intraperitoneal injection of a cocktail consisting of 20 mg ml$^{-1}$ ketamine (MWI, 120495) and 1.60 mg ml$^{-1}$ xylazine (MWI, 008679) in PBS at a dose of 100 mg kg$^{-1}$ of ketamine and 8 mg kg$^{-1}$ of xylazine, and their pupils were dilated by topical administration of 1% tropicamide ophthalmic solution (Akorn, 17478-102-12) and 10% phenylephrine ophthalmic solution (MWI Animal Health, 054243). The corneas were hydrated with GenTeal Severe Lubricant Eye Gel (0.3% Hypromellose, Alcon). Subretinal injections were performed using an ophthalmic surgical microscope (Zeiss). An incision was made through the cornea adjacent to the limbus at the nasal side, using a 27-gauge needle. A 34-gauge blunt-end needle (World Precision Instruments, NF34BL-2), connected to an RPE-KIT (World Precision Instruments, RPE-KIT) with SilFlex tubing (World Precision Instruments, SILFLEX-2), was inserted through the corneal incision while avoiding the lens, and advanced into the subretinal space. Each mouse received a 1-μl injection per eye, and volume and rate were controlled with a UMP3 UltraMicroPump (World Precision Instruments, UMP3-4). After surgery, the mice were placed on a heating pad, and anesthesia was reversed with intraperitoneal atipamezole (2.5 mg kg$^{-1}$, MWI Animal Health #032800). Triple antibiotic ophthalmic ointment (neomycin, polymyxin, and bacitracin, AmerisourceBergen 044525) was administered to the cornea to promote recovery.

### Two-photon imaging of RPE and photoreceptor transfection in mouse eyes

After euthanasia of the mice, their intact enucleated eyes were submerged in room temperature PBS. Pulsing IR light from a Ti:sapphire laser (Coherent, Vision S, Santa Clara, CA; tunable between 690 and 1050 nm) was set to 950 nm and attenuated in a controlled, variable manner with an electro-optic modulator (EOM). To image and spectrally separate GFP and tdTomato signals, two de-scanned spectral detectors were used with their detection bandwidths set to 490–545 nm for GFP, and 590–680 nm for tdTomato. A 1.0 NA 20x Leica objective was used for the imaging[50,51]. Maximum projection images were generated from the collected Z-stack function of using LAX 4.7.0 and the Image J analyze particle function was used to quantify transfected cells.

### Delivery of ABE in vivo

ABE was complexed with 1.1-fold excess of sgRNA in water with 10% (w/v) final sucrose at a protein concentration of 32 μM for 15 min at room temperature, diluted into TAS buffer, and kept at room temperature until injection. Right before injection, ABE RNP in TAS was added to concentrated CBB-lipidoid in DMSO or CBB-liposome in TAS, mixed by rapid pipetting, incubated with the reagents for at least 10 min at room temperature, and injected subretinally into *rd12* or WT mice. The remaining reagents were then applied onto *rd12* reporter cells in DMEM with 10% FBS in a 96-well plate (Fig. 5f, i).

### RPE dissociation, genomic DNA and RNA, and lysate preparation

Mouse eyes were dissected under a light microscope to separate the posterior eyecup (containing RPE, choroid, and sclera) from the retina and anterior segment. Each posterior eyecup was immediately immersed in RLT Plus (Qiagen). RPE, choroid, and scleral cells were detached from the posterior eyecup by gentle pipetting, followed by removal of the remaining posterior eyecup. Cells were then processed for genomic DNA and RNA using the AllPrep DNA/RNA Micro kit according to manufacturer instructions (Qiagen 80284).

## Electroretinography (ERG)

Prior to ERG recording, mice were dark adapted overnight. Under a safety light, mice were anesthetized by isoflurane (MWI, 502017) inhalation, and their pupils were dilated with topical administration of 1% tropicamide ophthalmic solution (Akorn; 17478-102-12) and 10% phenylephrine ophthalmic solution (MWI Animal Health #054243), followed by hypromellose (Akorn; 9050-1) for hydration. The mouse was placed on a heated 37 °C Diagnosys Celeris rodent-ERG device (Diagnosys LCC, Lowell, MA, USA). Ocular stimulator electrodes were placed on the corneas, the reference electrode was positioned sub-dermally between the ears, and a ground electrode was placed in the rear leg. The eyes were stimulated with a green-light stimulus of -0.3 log cd s m$^{-2}$ (peak emission 544 nm, bandwidth ~160 nm). The responses for 10 stimuli with an inter-stimulus interval of 10 sec were averaged, and the a- and b-wave amplitudes were acquired from the averaged ERG waveform. Data were analyzed with Espion 6.61.12 software (Diagnosys LLC).

## Retinoid analysis

Mice were dark adapted for 24 h before eye enucleation. Eyes were homogenized in 1 ml of a 10 mM Na phosphate buffer (pH 8.0), containing 50% methanol (v/v) (Sigma–Aldrich; 34860-1L-R) and 100 mM hydroxylamine (Sigma–Aldrich; 159417-100 G). After 15 min incubation at room temperature, 2 ml of 3 M NaCl was added. The resulting sample was extracted twice with 3 ml of ethyl acetate (Fisher Scientific; E195-4). Then, the combined organic phase was dried *in vacuo* and reconstituted in 300 μl hexanes. Extracted retinoids (100 μl) were separated on a normal-phase HPLC column (Zorbax Sil; 5 μm; 4.6 mm × 250 mm; Agilent Technologies) connected to an Agilent Infinity 1260 HPLC system equipped with a diode-array detector. Separation was achieved with a mobile phase of 0.6% ethyl acetate in hexane (Fisher Scientific; H302-4) at a flow rate of 1.4 ml min$^{-1}$ for 17 min, followed by a step increase to 10% ethyl acetate in hexane for an additional 25 min. Retinoids were detected by monitoring absorbance at 325 nm and 360 nm, using Agilent ChemStation 11 software.

## Liposome formulation

A mixture of CBB11 (0.25 μmol), ionizable lipid (2.5 μmol), DSPC (0.5 μmol, Avanti Polar Lipids), cholesterol (1.675 μmol, MilliporeSigma) and DMG-PEG 2000 (0.075 μmol, BroadPharm, San Diego, CA, USA) in ethanol/chloroform was dried with a gentle flow of nitrogen. The resulting blue film was resuspended in TAS buffer (500 μl), and soni-cated for 30 min to form liposomes Lipo1, Lipo3, and Lipo5 that con-tained ionizable lipids SM-102 (BroadPharm), ALC-0315 (Avanti Polar Lipids), and MC3 (BroadPharm), respectively. Liposomes without CBB11 (Lipo2, Lipo4, Lipo6) contained additional cholesterol in place of CBB11, 1.925 μmol in total. Liposomes without ionizable lipid (Lipo7 with CBB11 and Lipo8 without CBB11) contained additional DSPC (1 μmol total in both Lipo7 and Lipo8) and cholesterol (3.675 μmol total in Lipo7, 3.925 μmol total in Lipo8) instead.

## Size-exclusion chromatography

Free CBB11, Lipo1 and Lipo2 were diluted in TAS buffer to an equivalent of 80 μM CBB11 in 100 μl. Then, the samples were applied onto PD-10 desalting columns (Cytiva, 17-0851-01) equilibrated with TAS buffer, and resolved by adding 250-μl aliquots of TAS buffer with manual fractionation. The fractions were transferred to a clear-bottom 96-well plate and scanned for Coomassie fluorescence using a Bio-Rad Che-miDoc MP imager with Bio-Rad ImageLab Touch 2.4 software. The columns were also scanned for Coomassie fluorescence and photographed.

## Particle-size distribution analysis

Particle-size distribution was measured using a Malvern Zetasizer Advance Nano instrument (Malvern Panalytical, Malvern, UK) with Malvern ZS Xplorer 3.30 software. Freshly dialyzed liposomes (20 μl) were diluted to 200 μl with TAS buffer and subjected to measurement of particle-size distribution in triplicate. The particle size distributions were processed by the accompanying software to calculate average particle diameter and polydispersity index (PdI).

## Cryogenic transmission electron microscopy (Cryo-TEM)

Cryo-TEM specimen was prepared by plunge-freezing in liquid ethane on a Vitrobot Mark IV (Thermo Fisher Scientific). Five microliters of the sample were applied on a lacey carbon coated grid after glow dis-charging in a Leica Sputter Coater ACE200 (Leica Microsystems). Excess liquid was removed with blotting of 3.0 seconds under chamber temperature of 8 °C and 100% humidity. The grids were then clipped and transferred into a Glacios TEM (Thermo Fisher Scientific) for examination. The microscope was operated at 200 kV, and images were recorded on a CETA camera (Thermo Fisher Scientific) under low-dose conditions to minimize the radiation damage on the specimen. EPU 2.11.1.11REL (Thermo Fisher Scientific) was used to collect the data.

## Next-generation sequencing

The genomic context of RPE65 *rd12* ABE editing is shown in Supple-mentary Fig. 16. Complementary DNA (cDNA) was synthesized from RNA with the High-Capacity RNA-to-cDNA kit (Thermo Fisher 4387406), according to manufacturer's instructions. 0.5 – 1 μl of the isolated genomic DNA or cDNA was used as input for the first of two PCR reactions (PCR1). Genomic loci were amplified in PCR1, using Phusion Plus polymerase (Thermo Fisher Scientific F631S). PCR1 pri-mers are listed in Supplementary Data 1. PCR1 was performed as fol-lows: 98 °C for 30 sec; 30 cycles at 98 °C for 10 sec, 67 °C for 20 sec, and 72 °C for 30 sec; 72 °C for 5 min. PCR1 products were confirmed on a 2% agarose gel. One microliter of PCR1 was used as input for PCR2 to install Illumina barcodes. PCR2 was conducted for nine cycles of amplification using a Phusion HS II kit (Life Technologies). Following PCR2, samples were pooled, and gel-purified in a 1% agarose gel using a Qiaquick Gel Extraction Kit (Qiagen). Library concentration was quantified using the Qubit High-Sensitivity Assay Kit (Thermo Fisher Scientific). Samples were sequenced on an Illumina MiSeq instrument (paired-end read, read 1: 200–280 cycles, read 2: 0 cycles), using an Illumina MiSeq 300 v2 Kit (Illumina) and analyzed using Illumina MiSeq Reporter 2.6.

## High-throughput sequencing data analysis

Sequencing reads were demultiplexed using the MiSeq Reporter 2.6 software (Illumina) and analyzed using CRISPResso2 2.2.7 as pre-viously described[52,53]. Batch analysis mode (one batch for each unique amplicon and sgRNA combination analyzed) was used in all cases. The prompt is presented in the Supplementary Information. Reads were filtered according to minimum average quality score (Q > 60) prior to analysis. The following quantification window parameters were used: -w 20 -wc -3. Base editing efficiencies are reported as the percentage of sequencing reads containing a given base conversion at a specific position. Prism 10 (GraphPad) was used to generate the plots.

## Statistics & reproducibility

No sample sizes were pre-determined. Animals were treated by litter cohorts (5-9 animals per litter, 1-2 eyes per animal). Sample sizes were 2-3 animals per group. In vitro cell studies included 2-4 replicates per experiment. Mice with failed subretinal injections were excluded on the basis of no recordable ERG a- or b-wave by a blinded observer. This exclusion criteria were pre-established. In vitro experiments were replicated 2 to 4 times independently. All animal experiments were replicated at least four times independently. In all studies, samples and animals were assigned randomly to treatment groups. Deep sequen-cing and ERG analysis were conducted blind. Other experiments were not conducted blindly.

No sex-based stratification or sex-based analysis was performed in this study. GraphPad Prism 10 was used to analyze the data. The data were tested for normality, and parametric one-way ANOVA was applied for datasets that passed the normality tests. The data that did not pass the normality test were analyzed with non-parametric ANOVA.

## Ethics statement

All animal procedures were approved by the Institutional Animal Care and Use Committee of the University of California, Irvine (protocol AUP-24-073), and were conducted in accordance with the NIH guidelines for the care and use of laboratory animals, and with the Association for Research in Vision and Ophthalmology Statement for the Use of Animals in Ophthalmic and Visual Research.

## Reporting summary

Further information on research design is available in the Nature Portfolio Reporting Summary linked to this article.

## Data availability

The high-throughput sequencing data generated in this study were deposited in the National Center for Biotechnology Information Sequence Read Archive database under accession codes: PRJNA1240374 and PRJNA1358750. The data underlying the manuscript were deposited in Dryad: doi.org/10.5061/dryad.d51c5b0hd[54] [https://datadryad.org/dataset/doi:10.5061/dryad.d51c5b0hd]. Source data are provided in this paper.

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

## Acknowledgements

We thank Marco Bassetto, Elliot H. Choi, Huajun B. Yan, and Alexander L. Yan for technical assistance. We acknowledge Dorota Skowronska-Krawczyk for access to a qPCR thermocycler, fluorescent microscope, and gel imager. We thank Suvrajit Sengupta at the UCI Department of Chemistry for technical support with NMR. We thank Jennifer Atwood and Michael Hou for technical assistance with flow cytometry. We thank Yekaterina Kadyshevskaya (University of Southern California) for the preparation of Fig. 1. We thank our colleagues at the UCI Center for Translational Vision Research and the Gavin Herbert Eye Institute for their comments regarding this manuscript. This work was supported in part by grants from the National Institutes of Health, including R01EY009339 (K.P.), 1R01EY034501 (K.P.), NSF-CHE-1904530 (G.P.T.), NIAID 75N93022C00054 (P.L.F.), DTRA grant N66001-21-C-4013 (P.L.F.), UG3AI150551 (D.R.L.), U01AI142756 (D.R.L.), R35GM118062 (D.R.L.), RM1HG009490 (D.R.L.), T32GM008620 (S.W.D.), F30EY033642 (S.W.D.); Foundation Fighting Blindness (award number TA-GT-0423-0847-UCI-TRAP) (K.P.); the Howard Hughes Medical Institute (HHMI) (D.R.L.); and Knights Templar Eye Foundation Career-Starter Research Grant (R.H.). R.H. is a Beckman Scholar in Retinal Research. The authors acknowledge support to the Department of Ophthalmology, Gavin Herbert Eye Institute at the University of California, Irvine, from an unrestricted Research to Prevent Blindness award, from NIH core grant P30EY034070, and from a University of California, Irvine School of Medicine Dean's Office grant. The authors acknowledge support for the Chao Family Comprehensive Cancer Center's Institute for Immunology Flow Cytometry Facility shared resource by the National Cancer Institute of the National Institutes of Health under award number P30CA062203. This article is subject to the Open Access to Publications policy of the Howard Hughes Medical Institute (HHMI). HHMI-supported authors have previously granted a nonexclusive CC BY 4.0 license to the public and a sublicensable license to HHMI in their research articles. Pursuant to those licenses, the author-accepted version of this manuscript can be made freely available under a CC BY 4.0 license immediately upon publication.

## Author contributions

Conceptualization was performed by J.Z., G.P.T, P.L.F., and K.P. Experimental investigations were performed by J.Z., S.W.D., J.H.F., R.S., R.H., G. P., G.P.T., C.R.M., Y.H., E.R., Z.D., X.M., M.H.S., L. X., M.H., P.Z.C., and B.L. Data analysis was performed by J.Z., R.H., S.W.D., J.H.F., R.S., C.R.M., E.R., G.P., and Y.H. Figures were prepared by J.Z., R.H., Y.H., G.P., and G.P.T. Manuscript was written by J.Z., R.H., Y.H., G.P.T., and K.P. Project was supervised by D.R.L., G.P.T., and K.P. Funds were acquired by R.H., S.W.D., D.R.L., G.P.T, P.L.F., and K.P. All authors contributed to the research and writing, and approved the manuscript.

## Competing interests

K.P. is a consultant for Polgenix Inc. and AbbVie Inc., and serves on the Scientific Advisory Board of Hyperion Eye Ltd. K.P. and G.P.T. are equity holders in Eyesomer Therapeutics. D.R.L. is a consultant and/or equity owner for Prime Medicine, Beam Therapeutics, Pairwise Plants, and nChroma Bio, companies that use or deliver genome-editing or epigenome-engineering agents. All other authors have declared that no competing interests exist.
