## [Transparent Peer Review file · Nature Communications]

A Combinatorial Synthetic Strategy for Developing Genome-Editing Protein-Delivery Agents Targeting Mouse Retina

Corresponding Author: Dr Krzysztof Palczewski

This manuscript has been previously reviewed at another journal. This document only contains information relating to versions considered at Nature Communications, with the exception of the previous reviewer #1 who provided permission for their report to be published.

Version 1:

Reviewer comments:

Reviewer #2

(Remarks to the Author)

This manuscript reports the development of a new lipidoid using Coomassie Brilliant Blue (CBB) as a building block and explores its application for protein delivery, including Cre recombinase and adenine base editors (ABEs). The authors further demonstrate *in vivo* gene editing in mouse eyes, suggesting a potential therapeutic approach for LCA-related diseases. Functional assessments, including ERG responses and 11-*cis*-retinal levels, indicate improvement in mice treated with the Lipo1 formulation (SM102 doped with 10% CBB11, along with other helper lipids) compared to those treated with the Lipo2 formulation (SM102 without CBB11). On- and off-target editing events were also analyzed using NGS.

Overall, the study is well executed and the data are convincing. However, my main concern lies in whether the novelty and significance of the work meet the publication standards of Nature Communications.

The rationale for developing a CBB-based lipidoid is to leverage its protein-binding ability to facilitate complexation and delivery. However, the data indicate that CBB alone is unstable, forms aggregates with proteins, and cannot be used independently for protein delivery. Instead, a conventional LNP formulation—comprising SM102, ALC-0315, or MC3 as the ionizable lipid, together with co-lipids—is still required for efficient protein (including ABE) delivery. It is well established that such LNP systems can deliver RNPs, and while the addition of CBB may enhance delivery efficiency (as shown in Fig. 6a–f), it also significantly increases particle size from ~100 nm to several hundred nanometers (Fig. 5b), suggesting reduced LNP/RNP stability. This instability could pose challenges for future clinical translation.

If CBB11 merely serves as an additional component to fine-tune LNP delivery efficiency, similar strategies have already been demonstrated, such as in the SORT technology developed by Daniel Siegwart's group. Therefore, unless the authors can demonstrate that CBB11 forms a stable formulation with RNPs capable of achieving efficient *in vivo* delivery without relying on other ionizable lipids (e.g., SM102 or related molecules), the overall impact and novelty of this work appear limited.

Reviewer #4

(Remarks to the Author)

Please see the attached file for comments

Reviewer #5

(Remarks to the Author)

After evaluating the concerns raised by reviewer #3, the author's response was sufficient to address all the relevant points.

The article will be ready for publication after the minor corrections indicated below:

Figure 9S panels d and e, along with f and g, require statistical descriptions and annotations to be presented. The absence of this information significantly weakens the figure's impact and prevents evaluation of the authors' claims regarding the safety and efficacy of their new Lipo1 batch. Please add the statistical test description, or at least the p-value, to the legend, and significance annotations to the graphs.

Figure 9S panel c: how was mild inflammation observed in the OCT images? Please, describe.

Minor errors in typing:

Lines 154, 232, 276: change "florescence" to "fluorescence".

Line 277: change "efeective" to "effective".

Version 2:

Reviewer comments:

Reviewer #2

(Remarks to the Author)

The authors have done a good job in addressing the reviewers' concerns. the manuscript is significantly improved. i recommend its acceptance for publication.

Reviewer #4

(Remarks to the Author)

Please see the attached file for my comments.

Point-by-Point Response to Reviewers:
[Our original responses are abbreviated.]

Reviewer #2

Reviewer Comments 2.1. Overall, the study is well executed and the data are convincing. However, my main concern lies in whether the novelty and significance of the work meet the publication standards of Nature Communications.

The rationale for developing a CBB-based lipidoid is to leverage its protein-binding ability to facilitate complexation and delivery. However, the data indicate that CBB alone is unstable, forms aggregates with proteins, and cannot be used independently for protein delivery. Instead, a conventional LNP formulation—comprising SM102, ALC-0315, or MC3 as the ionizable lipid, together with co-lipids—is still required for efficient protein (including ABE) delivery. It is well established that such LNP systems can deliver RNPs, and while the addition of CBB may enhance delivery efficiency (as shown in Fig. 6a–f), it also significantly increases particle size from ~100 nm to several hundred nanometers (Fig. 5b), suggesting reduced LNP/RNP stability. This instability could pose challenges for future clinical translation.

Response 2.1: We agree that CBB alone is not a suitable standalone vehicle for RNPs, as it forms larger colloids and can aggregate with protein. Accordingly, our *in vivo* strategy uses CBB-doped liposomes (Lipo1-6), which present the CBB moiety at the bilayer surface to engage proteins while ionizable lipids engage the guide RNA, an intended mechanism supported both by cryo-TEM (RNP on the surface) and by a marked zeta-potential shift upon RNP addition.

Size/stability. The reviewer's size interpretation stems from CBB11 alone (400–500 nm). In contrast, CBB11-doped liposomes are ~140-160 nm and highly monodisperse, and we observed no RNP-induced precipitation with Lipo1-6. Thus, adding CBB does not enlarge the liposomes to “several hundred nanometers” nor indicate reduced stability in our system.

Functional benefit and novelty. CBB-liposomes outperform control liposomes *in vitro* and *in vivo*. Beyond efficacy, this platform is not a conventional LNP; the promiscuous protein-binding domain (CBB) is the key design element, enabling surface recruitment of diverse proteins/RNPs rather than relying solely on electrostatics or encapsulation.

Translational considerations. Our current liposome sizes already are in the 140–160 nm window typical for clinically advanced nanoparticles, and size can be further tuned by standard formulation parameters, if needed. Importantly, this liposomal (non-microfluidic) assembly avoids low pH and high

ethanol used in some RNP-LNP processes that can denature editors, and it improves precise editing relative to our prior SM102-LNP (~0.30% precise gDNA with LNP vs. ~2.7% with CBB-liposomes; ~9-fold increase). Taken together, CBB-liposomes address the instability seen with CBB alone, maintain clinically appropriate size and monodispersity, and deliver meaningful functional gains, thereby supporting both novelty and translational potential.

Revision 2.1: To address this comment, we added the following sentences to Paragraph 5 of the discussion:

“An additional practical determinant is particle size. Dynamic-light-scattering measurements indicated that CBB11/Cre and CBB14/Cre form ~1.4–1.5 μm assemblies, with a significant fraction of CBB14-Cre forming smaller, 160-nm particles, whereas CBB11 alone forms ~400–500 nm colloids; importantly, CBB11-doped liposomes (Lipo1–6) are ~140–160 nm and highly monodisperse. Micrometer-scale complexes are unlikely to diffuse broadly within the neural retina or traverse ocular barriers from systemic delivery, whereas systemic delivery of the smaller, narrowly distributed liposomal formulations would be effective. Notably, despite their large hydrodynamic diameters, CBB11 and CBB14 still delivered Cre via subretinal injection.”

Reviewer Comments 2.2. If CBB11 merely serves as an additional component to fine-tune LNP delivery efficiency, similar strategies have already been demonstrated, such as in the SORT technology developed by Daniel Siegwart’s group. Therefore, unless the authors can demonstrate that CBB11 forms a stable formulation with RNPs capable of achieving efficient in vivo delivery without relying on other ionizable lipids (e.g., SM102 or related molecules), the overall impact and novelty of this work appear limited.

Response 2.2: Our approach is *not* equivalent to SORT and its impact does not depend on eliminating ionizable lipids. CBB does more than “fine-tune” delivery; it introduces a general protein-binding capture modality that synergizes with ionizable lipids to efficiently deliver RNPs. To clarify the reliance of our CBB liposomes on ionizable lipids, we formulated DSPC-cholesterol-PEG liposomes that do not contain ionizable lipids, with and without CBB11 (Lipo7 and Lipo8, respectively), and compared their RNP delivery efficiency side-by-side with Lipo1 and Lipo2 that contain SM102. Liposomes that did not contain CBB11 or SM102 (Lipo8) did not aid ABE-RNP delivery into the *rd12* reporter cells, as the eGFP fluorescence was similar as for ABE-RNP without additional reagents. Liposomes free from SM102 but supplemented with CBB11 (Lipo7) delivered ABE-RNP with modest efficiency. Lipo2 (with SM102, without CBB11) also delivered the RNP, but we again observed the highest ABE-RNP delivery efficiency with Lipo1 that contained both CBB11 and SM102. Our experimental results support our conclusion that CBB and ionizable lipids work together to achieve optimal delivery efficiency of ABE-RNP, highlighting

the novelty and significance of this platform. This is further corroborated by our strong *in vivo* results, where we achieved healthier ERG responses and more precise editing compared to eVLP and lentivirus. Importantly, CBB and ionizable lipids work together to achieve efficient RNP delivery – CBB by binding ABE protein, and ionizable lipids by binding guide RNA. We added the results obtained for ionizable-lipid-free liposomes to the new Supplementary Figure S9 and integrated the results within the manuscript:

“The RNP delivery enhancement mediated by CBB11 did not depend on the ionizable lipid, as we observed similar enhancement with liposomes made with ALC-0315 and MC3 (Fig. 5g, h, & Fig. S6e), and with liposomes formulated without ionizable lipids (Fig. S9).”

Fig S9. Synergistic engagement of ABE-RNP by CBB liposomes. (a) Fluorescence microscopic quantification of delivery of ABE8e-RNP into *rd12* reporter cells mediated by liposomes with CBB11 (Lipo1, Lipo7) and SM102 (Lipo1, Lipo2). Lipofectamine 3000 (LF) was used as a positive control; buffer (TAS) as a negative control. Four biological replicates, mean \pm S.D. (b) Representative fluorescence microscopic images of *rd12* reporter cells treated with ABE8e-RNP complexed with liposomes. Scale bar, 200 μ m.

How our platform differs from SORT. SORT is an *mRNA/DNA* LNP strategy that adds a fifth lipid to tune *biodistribution via* surface charge/composition. It does not address protein capture/display. In contrast, our system presents a Coomassie-based, promiscuous protein-binding domain (CBB) on a lipid bilayer to recruit protein or RNP cargo at the vesicle surface under mild, neutral conditions. Thus, SORT modifies *where* nucleic-acid LNPs go; our platform changes *what* and *how* cargo is loaded and delivered.

Stability and size. Whereas CBB alone forms heterogeneous colloids, CBB-doped liposomes are ~140–160 nm, exhibit a low-PdI, and do not precipitate upon RNP loading. ζ -potential shifts and cryo-TEM are consistent with RNP association at the vesicle surface, and the complexes drive robust *in vivo* editing and ERG recovery (Fig. 5d–e; Fig. 6). Functionally stable performance *in vivo*, together with the absence of precipitation in formulation, addresses the reviewer’s stability concern.

Translational advantage. Unlike low-pH/ethanol LNP assembly, which can compromise editor integrity, our neutral, ethanol-free mixing preserves RNP activity and can be prepared from two stable

components immediately prior to use (analogous to standard transfection reagents).

These features of our CBB liposome delivery system are presented in the introduction and discussion.

Revised introduction:

“After initial success, we further refined our approach by incorporating the CBB lipidoid into liposomes to achieve efficient editing in an *in vivo* inherited retinal disease (IRD) mouse model. Our approach is distinct from LNPs, as the CBB liposomes are formed without protein and are used as a neutral, aqueous suspension to complex the cargo in neutral pH and without organic solvents. Our CBB lipidoids engage in specific CBB-protein interactions, and thus they can be used for delivery of all types of proteins, with a potential to perform genome editing using recombinases (Cre, Bxb1), genome-editing proteins (zinc finger nucleases, TALENs), and genome-editing RNPs (Cas9, ABE, PE, and others). We maximized the potential of CBB liposomes for RNP delivery by incorporating ionizable lipids to achieve synergistic complexing of the RNPs *via* CBB-protein and ionizable lipid-guide RNA interactions. Incorporating CBB into liposomes enhanced RNP-mediated genome editing, allowing to reach clinically significant recovery of visual function and editing in the genome. This outcome reinforces our guiding principle that specialized delivery agents, including our novel CBB lipidoids, are essential for achieving the RNP-mediated editing efficiency necessary for further clinical translation.”

Revised discussion:

“Several of the CBB-lipidoids substantially enhanced the delivery of both Cre recombinase and ABE proteins to mammalian cells *in vitro* and *in vivo*. Particularly striking was the high efficacy observed when these compounds were formulated into ionizable cationic liposomal particles to improve the colloidal stabilities of the CBB-protein complexes. We designed our CBB liposomes with a goal of maintaining enzymatic activity of ABE RNP by minimizing denaturation. This imposed a requirement to avoid acidic pH and ethanol, both of which are crucial for achieving efficient encapsulation of nucleic acids and RNPs within the LNPs. As a first step, we prepared lipid films containing all the lipid constituents: liposome scaffold of DSPC, cholesterol, and DMG-PEG 2000, ionizable lipid (SM102, ALC-0315 or MC3) to bind guide RNA, and CBB11 to bind protein. Next, instead of dissolving the film in ethanol and complexing the RNP in microfluidic system, we formed the liposomes in our mild dispersant, TAS buffer, and made liposome-RNP complexes by mixing in neutral, aqueous environment of TAS. This procedure, which resembles traditional transfection, led to adsorption of the RNP on the liposome surface. This enzyme-focused approach led to striking editing efficiencies and clinically relevant restoration of vision in mice. For instance, Lipo1 containing 5% CBB11 achieved up to 9.9% genomic and 28.8% transcript-level edits of the *Rpe65* mutation in *rd12* mice, a correction sufficient to restore vision to an estimated 76% of the wild-type ERG response (**Fig. 6**). These results surpass those reported for virus (AAV, LV) or virus-like particle (VLP) delivery systems.^{34, 35, 40-42”}

Reviewer #4

Reviewer 4 Comment: Firstly, I would like to compliment the authors on the amount of effort that they have shown in addressing the original comments of Reviewer 1. Especially the addition of multiple *in vivo* experiments is laudable. I will address all author comments per point below. However, there are a few comments that have not been fully addressed / resolved, as discussed below.

Response to Reviewer 4 Comment: We are very grateful for the reviewer's comments and time invested in our manuscript, undoubtedly leading to improvement. We have eliminated all comments where the reviewer approved our changes to the manuscript. We address the new comments below.

Original Reviewer Comment 1.1: All *in vitro* results were done at a single concentration of the CBB derivatives and no dose-response curves are given... The optimal dose therefore needs to be established for each reagent, separately.

Original Response 1.1: Our previous one-point analysis was based on optimized conditions, and we would argue still valid. Nevertheless, we performed dose-response analyses as recommended (see **Fig. S2**). In the revised manuscript, we now provide full *in vitro* dose-response curves (1–8 μM CBB) for the nine CBB derivatives (CBB2, 7, 11, 12, 13, 14, 18, 19, and Z5) with Cre recombinase, or (0.2–1.6 μM CBB) with ABE8e-RNP, now included.

Reviewer Comment 4.1: Originally, reviewer 1 was somewhat vague in their wording, and their request could either be interpreted as a request to provide a dose-response curve in terms of the absolute amount that would be used for transfection, or an optimization of CBB-to-cargo ratio. Here, the authors opted to focus on a ratio optimization for all CBBs, which I agree is the best (and most relevant) answer to the final statement by reviewer 1 “**The optimal dose therefore needs to be established for each reagent, separately.**” It is furthermore additionally appreciated that this was done for both Cre recombinase and ABE. As such, I feel that these experiments are suitable to address the comment of Reviewer 1.

However, in this case the authors have performed the experiment twice ($n=2$). Whereas the data show high reproducibility and a robustness in the observed patterns, I would like to point out the following:

Please be aware that the Nature Portfolio checklist contains the following statement/guideline: “*We strongly discourage deriving statistics from technical replicates or less than 3 biological replicates, unless there is a clear scientific justification for why providing this information is important.*”

Especially considering these experiments are *in vitro* assays, I would unfortunately request the

authors to perform this experiment once more, unless (as is in line with the Nature Portfolio guidelines) “there is a clear scientific justification for why providing this information is important”.

Responses and Revisions Reviewer 4.1. Thank you for the comment. We repeated the titrations for Cre and ABE with four replicates. In this new experiment, Cre delivery efficiency was comparable, whereas ABE was this time not efficiently delivered into the *rd12* reporter cells by CBB12. We replaced the Figure S2 with the new data presented as separate data points.

Fig S2. Concentration dependence of the *in vitro* delivery of gene editors mediated by representative CBB compounds. (a) Quantification by fluorescence microscopic imaging of GFP-to-RFP conversion in CS cells. Cells were treated with Cre recombinase (0.5 μ M) complexed with CBB compounds at increasing concentrations (1-8 μ M). The cells are green at baseline and red after editing. Data are presented as mean \pm S.D.; n = 2. (b) Quantification by fluorescence microscopic imaging of the intensity ratio of eGFP to mCherry in *rd12*-reporter cells treated with ABE-RNPs (100 nM) complexed with CBB compounds at increasing concentrations (0.2-1.6 μ M). Data are presented as mean \pm S.D.; n = 2.

Current version:

Fig S2. Concentration dependence of the *in vitro* delivery of gene editors mediated by representative CBB compounds. (a) Quantification by fluorescence microscopic imaging of GFP-to-RFP conversion in CS cells. Cells were treated with Cre recombinase (0.5 μ M) complexed with CBB compounds at increasing concentrations (1-8 μ M). The cells are green at baseline and red after editing. Mean \pm S.D., 4 biological replicates. (b) Quantification by fluorescence microscopic imaging of the area ratio of eGFP to mCherry in *rd12*-reporter cells treated with ABE-RNPs (100 nM) complexed with CBB compounds at increasing concentrations (0.2-1.6 μ M). Data are presented as mean \pm S.D.; n = 4 biological replicates.

Original Reviewer Comment 1.2: The *in vivo* data focus on base editing results, but not so much on toxicity and off-target editing... I would at least expect to see some evidence that toxicity to the retina is absent or limited at the tested concentrations.

Original Response 1.2: To evaluate the safety of the ABE-RNP delivery to the retina mediated by Lipo1, we also administrated ABE-RNP-Lipo1 complex to WT mice (Fig. S9) *via* subretinal injection. Since the oxidation of CBB11 might induce epoxides in the liposome, only freshly formulated Lipo1 was administered to the WT mice. Different batches of Lipo1 showed similar morphology and comparable ABE-RNP delivery efficiency to *rd12* reporter cell lines (Fig. S9a, b). The Optical Coherence Tomography (OCT) imaging indicated no substantial alterations to the retinal structure of WT mice administered the complex of RNP (20 μ M) and Lipo1; except mild inflammation, which could be ameliorated substantially by lowering the RNP concentration (Fig. S9c). The ERG response and the generation of the visual chromophore, 11-*cis* retinal, of WT mice treated with Lipo1 and RNP (10 μ M) were also comparable to those of untreated WT mice (Fig. S9d, e). In parallel experiments in which *rd12* mice were treated with the same batch of Lipo1 and RNP complex, the subretinal injections of 1 μ l of Lipo1 with 10 μ M RNP achieved an optimal ERG response and visual chromophore recovery (Fig. S9d, e), comparable to those of *rd12* mice treated with Lipo1-ABE-RNP (20 μ M) (Fig. 6). These results suggest that optimal vision-function rescue requires a balance between the biological tolerance of the cells and the editing efficiency of the gene editors.

Off-target editing—Targeted deep sequencing of the ten highest-ranked off-target sites (identified using Cas-OFFinder) shows a background-level A•T→G•C substitution rate comparable to that of untreated animals. Additional data on off-target analyses are now included.

Reviewer Comment 4.2: Once more, I feel that the authors have chosen suitable experiments to address the points raised by reviewer 1. Whereas n=2 replicates for Fig S9B, I feel that this is justified in this case due to the robustness of the data, as well as the ethical implications to minimize the amount of animals. Data is clear, methods are described well, conclusions in the main text are appropriate.

One request: I do not have access to the Supplementary Tables, so I'm not sure if all OT genomic sequences / analysis sequences are listed. As such, I would like to request to please make sure to include the analyzed Off-targeted sequence (+genomic location), as well as potential relevant primer/probe sequences used for OT analysis (either to the methods section, or the Supplementary files).

Responses and Revisions Reviewer 4.2: Thank you for the comment. We have included the protospacer and guide RNA sequences, as well as genetic loci, in Supplementary Table 1. The sequencing data were deposited in the Sequencing Read Archive.

Reviewer Comment 1.3: Some claims are made in the discussion that are not covered by the data presented (toxicity, RNP encapsulation, immunogenicity). Better to remove such unsubstantiated claims.

Original Response and Revision 1.3: Thank you for bringing up this important point. We have removed unsubstantiated statements regarding toxicity (see **Response 1.2**), immunogenicity, and full "encapsulation". All remaining claims are now directly supported by new data (toxicity, off-target analysis) or by published references cited in the Discussion.

Reviewer Comment 4.3: The changes are appreciated. Indeed, published references can be used to support claims on safety as well. In line with this, I would request adding a reference that shows decreased bystander editing upon short-term ABE delivery (ie. as RNP) to the statement "In general, the delivery of ABE-RNPs mediated by CBB-liposomes could be considered as a more transient mode of delivery of gene-editing tools, consistent with decreased bystander editing".

Also, please add a reference to the statement "small particles would better penetrate the retina and make the therapeutic agent available for cellular uptake".

Responses and Revisions Reviewer 4.3: Thank you for the comment. In our previous study, we found that ABE8e-RNP delivery *in vitro* via Lipofectamine 3000 or lipid nanoparticles led to more precise editing than plasmid delivery,¹ (Figure 4, new panel "o," shown below). In another study, we found that ABE delivered as RNP encapsulated into virus-like particles (eVLPs) led to lower bystander and off-target editing than viral delivery *via* AAV and LV *in vivo*.² We added the references to the manuscript. Regarding the tissue permeation, small protein GCAP1 whose diameter was 6.5 – 8.7 nm, permeated the retina more efficiently than GCAP1 complexed with liposomes (size range 150-170 nm).³ Our own data do not provide sufficient evidence to support the statement of enhanced penetration of the retina by smaller particles; therefore, we removed that statement.

[Figure Redacted]

"Fig. 4 | Lipid nanoparticle delivery of ABE RNPs. o, Next-generation sequencing analysis of ABE-editing outcomes in the cells with cDNA encoding *Rpe65 rd12*, treated with ABE delivered on a

plasmid with Lipofectamine 3000 (L3k) or as LNP. Three analytical replicates, mean \pm S.D.”¹

Reviewer Comment 1.4: From Cre recombinase to base editors. A logical step in between could have been CRISPR-Cas9 based gene correction. Did you try this as well?

Original Response 1.4: Thank you for the suggestion. Cas9 in the eye causes severe off-target effects and apoptosis.¹ Therefore, we deliberately applied our CBB strategy to an adenine base editor (ABE) rather than to a Cas9 nuclease, since ABEs offer superior therapeutic promise. This choice is further validated by recent clinical successes, most notably a personalized ABE-based gene therapy administered to an infant patient, which underscores the clinical relevance of base editing over nuclease-driven approaches.²

Reviewer Comment 4.4: This is a valid explanation, and as reviewer 1 did not request any additional data or experiments this answer suffices. However, readers may from outside the ocular delivery field may have similar questions. As such, I would request that this is briefly explained/addressed in the discussion as well.

Response and Revision 4.4: We added the following sentences to the Discussion, paragraph 2:

“While CRISPR-Cas9 nuclease RNPs could be viewed as a logical intermediate between Cre and base editors, we prioritized ABE for in vivo studies because in comparison with Cas9 nuclease, ABE has the advantage of catalyzing single-base transitions without creating double strand breaks, with equivalent potential for RNP delivery as Cas9. The molecular interactions that we engage to deliver ABE RNP are applicable to other RNPs, and thus we envision that our CBB liposomes are applicable to other CRISPR nucleases and precise editors as well.”

Reviewer Comment 1.5: CBB dyes have denaturing capacities. Do you see any reduction in activity of Cre recombinase or base editors in the presence (without encapsulation)?

Original Response 1.5: Coomassie Brilliant Blue (CBB) dyes indeed show higher affinity for partially unfolded proteins (e.g., Triton-X100 increases binding of Coomassie G250 in the Bradford assay³), yet decades of successful non-denaturing affinity-chromatography with structurally similar Cibacron Blue–based “Blue Sepharose”⁴ demonstrate that such interactions are routinely compatible with native enzyme function (Fig 4e). The data below indicate that CBB11 and CBB14 do not affect the activity of ABE.

Reviewer Comment 4.5: Regarding the schematic of Figure 4e: the ABE8e editing window tends to span between spacer nucleotides 3/4 – 8. As such, would the following underlined A not also be

edited? GACCGTCAGAGGAGACTACA. Please check, and if indeed so, correct in the schematic.

Response and Revision 4.5: Thank you for pointing this out. In fact, adenines at positions 3 and 8 are often edited alongside the target adenine at position 6. The figure has been corrected to reflect this fact.

Figure 4. Delivery of ABE-RNPs to *rd12*-reporter cells, mediated by CBB lipidoids. (...) (e)

Schematic diagram of the ABE-activity assay. EndoV = endonuclease V. Blue, SpCas9 PAM; red, target base; orange, bystander base; arrowhead, nick site; FAM, fluorescein. (f) Representative urea-PAGE-gel image of products of ABE deamination cleaved by EndoV. The gel was imaged with a fluorescein filter. N, no enzyme; P, positive control: 60 bp DNA with inosine in the middle; A, ABE-RNP in TAS; D, ABE-RNP in TAS with 10% DMSO. Numbers indicate molar excess of the CBB compound versus ABE. (g) Fluorescence quantification of substrate and products of deamination and EndoV cleavage. Two independent samples per data point were assayed in parallel.

Reviewer Comment 1.11: In all formulations DMSO is present at ~10%. Dialysis could have easily removed it... Why not done?

Original Response 1.11: We removed the solvent by dialyzing the complex of ABE-RNP and CBB11 in TAS buffer with 10% DMSO through a 7000 MWCO membrane. The CBB11 amount in the resulting solution was quantified by LC-MS. The results indicated that most of the CBB11 (95.3%) was lost, suggesting that the CBB binding to protein is reversible, which helps support our stated goal of identifying molecules that would possess transient non-specific protein binding to assist in delivery. It should also be noted that DMSO is tolerated by the eye, and DMSO is present in some formulations.

The solvent is quickly removed through vitreous exchange, thus not presenting an issue even for clinical applications. We have added a sentence to the Results conveying more clearly the utility and safety of including DMSO in the formulation.

Reviewer Comment 4.11: Once more, this fully addresses the point raised by Reviewer 1. However, as other readers may have similar questions, and it appears that not removing DMSO is of pivotal importance for the stability and downstream applications of these formulations, this information seems of high importance to the readers as well. Especially as it may not be directly clear that currently DMSO is therefore a component of the final formulation, which is of high relevance for potential applications.

Whereas additional experiments / data that shows stability of specific CBBs are not required to discuss this issue, it is a potential limitation that should be discussed. Please add a section to the discussion section that underlines the importance of DMSO in the discussion. (indeed, this is not problematic for a variety of clinical formulations, but this could of course also be added to the discussion if the authors would desire to do so).

Response and Revision 4.11: Thank you for the suggestion. We have added the following text to the discussion:

“The CBB-RNP complex contained 7.25% DMSO, which was well tolerated by ABE and as such may be further explored as a vehicle for protein delivery.”

Reviewer Comment 1.13: What are the size of these CRE/CBB complexes?

Original Response and Revision 1.13. Dynamic light scattering indicated that the size of the Cre/CBB11 complex is 1400 nm and the size of the Cre/CBB14 complex is 1500 nm. These values are now included in the Result section.

Reviewer 4.13: Thank you for including this new data. As these particles are this large, this has implications for potential suitability for specific *in vivo* applications, which should be discussed.

Response and Revision 1.13: Thank you for the comment. We cannot draw conclusions about implications of charge and hydrophobicity of the Cre-CBB complexes on the *in vivo* efficacy, as both CBB11 and CBB14 effectively delivered Cre *in vivo* despite the higher hydrophobicity of CBB11. We rephrased this part of the Results section for clarity and added the following section to the discussion:

“An additional practical determinant is particle size. Dynamic-light-scattering measurements indicated that CBB11/Cre and CBB14/Cre form ~1.4–1.5 μm assemblies, with a significant fraction of CBB14-Cre forming smaller, 160-nm particles, whereas CBB11 alone forms ~400–500 nm colloids; importantly, CBB11-doped liposomes (Lipo1–6) are ~140–160 nm and highly monodisperse. Micrometer-scale

complexes are unlikely to diffuse broadly within the neural retina or traverse ocular barriers from systemic delivery, whereas systemic delivery of the smaller, narrowly distributed liposomal formulations would be effective. Notably, despite their large hydrodynamic diameters, CBB11 and CBB14 still delivered Cre via subretinal injection.”

Original Reviewer Comment 1.15: Please show experimental evidence that CBB11 is in fact incorporated into the liposomes.

Original Response 1.15: We have performed experiments related to the solubility of CBB11 and the liposome (Lipo 1). Unconjugated CBB11 is mostly insoluble, but when mixed with lipids it is retained in the suspension (see below). Please also see relevant response to **Comment 3.2**.

Response Figure 2: CBB11 (40 μ M) in TAS with 10% DMSO or Lipo1 (containing 40 μ M CBB11) in TAS was dialyzed overnight against TAS buffer, followed by centrifugation at 5000 rpm for 5 min. The CBB11 in the supernatant was quantified *via* LC-MS. The data suggest that formulation with the liposome prevented CBB11 aggregation. Revisions in Discussion part:

Particularly striking was the high efficacy observed when these compounds were formulated into ionizable cationic liposomal particles to improve the colloidal stabilities of the CBB-protein complexes.

These facts indicated that the protein binding on the surface of Lipo1 was intensified by CBB11. As expected, this enhanced liposome-protein interaction boosted intracellular protein delivery.

Comments Reviewer 4.15 Whereas the comparison of a control lipid to CBB11 in context of liposomes in Fig 5D is an interesting experiment, it does not fully answer the question of reviewer 1. The question specifically states “: Please show experimental evidence that CBB11 is in fact incorporated into the liposomes”. However, the data shown in Figure 5d could also be explained by direct nanocomplexation of ABE8e-RNP with CBB11. Response Figure 2 is a more conclusive / direct read-out for liposomal incorporation (especially since it’s LC-MS based analysis, which is a more direct read-out).

To answer this question as the authors appear to have access to LC-MS for formulation analysis, I would request that the authors incorporate data to the manuscript that directly confirms CBB11 integration to answer this comment.

General comment, based on these replies: please avoid the use of RPM in methods section, as the rcf (g) can differ dramatically based on the rotor diameter. Please check throughout the manuscript that all mentions of centrifugation speeds are shown in rcf (g), and not in RPM.

Response and Revision 4.15: Thank you for your comment. To directly answer the question of incorporation of CBB11 into the liposomes, we decided to perform gel filtration. We used PD10 desalting

columns with a Sephadex G-25 resin. We traced the elution of CBB11 by fluorescence. We found fluorescence signals originating from CBB at elution volume (3-3.5 ml) corresponding to Lipo1 (**Fig. R1a**). As expected, we did not observe CBB fluorescence at the elution volume for Lipo2 (negative control without CBB11) or in the eluate for the column containing uncomplexed CBB11. Visually, we found that free CBB11 was retained in the column (**Fig. R1b-3**), and fluorescence scans of the columns confirmed the retention of the free CBB11 in the column (**Fig. R1c-3**).

Figure R1: Gel filtration of CBB-liposomes. (a) Coomassie fluorescence scan of a 96-well plate containing samples collected from the PD10 desalting column. V_E = elution volume. (b) A photograph of the PD10 desalting columns after the gel filtration. (c) Coomassie fluorescence scan of the PD10 desalting columns after the gel filtration.

Response to General comment: We corrected the revolutions per minute to g throughout the manuscript, where applicable. In some instances (sample dissolution in bench top thermomixer, microbial shakers) we kept rpm, a customary measure of mixing speed.

Reviewer Comment 1.18: P9L299: 'encapsulation within the particle...' if you refer to a new method, clarify or remove.

Original Response and Revision 1.18: We now clarify that ABE-RNP predominantly adsorbs to, rather than being fully encapsulated by, the lipid corona; the text has been reworded accordingly.

Reviewer Comment 4.18: Thank you for explaining this. This is now in line with the TOC graphic. However, if possible, please make sure to include a bit more clarity to the TOC graphical abstract: right now it is implied that the H^+ is present in the cytosol, rather than the endocytosis pathway.

Response and Revision 1.18: Thank you for the advice. We expanded the TOC graphic to present the concept of endosomal escape triggered by acidification of the endosome. The TOC graphic has been modified accordingly, and the new version is included in the revised manuscript.

Figure 1. Construction of complexes, and proposed mechanism(s) underlying intracellular delivery of genome-editing ribonucleoproteins (RNPs) mediated by Coomassie Brilliant Blue (CBB) liposomes. The fundamental design principles for a CBB-containing delivery agent include: 1) the CBB protein-binding domain; 2) an ionizable-linker domain; and 3) a lipid or lipidoid domain. The CBB compounds are formulated into liposomes comprising ionizable lipid (orange), phospholipid (grey), cholesterol (red) and PEG lipid (light blue); and bind RNP on the liposome surface in aqueous, neutral environment. The RNP-liposome complex is absorbed by the cells *via* an endocytosis pathway. Subsequent protonation of the ionizable CBB-delivery agents and ionizable lipids releases the tagged proteins from the particles, enabling their intracellular transport to the nucleus to edit the targeted DNA sequence.

Reviewer #5

Reviewer Comment 5.1: After evaluating the concerns raised by reviewer #3, the author's response was sufficient to address all the relevant points. The article will be ready for publication after the minor corrections indicated below:

Figure 9S panels d and e, along with f and g, require statistical descriptions and annotations to be presented. The absence of this information significantly weakens the figure's impact and prevents evaluation of the authors' claims regarding the safety and efficacy of their new Lipo1 batch. Please add the statistical test description, or at least the p-value, to the legend, and significance annotations to the graphs.

Response and Revision 5.1: Thank you for the time and effort. These suggestions have clearly improved the manuscript. Specific to Figure S9 (now Figure S11), the text has been corrected and statistical test descriptions added, as shown below.

Figure S11. Evaluation of the safety and efficacy of a new batch of Lipo1. Four-week-old wild type

(blue) and *rd12* mice (green) were administered ABE8e-RNPs with/without fresh Lipo1 *via* subretinal injections. (a) The particle size of the fresh Lipo1 preparation (green) was 148 nm, with PDI = 0.23; this was similar to that of the stored sample (yellow), size =140 nm, PDI = 0.16. (b) Fluorescence microscopic quantification of the ratio of eGFP to mCherry for *rd12* reporter cells, treated with the fresh Lipo1-RNPs (100 nM). Lipofectamine 3000 (LF, purple) served as positive control, and ABE8e-RNPs delivered without vehicles (A, gray) served as negative control; [CBB]/[RNP] ratio = 8:1. Data are presented as mean \pm S.D., n = 2 biological replicates. (c) Representative OCT images of WT mice treated with 20 μ M ABE-RNPs alone (R20), with the complex of 10 μ M RNPs plus Lipo1 containing 80 μ M CBB11 (L80R10), with Lipo1 containing 160 μ M CBB11 (L160), or with the complex of 20 μ M RNPs with Lipo1 containing 160 μ M CBB11 (L160R20); WT mice without treatment served as control (WT N). The injection volume was 1 μ l. (d) b-wave amplitude quantification of ERG responses of WT mice whose OCT scans are shown in panel c; data are presented as mean \pm S.D., n =6 eyes. Ordinary one-way ANOVA with Šídák's multiple comparisons test, **, $P < 0.01$. (e) Quantification of 11-*cis*-retinal levels in WT mice whose ERG responses are presented in panel d; data are presented as mean \pm S.D., n = 5-6 eyes. Kruskal-Wallis test with Dunn's multiple comparisons test, **, $P < 0.01$. (f) Quantification of b-wave amplitudes of ERG responses of *rd12* mice treated with 0.5 μ l of L80R10 (light green); 1 μ l of L80R10 (green); or 1 μ l of L160R20 (dark green). Untreated WT mice (WT N, black) and *rd12* mice (*rd12* N, gray) served as positive and negative controls, respectively; data are presented as mean \pm S.D., n = at least 4 eyes. Ordinary one-way ANOVA with Šídák's multiple comparisons test, **, $P < 0.01$, ****, $P < 0.0001$. (g) Quantification of 11-*cis*-retinal levels in *rd12* mice whose ERG responses are presented in panel f; data are presented as mean \pm S.D., n = at least 5 eyes. Kruskal-Wallis test with Dunn's multiple comparisons test *, $P < 0.05$.

Reviewer Comment 5.2: Figure 9S panel c: how was mild inflammation observed in the OCT images? Please, describe.

Response and Revision 5.2: The text has been revised and referenced as follows.

“The optical coherence tomography (OCT) imaging indicated no substantial alterations to the retinal structure of WT mice administrated the complex of RNP (20 μ M) and Lipo1. We noticed the presence of hyperreflective spots visible in the vitreous, indicating possible inflammation,¹ which could be ameliorated by lowering the RNP concentration (Fig. S1c)”

Reviewer Comment 5.3: Minor errors in typing: Lines 154, 232, 276: change “floreescence” to “fluorescence”. Line 277: change “efeective” to “effective”.

Response and Revision 5.3: The text has been corrected.

REFERENCES

1. Holubowicz, R. et al. Safer and efficient base editing and prime editing via ribonucleoproteins delivered through optimized lipid-nanoparticle formulations. *Nat Biomed Eng* **9**, 57-78 (2025).
2. Banskota, S. et al. Engineered virus-like particles for efficient in vivo delivery of therapeutic proteins. *Cell* **185**, 250-265 e216 (2022).

3. Asteriti, S. et al. Recombinant protein delivery enables modulation of the phototransduction cascade in mouse retina. *Cell Mol Life Sci* **80**, 371 (2023).
4. Liu, H., Zhu, Y., Li, M. & Gu, Z. Precise genome editing with base editors. *Med Rev (2021)* **3**, 75-84 (2023).
5. Xu, W., Zhang, S., Qin, H. & Yao, K. From bench to bedside: cutting-edge applications of base editing and prime editing in precision medicine. *J Transl Med* **22**, 1133 (2024).
6. Hoang, C. et al. Application of Dimethyl Sulfoxide as a Therapeutic Agent and Drug Vehicle for Eye Diseases. *J Ocul Pharmacol Th* **37**, 441-451 (2021).

Reviewer #4

Once more, I'd like to compliment the authors on their efforts to address all comments. With these current revisions, I feel that the concerns of reviewer 1 have been sufficiently addressed (**see below – comments to revisions are highlighted in yellow**). However, I do have one request for one minor revision: given the value and clarity of the results shown in figure R1, I feel that these data belong in the Supplementary Information file, rather than solely as data presented to the reviewers. Furthermore, I have pointed out two small language suggestions for the figure legend of S2. Once these minor revisions are adjusted, I would recommend the acceptance of this manuscript for publication.

Thank you for your compliment. Our responses are below in blue and italic font.

Reviewer Comment 4.1: Originally, reviewer 1 was somewhat vague in their wording, and their request could either be interpreted as a request to provide a dose-response curve in terms of the absolute amount that would be used for transfection, or an optimization of CBB-to-cargo ratio. Here, the authors opted to focus on a ratio optimization for all CBBs, which I agree is the best (and most relevant) answer to the final statement by reviewer 1 “**The optimal dose therefore needs to be established for each reagent, separately.**” It is furthermore additionally appreciated that this was done for both Cre recombinase and ABE. As such, I feel that these experiments are suitable to address the comment of Reviewer 1.

However, in this case the authors have performed the experiment twice (n=2). Whereas the data show high reproducibility and a robustness in the observed patterns, I would like to point out the following:

Please be aware that the Nature Portfolio checklist contains the following statement/guideline: “*We strongly discourage deriving statistics from technical **replicates or less than 3 biological replicates**, unless there is a clear scientific justification for why providing this information is important.*”

Especially considering these experiments are in vitro assays, I would unfortunately request the authors to perform this experiment once more, unless (as is in line with the Nature Portfolio guidelines) “there is a clear scientific justification for why providing this information is important”.

Responses and Revisions Reviewer 4.1. Thank you for the comment. We repeated the titrations for Cre and ABE with four replicates. In this new experiment, Cre delivery efficiency was comparable, whereas ABE was this time not efficiently delivered into the *rd12* reporter cells by CBB12. We replaced the Figure S2 with the new data presented as separate data points.

Fig S2. Concentration dependence of the *in vitro* delivery of gene editors mediated by representative CBB compounds. (a) Quantification by fluorescence microscopic imaging of GFP-toRFP conversion in CS cells. Cells were treated with Cre recombinase (0.5 μ M) complexed with CBB compounds at increasing concentrations (1-8 μ M). The cells are green at baseline and red after editing. Data are presented as mean \pm S.D.; n = 2. (b) Quantification by fluorescence microscopic imaging of the

intensity ratio of eGFP to mCherry in *rd12*-reporter cells treated with ABE-RNPs (100 nM) complexed with CBB compounds at increasing concentrations (0.2-1.6 μ M). Data are presented as mean \pm S.D.; n = 2.

Current version:

Fig S2. Concentration dependence of the *in vitro* delivery of gene editors mediated by representative CBB compounds. (a) Quantification by fluorescence microscopic imaging of GFP-to-RFP conversion in CS cells. Cells were treated with Cre recombinase (0.5 μ M) complexed with CBB compounds at increasing concentrations (1-8 μ M). The cells are green at baseline and red after editing. Mean \pm S.D., 4 biological replicates. (b) Quantification by fluorescence microscopic imaging of the area ratio of eGFP to mCherry in *rd12*-reporter cells treated with ABE-RNPs (100 nM) complexed with CBB compounds at increasing concentrations (0.2-1.6 μ M). Data are presented as mean \pm S.D.; n = 4 biological replicates.

With these new data, this point has been sufficiently addressed. Please make the following corrections in the figure legend: “GFP-toRFP conversion” should be “GFP-to-RFP”. Furthermore “the cells are green at basaeline and red after editing” is too colloquial in its phrasing – please include “fluorescence in this phrasing.”

Thank you for accepting our revision. This Figure is now Supplementary Figure 4. We changed the axis label accordingly, and we changed the phrasing of the figure legend to “The cells emit green fluorescence at baseline and red fluorescence after editing.”

Original Reviewer Comment 1.15: Please show experimental evidence that CBB11 is in fact incorporated into the liposomes.

Original Response 1.15: We have performed experiments related to the solubility of CBB11 and the liposome (Lipo 1). Unconjugated CBB11 is mostly insoluble, but when mixed with lipids it is retained in the suspension (see below). Please also see relevant response to **Comment 3.2**.

Response Figure 2: CBB11 (40 μ M) in TAS with 10% DMSO or Lipo1 (containing 40 μ M CBB11) in TAS was dialyzed overnight against TAS buffer, followed by centrifugation at 5000 rpm for 5 min. The CBB11 in the supernatant was quantified *via* LC-MS. The data suggest that formulation with the liposome prevented CBB11 aggregation. Revisions in Discussion part:

Particularly striking was the high efficacy observed when these compounds were formulated into ionizable cationic liposomal particles to improve the colloidal stabilities of the CBB-protein complexes. These facts indicated that the protein binding on the surface of Lipo1 was intensified by CBB11. As expected, this enhanced liposome-protein interaction boosted intracellular protein delivery.

Comments Reviewer 4.15 Whereas the comparison of a control lipid to CBB11 in context of liposomes in Fig 5D is an interesting experiment, it does not fully answer the question of reviewer 1. The question specifically states “: Please show experimental evidence that CBB11 is in fact incorporated into the liposomes”. However, the data shown in Figure 5d could also be explained by direct nanocomplexation of ABE8e-RNP with CBB11. Response Figure 2 is a more conclusive / direct read-out for liposomal incorporation (especially since it’s LC-MS based analysis, which is a more direct read-out).

To answer this question as the authors appear to have access to LC-MS for formulation analysis, I would request that the authors incorporate data to the manuscript that directly confirms CBB11 integration to answer this comment.

General comment, based on these replies: please avoid the use of RPM in methods section, as the rcf (g) can differ dramatically based on the rotor diameter. Please check throughout the manuscript that all mentions of centrifugation speeds are shown in rcf (g), and not in RPM.

Response and Revision 4.15: Thank you for your comment. To directly answer the question of incorporation of CBB11 into the liposomes, we decided to perform gel filtration. We used PD10 desalting columns with a Sephadex G-25 resin. We traced the elution of CBB11 by fluorescence. We found fluorescence signals originating from CBB at elution volume (3-3.5 ml) corresponding to Lipo1 (**Fig.**

R1a). As expected, we did not observe CBB fluorescence at the elution volume for Lipo2 (negative control without CBB11) or in the eluate for the column containing uncomplexed CBB11. Visually, we found that free CBB11 was retained in the column (**Fig. R1b-3**), and fluorescence scans of the columns confirmed the retention of the free CBB11 in the column (**Fig. R1c-3**).

Figure R1: Gel filtration of CBB-liposomes. (a) Coomassie fluorescence scan of a 96-well plate containing samples collected from the PD10 desalting column. V_E = elution volume. (b) A photograph of the PD10 desalting columns after the gel filtration. (c) Coomassie fluorescence scan of the PD10 desalting columns after the gel filtration.

With this experiment, I feel that the initial concern of Reviewer 1 has indeed been addressed. However, this proof is currently not included in the manuscript. Given the value of these data to support the general conclusions of this manuscript, and the clear results of this assay, I feel that these data belong, at the very least, in the Supplementary Info of the manuscript. Given how clear the results are, I feel that the current figure would suffice. However, if the authors wish to include a graph showing fluorescence per fraction instead of an image of the 96-well plate, this is equally acceptable.

Thank you for accepting our experiment. We included this result as a new Supplementary Figure 8. We incorporated the result in the manuscript, updated the figure numbering and Methods section.

Supplementary Figure 8: Size-exclusion chromatography of CBB-liposomes. (a) Coomassie fluorescence scan of a 96-well plate containing samples collected from PD10 desalting columns. V_e = elution volume. (b,c) A photograph (b) and a Coomassie fluorescence scan (c) of the PD10 desalting columns after the size-exclusion chromatography.

“We noted an improvement of colloidal properties of CBB11 after incorporating it into the liposomes. Upon buffer exchange, free CBB11 was absorbed by a size-exclusion chromatography resin, whereas liposome-bound CBB11 was recovered in the eluate (**Supplementary Fig. 8**).”

Prior to giving my feedback on the rebuttal I would like to notify the authors for transparency that this round was not reviewed by the same reviewer as the initial “Reviewer 1” from the previous round of peer review. As reviewer number 1 was no longer able to comment on this manuscript, I was requested as a replacing reviewer to evaluate the responses provided to the original comments of reviewer 1, with the following request: “We ask that no new concerns be raised unless they are critical to the validity of the arguments made in the manuscript”.

Firstly, I would like to compliment the authors on the amount of effort that they have shown in addressing the original comments of Reviewer 1. Especially the addition of multiple *in vivo* experiments is laudable. I will address all author comments per point below. However, there are a few comments that have not been fully addressed / resolved, as discussed below. To maintain clarity, my comments will be in blue, the comments from the original reviewer will remain shown in red.

Reviewer #1

Reviewer Comment 1.1: All *in vitro* results were done at a single concentration of the CBB derivatives and no dose-response curves are given... The optimal dose therefore needs to be established for each reagent, separately.

Response 1.1: Our previous one-point analysis was based on optimized conditions, and we would argue still valid. Nevertheless, we performed dose-response analyses as recommended.

Revision: In the revised manuscript, we now provide full *in vitro* dose-response curves (1–8 μ M CBB) for the nine CBB derivatives (CBB2, 7, 11, 12, 13, 14, 18, 19, and Z5) with Cre recombinase, or (0.2–1.6 μ M CBB) with ABE8e-RNP, now included as **Fig. S2**.

Figure S2. Concentration dependence of the *in vitro* delivery of gene editors mediated by representative CBB compounds. (a) Quantification by fluorescence microscopic imaging of GFP-to-RFP conversion in CS cells. Cells were treated with Cre recombinase (0.5 μ M) complexed with CBB compounds at increasing concentrations (1–8 μ M). The cells are green at baseline and red after editing.

Data are presented as mean \pm S.D.; n = 2. (b) Quantification by fluorescence microscopic imaging of the intensity ratio of eGFP to mCherry in *rd12*-reporter cells treated with ABE-RNPs (100 nM) complexed with CBB compounds at increasing concentrations (0.2-1.6 μ M). Data are presented as mean \pm S.D.; n = 2.

Revisions in Results part:

***In Vitro* Delivery of Cre by Coomassie-Derived Lipidoids**

Nine compounds (CBB2, 7, 11, 12, 13, 14, 18,19, Z1) were selected for the titration of the [CBB] to [Enzyme] ratio. The CBB lipidoids were dissolved in DMSO (1 volume) and complexed at room temperature with Cre recombinase (19 volumes) at a 2:1, 4:1, 8:1 or 16:1 molar ratio before delivery (**Fig. S2a**). The enzymatic activity of cytosolic Cre was evaluated by fluorescence microscopy according to the extent of GFP to RFP conversion, which indicated that Cre recombinase intracellular delivery was dependent on the CBB-lipidoids concentration, and the dose-response relationship showed that the effect was generally saturated at a ratio 8:1 for most compounds.

***In Vitro* Delivery of ABE-RNP**

The titration of [CBB2, 7, 11, 12, 13, 14, 18,19, Z1] to [ABE-RNP] with fluorescence microscopy also indicated that higher concentrations of CBB compounds facilitated RNP delivery (**Fig. S2b**). In the broad screening of CBB compounds, the ratio of [CBB]/[Enzyme] was kept at 8:1 to decrease the blue precipitation during flow cytometry measurement.

Revisions in Methods part:

Delivery of Cre recombinase *in vitro*

at final concentrations of 10 μ M of Cre enzyme and 20-160 μ M of CBB compound.

Delivery of ABE *in vitro*

Next, ABE-RNP was diluted in TAS buffer (20.1 mM Tris, 4.3 mM sodium acetate, pH 7.3, sucrose 10% (v/v)) and added to CBB-lipidoids at 1:2 to 1:16 ratios of enzyme:compound, 5%/7.25% (v/v) final DMSO, 10 μ M final RNP, and incubated for a further 15 min (**Fig. 4**) or 5 h (**Fig. 5 f, i**) at room temperature.

Originally, reviewer 1 was somewhat vague in their wording, and their request could either be interpreted as a request to provide a dose-response curve in terms of the absolute amount that would be used for transfection, or an optimization of CBB-to-cargo ratio. Here, the authors opted to focus on a ratio optimization for all CBBs, which I agree is the best (and most relevant) answer to the final statement by reviewer 1 “The optimal dose therefore needs to be established for each reagent, separately.” It is furthermore additionally appreciated that this was done for

both Cre recombinase and ABE. As such, I feel that these experiments are suitable to address the comment of Reviewer 1.

However, in this case the authors have performed the experiment twice (n=2). Whereas the data show high reproducibility and a robustness in the observed patterns, I would like to point out the following:

Please be aware that the Nature Portfolio checklist contains the following statement/guideline: *“We strongly discourage deriving statistics from technical replicates or less than 3 biological replicates, unless there is a clear scientific justification for why providing this information is important.”*

Especially considering these experiments are in vitro assays, I would unfortunately request the authors to perform this experiment once more, unless (as is in line with the Nature Portfolio guidelines) “there is a clear scientific justification for why providing this information is important”.

Reviewer Comment 1.2: The in vivo data focus on base editing results, but not so much on toxicity and off-target editing... I would at least expect to see some evidence that toxicity to the retina is absent or limited at the tested concentrations.

Response 1.2:

Toxicity: We have performed a new series of experiments investigating the effects of our molecules on wild-type mice (C57BL/6). Here we present OCT images of wild-type retinas at 7 days post-injection, followed by a functional ERG analysis at 14 days and retinoid analyses at 21 days that reflect the content of rhodopsin. The data indicates that the wild-type mice treated with the complex of a new batch of Lipo1 with RNP (10 μ M) showed minimal changes to the retinal structure and function. The minor changes observed are due to retinal surgery. The mouse eyeball has a diameter of \sim 3.4 mm and volume \sim 24 mm³ (\sim 24 μ L); and the lens represents about 15-20% of the total eye volume. Mouse lenses are relatively large compared to the eye size (compared to human proportions), making injection to subretinal space challenging without disturbing the lens. Any perturbation of the lens would lead to cataract, and the eye would be useless for imaging or physiological testing. Only skilled personnel did the injections, and no such gross effects were observed.

Revision:

Figure S9. Evaluation of the safety and efficacy of a new batch of Lipo1. Four-week-old wild type (blue) and *rd12* mice (green) were administered ABE8e-RNPs with/without new Lipo1 *via* subretinal injections. (a) The particle size of the new Lipo1 preparation (green) was 148 nm, with PDI = 0.23; this was similar to that of the old batch (yellow), size = 140 nm, PDI = 0.16. (b) Quantification *via* fluorescence microscopy of the ratio of eGFP to mCherry for *rd12* reporter cells, treated with the new Lipo1-RNPs (100 nM). Lipofectamine 3000 (LF, purple) served as positive control, and ABE8e-RNPs delivered without

vehicles (A, gray) served as negative control; [CBB]/[RNP] ratio = 8:1. Data are presented as mean \pm S.D., n = 2 replicates. (c) Representative OCT images of WT mice treated with 20 μ M ABE-RNPs alone (R20), with the complex of 10 μ M RNPs plus Lipo1 containing 80 μ M CBB11 (L80R10), with Lipo1 containing 160 μ M CBB11 (L160), or with the complex of 20 μ M RNPs with Lipo1 containing 160 μ M CBB11 (L160R20); WT mice without treatment served as control (WT N). The injection volume was 1 μ l. (d) b-wave amplitude quantification of ERG responses of WT mice whose representative OCT scans are shown in panel c; data are presented as mean \pm S.D., n = 6 eyes. (e) Quantification of 11-*cis*-retinal levels in WT mice whose ERG responses are presented in panel d; data are presented as mean \pm S.D., n = 5-6 eyes. (f) Quantification of b-wave amplitudes of ERG responses of *rd12* mice treated with 0.5 μ l of L80R10 (light green); 1 μ l of L80R10 (green); or 1 μ l of L160R20 (dark green). Untreated WT mice (WT N, black) and *rd12* mice (*rd12* N, gray) served as positive and negative controls, respectively; data are presented as mean \pm S.D., n = 6-8 eyes. (g) Quantification of 11-*cis*-retinal levels in *rd12* mice whose ERG responses are presented in panel f; data are presented as mean \pm S.D., n = 5-7 eyes.

Revisions in Results part:

Protein Delivery by CBB-liposomes *In Vivo*

To evaluate the safety of the ABE-RNP delivery to the retina mediated by Lipo1, we also administrated ABE-RNP-Lipo1 complex to WT mice (**Fig. S9**) *via* subretinal injection. Since the oxidation of CBB11 might induce epoxides in the liposome, only freshly formulated Lipo1 was applied to the WT mice. Different batches of Lipo1 showed similar morphology and comparable ABE-RNP delivery efficiency to *rd12* reporter cell lines (**Fig. S9a, b**). The Optical Coherence Tomography (OCT) imaging indicated no substantial alterations to the retinal structure of WT mice administrated the complex of RNP (20 μ M) and Lipo1; except mild inflammation, which could be ameliorated substantially by lowering the RNP concentration (**Fig. S9c**). The ERG response and the generation of the visual chromophore, 11-*cis* retinal, of WT mice treated with Lipo1 and RNP (10 μ M) were also comparable to those of untreated WT mice (**Fig. S9d, e**). In parallel experiments in which *rd12* mice were treated with the same batch of Lipo1 and RNP complex, the subretinal injections of 1 μ l of Lipo1 with 10 μ M RNP achieved an optimal ERG response and visual chromophore recovery (**Fig. S9d, e**), comparable to those of *rd12* mice treated with Lipo1-ABE-RNP (20 μ M) (**Fig. 6**). These results suggested that optimal vision-function rescue requires a balance between the biological tolerance of the cells and the editing efficiency of the gene editors.

Revisions in Methods part:

Delivery of ABE *in vivo*

ABE RNP in TAS was added to concentrated CBB-lipidoid in DMSO or CBB-liposome in TAS, mixed by rapid pipetting and incubated with the reagents for at least 10 min at room temperature, and injected

subretinally into *rd12* or WT mice. The remaining reagents were then applied onto *rd12* reporter cells in DMEM with 10% FBS in a 96-well plate (**Fig. 5f, i**).

Off-target editing—Targeted deep sequencing of the ten highest-ranked off-target sites (identified using Cas-OFFinder) shows a background-level A•T→G•C substitution rate comparable to that of untreated animals. Additional data on off-target analyses are now included.

Revisions: These data are now included in the Results as an expanded **Fig. 6g**.

Figure 6 (g) Off-target genome-editing by ABE-RNPs delivered with Lipo1 *in vivo*. DNA from the RPE of mice treated with 20 μ M ABE-RNP complexed with Lipo1, and from untreated controls were subjected to high-throughput targeted-amplicon sequencing. Adenosine deamination at the on-target and off-target sites (OT) were analyzed using CRISPResso. N = 4 replicates.

Revisions in Results part:

Protein Delivery by CBB-liposomes *In Vivo*

The targeted deep sequencing of the ten highest-ranked off-target sites (identified using Cas-OFFinder) shows a background-level A to G substitution rate on these sites for the *rd12* mice treated with Lipo1-ABE8e-RNPs which is comparable to that of untreated animals (**Fig. 6g**).

Once more, I feel that the authors have chosen suitable experiments to address the points raised by reviewer 1. Whereas n=2 replicates for Fig S9B, I feel that this is justified in this case due to the robustness of the data, as well as the ethical implications to minimize the amount of animals. Data is clear, methods are described well, conclusions in the main text are appropriate.

One request: I do not have access to the Supplementary Tables, so I'm not sure if all OT genomic sequences / analysis sequences are listed. As such, I would like to request to please make sure to include the analyzed Off-targeted sequence (+genomic location), as well as potential relevant primer/probe sequences used for OT analysis (either to the methods section, or the Supplementary files).

Reviewer Comment 1.3: Some claims are made in the discussion that are not covered by the data presented (toxicity, RNP encapsulation, immunogenicity). Better to remove such unsubstantiated claims.

Response and Revision 1.3: Thank you for bringing up this important point. We have removed unsubstantiated statements regarding toxicity (see **Response 1.2**), immunogenicity, and full "encapsulation". All remaining claims are now directly supported by new data (toxicity, off-target analysis) or by published references cited in the Discussion.

The changes are appreciated.

Indeed, published references can be used to support claims on safety as well. In line with this, I would request adding a reference that shows decreased bystander editing upon short-term ABE delivery (ie. as RNP) to the statement "In general, the delivery of ABE-RNPs mediated by CBB-liposomes could be considered as a more transient mode of delivery of gene-editing tools, consistent with decreased bystander editing".

Also, please add a reference to the statement "small particles would better penetrate the retina and make the therapeutic agent available for cellular uptake".

Reviewer Comment 1.4: From Cre recombinase to base editors. A logical step in between could have been CRISPR-Cas9 based gene correction. Did you try this as well?

Response 1.4: Thank you for the suggestion. Cas9 in the eye causes severe off-target effects and apoptosis.¹ Therefore, we deliberately applied our CBB strategy to an adenine base editor (ABE) rather than to a Cas9 nuclease, since ABEs offer superior therapeutic promise. This choice is further validated by recent clinical successes, most notably a personalized ABE-based gene therapy administered to an infant patient, which underscores the clinical relevance of base editing over nuclease-driven approaches.²

This is a valid explanation, and as reviewer 1 did not request any additional data or experiments this answer suffices. However, readers may from outside the ocular delivery field may have similar questions. As such, I would request that this is briefly explained/addressed in the discussion as well.

Reviewer Comment 1.5: CBB dyes have denaturing capacities. Do you see any reduction in activity of Cre recombinase or base editors in the presence (without encapsulation)?

Response 1.5: Coomassie Brilliant Blue (CBB) dyes indeed show higher affinity for partially unfolded proteins (e.g. Triton-X100 increase binding Coomassie G250 in the Bradford assay ³), yet decades of successful non-denaturing affinity-chromatography with structurally similar Cibacron Blue-based "Blue Sepharose"⁴ demonstrate that such interactions are routinely compatible with native enzyme function. The only adverse phenomenon we observed was formulation-dependent precipitation with CBB lipoids. The data below indicate that CBB11 and CBB14 do not affect the activity of ABE.

Figure 4(e) Schematic diagram of the ABE-activity assay. EndoV = endonuclease V. Blue, SpCas9 PAM; red, target base; arrowhead, nick site; FAM, fluorescein. **(f)** Representative urea-PAGE-gel image of products of ABE deamination cleaved by EndoV. The gel was imaged with a fluorescein filter. N, no enzyme; P, positive control: 60 bp DNA with inosine in the middle; A, ABE-RNP in TAS; D, ABE-RNP in TAS with 10% DMSO. Numbers indicate molar excess of the CBB compound *versus* ABE. **(g)** Fluorescence quantification of substrate and products of deamination and EndoV cleavage. Two independent samples per data point were assayed in parallel.

Revisions: We have added the new figure to the manuscript

Revisions in Results part:

***In Vitro* Delivery of ABE-RNP**

The complexing of ABE-RNPs with CBB lipidoids with 10% DMSO did not deteriorate the deaminase activity (**Fig. 4e-g**)⁵.

Revision in Methods part:

ABE activity assay

Regarding the schematic of Figure 4e: the ABE8e editing window tends to span between spacer nucleotides 3/4 – 8. As such, would the following underlined A not also be edited?

GACCGTCAGAGGAGACTACA

Please check, and if indeed so, correct in the schematic.

Reviewer Comment 1.6: P3L84: Lipofectin should be lipofectamine or else describe the procedure of lipofection.

Response and Revision 1.6: We were unclear, and appreciate the comment. We agree that the two terms mean different things, and we have now used them correctly. In the introduction, we correctly mentioned “Lipofectin.”⁶ Also, we removed all the lipofectin in the results and methods part.

Revision in Introduction part:

The first commercial cationic liposome-based transfection product, Lipofectin, was introduced in 1988 to deliver plasmids into cells *in vitro*.⁷ Since then, further development has resulted in numerous ionizable/permanently ionized lipids with multiple functionalities like Lipofectamine, MC3;⁸ and delivery of large RNPs has been reported using lecithin-based liposomes⁹ and ionizable lipid nanoparticles.¹⁰

Thank you, this fully addresses the point of the reviewer.

Reviewer Comment 1.7: P4L956... remove claim of potential safety about your library of CBB lipidoids as you only tested a few.

Response 1.7: We would like to clarify that one component of the CBB lipidoids, specifically G250, is approved for clinical use. However, we acknowledge that the entire set of CBB lipidoids will need to be thoroughly investigated for safety and efficacy before any potential clinical application.

Revision: We have modified our statements regarding safety to better reflect the speculative nature of our statements regarding safety.

Revision in Introduction part.

We sought to improve the existing genome editing toolkit by developing a safe and efficient protein delivery approach.

Thank you, this fully addresses the point of the reviewer.

Reviewer Comment 1.8: Fig. 2. Only one concentration of CBB derivatives was tested... some might be active at lower/higher concentrations.

Response and Revision 1.8: Thank you for this suggestion. The response to this inquiry is addressed in our response to Comment 1.1.

Thank you, this fully addresses the point of the reviewer.

Reviewer Comment 1.9: There does not seem to be any SAR in the tested derivatives...

Response 1.9: While we cannot yet provide a predictive model for the construction of an ideal CBB lipidoid to facilitate RNP delivery, the presented data provide a foundation for a SAR. Our work presents a novel concept in gene delivery, and we are confident that reviewers will appreciate how extensive our studies are at this point. The next step would be to focus on a few CBBs and work to generate a family of highly related compounds to discover the key molecular determinants of facilitated delivery. We expect that the current studies will inspire other laboratories also to build on this approach to improve delivery of gene-editing agents.

Revisions: We added few sentences in the limitation section of the manuscript.

Revision in Results part:

CBB18, a combination of CPP5 and TFA, further increased the color conversion to $49.0 \pm 3.6\%$, suggesting synergistic effects between cell-penetrating peptides and CBB. In general, the leading CBB compounds for Cre recombinase delivery, CBB14, 18, 19 and 21, are more positively charged under neutral aqueous conditions, and their lipid moieties are less hydrophobic. Even though such properties of these compounds do not substantially change the particle size of the respective CBB/Cre complexes (The particle diameter of CBB11/Cre is about 1440 nm and that of CBB14/Cre is around 1510 nm), they still could improve the colloidal stabilities of the CBB/Cre complexes.

This, in combination with the final sections and the discussion on C16-C18 tails in the discussion section sufficiently addresses this point (moreover, the reviewer did not request any specific deliverables).

Reviewer Comment 1.10: P5L150 'did not enable Cre uptake...' Wrong conclusion as you are not looking at cellular uptake but Cre activity. Please change.

Response 1.10: Thank you for pointing out this inaccuracy. We agree that lack of evidence of Cre activity in the reporter cells does not mean that Cre was not absorbed by the cells.

Revision: Accordingly, we have changed the wording in the text from "did not enable Cre uptake" to "did not lead to detectable Cre-mediated recombination activity" in the reporter cells.

Revision in Results part:

***In Vitro* Delivery of Cre by Coomassie-Derived Lipidoids**

The enzymatic activity of cytosolic Cre was evaluated by fluorescence microscopy according to the extent of GFP to RFP conversion, which indicated that Cre recombinase intracellular delivery was dependent on the CBB-lipidoids concentration,

Thank you, this fully addresses the point of the reviewer.

Reviewer Comment 1.11: In all formulations DMSO is present at ~10%. Dialysis could have easily removed it... Why not done?

Response 1.11: We removed the solvent by dialyzing the complex of ABE-RNP and CBB11 in TAS buffer with 10% DMSO through a 7000 MWCO membrane. The CBB11 amount in the resulting solution was quantified by LC-MS. The results indicated that most of the CBB11 (95.3%) was lost, suggesting that the CBB binding to protein is reversible, which helps support our stated goal of identifying molecules that would possess transient non-specific protein binding to assist in delivery. It should also be noted that DMSO is tolerated by the eye, and DMSO is present in some formulations. The solvent is quickly removed through vitreous exchange, thus not presenting an issue even for clinical applications.

Revision: We have added a sentence to the Results conveying more clearly the utility and safety of including DMSO in the formulation.

Revision in Results part:

***In Vitro* Delivery of ABE-RNP**

The complexing of ABE-RNP with CBB lipidoids with 10% DMSO did not deteriorate the deaminase activity (**Fig. 4e-g**).

Once more, this fully addresses the point raised by Reviewer 1. However, as other readers may have similar questions, and it appears that not removing DMSO is of pivotal importance for the stability and downstream applications of these formulations, this information seems of high importance to the readers as well. Especially as it may not be directly clear that currently DMSO is therefore a component of the final formulation, which is of high relevance for potential applications.

Whereas additional experiments / data that shows stability of specific CBBs are not required to discuss this issue, it is a potential limitation that should be discussed. Please add a section to the discussion section that underlines the importance of DMSO in the discussion.

(indeed, this is not problematic for a variety of clinical formulations, but this could of course also be added to the discussion if the authors would desire to do so).

Reviewer Comment 1.12: P6 concentrations for subretinal injection of Cre recombinase do not match those in the MandM section.

Response and Revision 1.12: Thank you. We corrected this mistake. Now, the concentrations in the Methods match the main text (Cre, 2 µg per µL; injection volume = 1 µL).

Thank you, this fully addresses the point of the reviewer.

Reviewer Comment 1.13: What are the size of these CRE/CBB complexes?

Response and Revision 1.13: Dynamic light scattering indicated that the size of the Cre/CBB11 complex is 1400 nm and the size of the Cre/CBB14 complex is 1500 nm. These values are now included in the Result section.

Revisions in Results part:

***In Vitro* Delivery of Cre by Coomassie-Derived Lipidoids**

Even though such properties of these compounds do not substantially change the particle size of the respective CBB/Cre complexes (The particle diameter of CBB11/Cre is about 1440 nm and that of CBB14/Cre is around 1510 nm), they still could improve the colloidal stabilities of the CBB/Cre complexes.

Thank you for including this new data. As these particles are this large, this has implications for potential suitability for specific *in vivo* applications, which should be discussed.

Reviewer Comment 1.14: Explain why CBB14 is good for Cre recombinase but not for the ABE-RNP base editor.

Response 1.14: Cre-CBB and ABE-CBB are different entities. They dramatically differ in size, and their cellular pathways to the nucleus could be affected by their physicochemical properties. We have collected new data in an attempt to understand these differences. A key issue is the relative solubility of the two complexes; and we are actively investigating whether differential performance is attributable to colloidal stability and net charge. Cre (38 kDa, pI ~ 9.6, net positive charge) forms a stable shell with CBB14, whereas the larger, anionic ABE-RNP (180 kDa protein and 33 kDa RNA) induces rapid aggregation of CBB14 (See **Response Figure 1**, below).

Response Figure1. Soluble versus Precipitable CBB14 in the complexes of CBB14 with Cre or RNP after gentle centrifugation. Ten microliters of stock solutions of Cre or ABE-RNP (final 10 μ M) were mixed with a solution of CBB14 (final 80 μ M) in cell-culture medium with 5% DMSO; then centrifuged at 800 g for 10 min. The amounts of CBB14 in the supernatant and precipitate were analyzed *via* LC-MS. The larger amount of CBB14 in the precipitate of the CBB-RNP complex suggests that the CBB-RNP complex has lower colloidal stability.

Revision: We have added a brief description of this pilot experiment and its implication to the Discussion, but these data are preliminary, and will be the focus of future work.

Revisions in Discussion part:

The CBB lipidoids facilitated the cellular uptake and recombination event of the small positively charged protein, Cre. Remarkable Cre-recombinase delivery mediated by CBB14 was achieved *in vitro* (**Fig. 2c**) and *in vivo* (**Fig. 3b**). The colloidal stability and protein-delivery efficiency of CBB compounds dropped dramatically when they were bound to the large, aggregation-prone and negatively charged ABE-RNP (**Fig. 4c**).

Thank you, this fully addresses the point of the reviewer.

Reviewer Comment 1.15: Please show experimental evidence that CBB11 is in fact incorporated into the liposomes.

Response 1.15: We have performed experiments related to the solubility of CBB11 and the liposome (Lipo 1). Unconjugated CBB11 is mostly insoluble, but when mixed with lipids it is retained in the suspension (see below). Please also see relevant response to **Comment 3.2**.

Response Figure 2: CBB11 (40 µM) in TAS with 10% DMSO or Lipo1 (containing 40 µM CBB11) in TAS was dialyzed overnight against TAS buffer, followed by centrifugation at 5000 rpm for 5 min. The CBB11 in the supernatant was quantified *via* LC-MS. The data suggest that formulation with the liposome prevented CBB11 aggregation.

Moreover, solubilization of CBB11 is documented further by comparing the effects of Lipo 1 *versus* Lipo 2 (see **Fig. 5 d**, below). Lipo1 is less negatively charged than Lipo2. The observed difference also indicates that CBB11 is in fact incorporated into the liposome membrane. This interpretation is further supported by the fact that Lipo1 became much more negatively charged after complexing with RNP, indicating intensified protein binding to the liposome surface. We attribute the differences to the incorporation of CBB11 in the Lipo1 membrane.

Revision: We have added a verbal description of these observations to the results.

Revisions in Results part:

***In Vitro* Delivery of ABE-RNP**

Notably, eGFP expression in *rd12* reporter cells dropped from $28.6 \pm 0.7\%$ to $6.1 \pm 1.4\%$ as the incubation time of CBB11 with ABE-RNP (final concentration at 100 nM) increased from 15 min (**Fig. 4c**) to 5 h (**Fig. 5d**), accompanied by precipitation. Apparently, the colloidal instability of CBB11 hindered the uptake of ABE-RNPs by the cells, or possibly interfered with editing. To improve the distribution of CBB11 in an aqueous environment, we formulated novel liposomes (Lipo1/3/5) containing, in molar ratio, 5% CBB11, 50% ionizable lipids (SM102, ALC-0315, MC3), 10% 1,2-distearoyl-sn-glycero-3-phosphocholine (DSPC), 33.5% cholesterol, and 1.5% 1,2-dimyristoyl-rac-glycero-3-methoxypolyethylene glycol-2000 (DMG-PEG 2000); and a control liposome (Lipo2/4/6) without CBB11 (**Fig. 5a**).

We also did not observe precipitation of ABE-RNP with either Lipo1/3/5 or Lipo2/4/6.

Revisions in Discussion part:

Particularly striking was the high efficacy observed when these compounds were formulated into ionizable cationic liposomal particles to improve the colloidal stabilities of the CBB-protein complexes.

Figure 5 (d) The zeta potentials of Lipo1 (blue) and Lipo2 (yellow) in the absence (-, light color) or the presence (+, dark color) of ABE-RNP. The ratio of ABE-RNP to CBB11 is 1:8.

Revisions in Results part:

***In Vitro* Delivery of ABE-RNP**

Furthermore, the presence of ABE-RNP dramatically changed the zeta potential on the Lipo1 surface (a net decrease of 10.16 mV), but marginally affected the zeta potential on the Lipo2 surface (a net decrease of 0.64 mV) (**Fig. 5d**).

These facts indicated that the protein binding on the surface of Lipo1 was intensified by CBB11. As expected, this enhanced liposome-protein interaction boosted intracellular protein delivery.

Whereas the comparison of a control lipid to CBB11 in context of liposomes in Fig 5D is an interesting experiment, it does not fully answer the question of reviewer 1. The question specifically states “: Please show experimental evidence that CBB11 is in fact incorporated into the liposomes”. However, the data

shown in Figure 5d could also be explained by direct nanocomplexation of ABE8e-RNP with CBB11. Response Figure 2 is a more conclusive / direct read-out for liposomal incorporation (especially since it's LC-MS based analysis, which is a more direct read-out).

To answer this question As the authors appear to have access to LC-MS for formulation analysis, I would request that the authors incorporate data to the manuscript that directly confirms CBB11 integration to answer this comment.

General comment, based on these replies: please avoid the use of RPM in methods section, as the rcf (g) can differ dramatically based on the rotor diameter. Please check throughout the manuscript that all mentions of centrifugation speeds are shown in rcf (g), and not in RPM.

Reviewer Comment 1.16: It is unclear what is shown in Fig 5C... Please explain.

Response 1.16: Fig. 5c reports the liposome-facilitated Cre recombinase delivery into color-switching cells. The RFP/GFP conversion measured by fluorescence microscopy indicates the presence of Cre recombination activity in the nucleus.

Fig. 5f, i report the liposome-facilitated ABE8e-RNP delivery into *rd12* reporter cells. The discrepancy between **Fig. 4** and **Fig. 5e** is due to the slow colloidal aggregation triggered by protein binding to the colloids formed with CBB11 alone. The complexing time of CBB compound with RNP in **Fig. 4** is 15 min and in **Fig 5f, i** is 5 h because the complex was first used for subretinal injection, then for the *in vitro* assay.

Revision: More details are described under Methods

Revisions in Methods parts:

Delivery of ABE *in vitro*.

Next, ABE-RNP was diluted in TAS buffer (20.1 mM Tris, 4.3 mM sodium acetate, pH 7.3, sucrose 10% (v/v)) and added to CBB-lipidoids at a 1:2 to 1:16 ratio of enzyme:compound, 5%/7.25% (v/v) final DMSO, 10 μ M final RNP, and incubated for a further 15 min (**Fig. 4**) or 5 h (**Fig. 5 f, i**) at room temperature.

Thank you for the explanation, and the extra information in the methods section.

Reviewer Comment 1.17: Fig 6: 20 μ M ABE-RNP was dosed *in vivo*... 100x higher than *in vitro*. Provide rationale. Wouldn't you expect toxicity? Show toxicology.

Response and Revision 1.17: The maximum volume that can be injected into a mouse eye is 1-2 μ l. However, the subretinal space is highly restricted and prevents any dilution with the vitreous. In tissue culture, the cells are present as adhesion cells on the surface with large access to tissue culture medium,

therefore requiring higher concentrations as our reagent is highly diluted and the ratio to the cells is drastically lower than *in vivo*. In terms of toxicity, please see response to comment 1.2.

Thank you for the explanation.

Reviewer Comment 1.18: P9L299: 'encapsulation within the particle...' if you refer to a new method, clarify or remove.

Response and Revision 1.18: We now clarify that ABE-RNP predominantly adsorbs to, rather than being fully encapsulated by, the lipid corona; the text has been reworded accordingly.

Thank you for explaining this. This is now in line with the TOC graphic. However, if possible, please make sure to include a bit more clarity to the TOC graphical abstract: right now it is implied that the H⁺ is present in the cytosol, rather than the endocytosis pathway.

Revisions in Results part:

Protein Delivery by CBB-liposomes *In Vivo*

Precisely edited gDNA sequences in mice injected with ABE-RNPs absorbed to the corona layer on Lipo1 were 127-, 20-, 50-, 9.7-, and 6.5-fold higher than those with TAS, DMSO, CBB11, CBB14, and Lipo2, respectively.

Thank you, this fully addresses the point of the reviewer.

Reviewer Comment 1.19: P10L318: '... CBB-lipoids also showed reduced ocular toxicity... Where can I find these results?'

Response and Revision 1.19: This comment is broadly addressed with the new toxicity data addressed in response to **comment 1.2**.

Thank you, this fully addresses the point of the reviewer.

Reviewer Comment 1.20

P10L328: 'We predict less immunogenic than a liposome without CBB...' The reviewer argues the opposite. Please comment.

Response 1.20

Revision: We have removed the speculative statement, and again refer to the response to **comment 1.2**.

Thank you, this fully addresses the point of the reviewer.

- 1 Pulman, J. *et al.* Direct delivery of Cas9 or base editor protein and guide RNA complex enables genome editing in the retina. *Mol Ther Nucleic Acids* **35**, 102349 (2024). <https://doi.org/10.1016/j.omtn.2024.102349>
- 2 Musunuru, K. *et al.* Patient-Specific In Vivo Gene Editing to Treat a Rare Genetic Disease. *N. Engl. J. Med.* **392**, 2235–2243 (2025). <https://doi.org/10.1056/NEJMoa2504747>
- 3 Friedenauer, S. & Berlet, H. H. Sensitivity and variability of the Bradford protein assay in the presence of detergents. *Anal Biochem* **178**, 263–268 (1989). [https://doi.org/10.1016/0003-2697\(89\)90636-2](https://doi.org/10.1016/0003-2697(89)90636-2)
- 4 Bohme, H. J., Kopperschlager, G., Schulz, J. & Hofmann, E. Affinity chromatography of phosphofructokinase using Cibacron blue F3G-A. *J Chromatogr* **69**, 209–214 (1972). [https://doi.org/10.1016/s0021-9673\(00\)83103-9](https://doi.org/10.1016/s0021-9673(00)83103-9)
- 5 Holubowicz, R. *et al.* Safer and efficient base editing and prime editing via ribonucleoproteins delivered through optimized lipid-nanoparticle formulations. *Nat Biomed Eng* **9**, 57–78 (2025). <https://doi.org/10.1038/s41551-024-01296-2>
- 6 Malone, R. W., Felgner, P. L. & Verma, I. M. Cationic liposome-mediated RNA transfection. *Proc. Natl. Acad. Sci. U. S. A.* **86**, 6077–6081 (1989). <https://doi.org/10.1073/pnas.86.16.6077>
- 7 Felgner, P. L. *et al.* Lipofection: a highly efficient, lipid-mediated DNA-transfection procedure. *Proc Natl Acad Sci U S A* **84**, 7413–7417 (1987). <https://doi.org/10.1073/pnas.84.21.7413>
- 8 Cullis, P. R. & Felgner, P. L. The 60-year evolution of lipid nanoparticles for nucleic acid delivery. *Nat Rev Drug Discov* **23**, 709–722 (2024). <https://doi.org/10.1038/s41573-024-00977-6>
- 9 Cho, E. Y. *et al.* Lecithin nano-liposomal particle as a CRISPR/Cas9 complex delivery system for treating type 2 diabetes. *J Nanobiotechnology* **17**, 19 (2019). <https://doi.org/10.1186/s12951-019-0452-8>
- 10 Riley, R. S. *et al.* Ionizable lipid nanoparticles for in utero mRNA delivery. *Sci Adv* **7** (2021). <https://doi.org/10.1126/sciadv.aba1028>

Reviewer #4

Once more, I'd like to compliment the authors on their efforts to address all comments. With these current revisions, I feel that the concerns of reviewer 1 have been sufficiently addressed (**see below – comments to revisions are highlighted in yellow**). However, I do have one request for one minor revision: given the value and clarity of the results shown in figure R1, I feel that these data belong in the Supplementary Information file, rather than solely as data presented to the reviewers. Furthermore, I have pointed out two small language suggestions for the figure legend of S2. Once these minor revisions are adjusted, I would recommend the acceptance of this manuscript for publication.

Reviewer Comment 4.1: Originally, reviewer 1 was somewhat vague in their wording, and their request could either be interpreted as a request to provide a dose-response curve in terms of the absolute amount that would be used for transfection, or an optimization of CBB-to-cargo ratio. Here, the authors opted to focus on a ratio optimization for all CBBs, which I agree is the best (and most relevant) answer to the final statement by reviewer 1 **“The optimal dose therefore needs to be established for each reagent, separately.”** It is furthermore additionally appreciated that this was done for both Cre recombinase and ABE. As such, I feel that these experiments are suitable to address the comment of Reviewer 1.

However, in this case the authors have performed the experiment twice (n=2). Whereas the data show high reproducibility and a robustness in the observed patterns, I would like to point out the following:

Please be aware that the Nature Portfolio checklist contains the following statement/guideline: *“We strongly discourage deriving statistics from technical **replicates or less than 3 biological replicates**, unless there is a clear scientific justification for why providing this information is important.”*

Especially considering these experiments are in vitro assays, I would unfortunately request the authors to perform this experiment once more, unless (as is in line with the Nature Portfolio guidelines) *“there is a clear scientific justification for why providing this information is important”*.

Responses and Revisions Reviewer 4.1. Thank you for the comment. We repeated the titrations for Cre and ABE with four replicates. In this new experiment, Cre delivery efficiency was comparable, whereas ABE was this time not efficiently delivered into the *rd12* reporter cells by CBB12. We replaced the Figure S2 with the new data presented as separate data points.

Fig S2. Concentration dependence of the *in vitro* delivery of gene editors mediated by representative CBB compounds. (a) Quantification by fluorescence microscopic imaging of GFP-toRFP conversion in CS cells. Cells were treated with Cre recombinase (0.5 μM) complexed with CBB compounds at increasing concentrations (1-8 μM). The cells are green at baseline and red after editing. Data are presented as mean \pm S.D.; n = 2. (b) Quantification by fluorescence microscopic imaging of the

intensity ratio of eGFP to mCherry in *rd12*-reporter cells treated with ABE-RNPs (100 nM) complexed with CBB compounds at increasing concentrations (0.2-1.6 μM). Data are presented as mean \pm S.D.; n = 2.

Current version:

Fig S2. Concentration dependence of the *in vitro* delivery of gene editors mediated by representative CBB compounds. (a) Quantification by fluorescence microscopic imaging of GFP-toRFP conversion in CS cells. Cells were treated with Cre recombinase (0.5 μM) complexed with CBB compounds at increasing concentrations (1-8 μM). The cells are green at baseline and red after editing. Mean \pm S.D., 4 biological replicates. (b) Quantification by fluorescence microscopic imaging of the area ratio of eGFP to mCherry in *rd12*-reporter cells treated with ABE-RNPs (100 nM) complexed with CBB compounds at increasing concentrations (0.2-1.6 μM). Data are presented as mean \pm S.D.; n = 4 biological replicates.

With these new data, this point has been sufficiently addressed. Please make the following corrections in the figure legend: “GFP-toRFP conversion” should be “GFP-to-RFP”. Furthermore “the cells are green at basaeline and red after editing” is too colloquial in its phrasing – please include “fluorescence in this phrasing.”

Reviewer Comment 4.2: Once more, I feel that the authors have chosen suitable experiments to address the points raised by reviewer 1. Whereas n=2 replicates for Fig S9B, I feel that this is justified in this case due to the robustness of the data, as well as the ethical implications to minimize the amount of animals. Data is clear, methods are described well, conclusions in the main text are appropriate.

One request: I do not have access to the Supplementary Tables, so I’m not sure if all OT genomic sequences / analysis sequences are listed. As such, I would like to request to please make sure to include the analyzed Off-targeted sequence (+genomic location), as well as potential relevant primer/probe sequences used for OT analysis (either to the methods section, or the Supplementary files).

Responses and Revisions Reviewer 4.2: Thank you for the comment. We have included the protospacer and guide RNA sequences, as well as genetic loci, in Supplementary Table 1. The sequencing data were deposited in the Sequencing Read Archive.

Thank you, this fully addresses my previous comment.

Reviewer Comment 4.3: The changes are appreciated. Indeed, published references can be used to support claims on safety as well. In line with this, I would request adding a reference that shows decreased bystander editing upon short-term ABE delivery (ie. as RNP) to the statement “In general, the delivery of ABE-RNPs mediated by CBB-liposomes could be considered as a more transient mode of delivery of gene-editing tools, consistent with decreased bystander editing”.

Also, please add a reference to the statement “small particles would better penetrate the retina and make the therapeutic agent available for cellular uptake”.

Responses and Revisions Reviewer 4.3: Thank you for the comment. In our previous study, we found that ABE8e-RNP delivery *in vitro* via Lipofectamine 3000 or lipid nanoparticles led to more precise editing than plasmid delivery,¹ (Figure 4, new panel “o,” shown below). In another study, we found that ABE delivered as RNP encapsulated into virus-like particles (eVLPs) led to lower bystander and offtarget editing than viral delivery *via* AAV and LV *in vivo*.² We added the references to the manuscript.

Regarding the tissue permeation, small protein GCAP1 whose diameter was 6.5 – 8.7 nm, permeated the retina more efficiently than GCAP1 complexed with liposomes (size range 150-170 nm).³ Our own data do not provide sufficient evidence to support the statement of enhanced penetration of the retina by smaller particles; therefore, we removed that statement.

[Figure Redacted]

“Fig. 4 | Lipid nanoparticle delivery of ABE RNPs. o, Next-generation sequencing analysis of ABE-editing outcomes in the cells with cDNA encoding *Rpe65 rd12*, treated with ABE delivered on a plasmid with Lipofectamine 3000 (L3k) or as LNP. Three analytical replicates, mean \pm S.D.”¹

Thank you, this fully addresses my previous comment.

Reviewer Comment 1.4: From Cre recombinase to base editors. A logical step in between could have been CRISPR-Cas9 based gene correction. Did you try this as well?

Original Response 1.4: Thank you for the suggestion. Cas9 in the eye causes severe off-target effects and apoptosis.¹ Therefore, we deliberately applied our CBB strategy to an adenine base editor (ABE) rather than to a Cas9 nuclease, since ABEs offer superior therapeutic promise. This choice is further validated by recent clinical successes, most notably a personalized ABE-based gene therapy administered to an infant patient, which underscores the clinical relevance of base editing over nuclease-driven approaches.²

Reviewer Comment 4.4: This is a valid explanation, and as reviewer 1 did not request any additional data or experiments this answer suffices. However, readers may from outside the ocular delivery field may have similar questions. As such, I would request that this is briefly explained/addressed in the discussion as well.

Response and Revision 4.4: We added the following sentences to the Discussion, paragraph 2:

“While CRISPR-Cas9 nuclease RNPs could be viewed as a logical intermediate between Cre and base editors, we prioritized ABE for in vivo studies because in comparison with Cas9 nuclease, ABE has the advantage of catalyzing single-base transitions without creating double strand breaks, with equivalent potential for RNP delivery as Cas9. The molecular interactions that we engage to deliver ABE RNP are applicable to other RNPs, and thus we envision that our CBB liposomes are applicable to other CRISPR nucleases and precise editors as well.”

Thank you for including this additional explanation to the manuscript.

Reviewer Comment 4.5: Regarding the schematic of Figure 4e: the ABE8e editing window tends to span between spacer nucleotides 3/4 – 8. As such, would the following underlined A not also be edited? GACCGTCAGAGGAGACTACA. Please check, and if indeed so, correct in the schematic.

Response and Revision 4.5: Thank you for pointing this out. In fact, adenines at positions 3 and 8 are often edited alongside the target adenine at position 6. The figure has been corrected to reflect this fact.

Figure 4. Delivery of ABE-RNPs to *rd12*-reporter cells, mediated by CBB lipiodoids. (...) (e)

Schematic diagram of the ABE-activity assay. EndoV = endonuclease V. Blue, SpCas9 PAM; red, target base; orange, bystander base; arrowhead, nick site; FAM, fluorescein. (f) Representative urea-PAGE-gel image of products of ABE deamination cleaved by EndoV. The gel was imaged with a fluorescein filter. N, no enzyme; P, positive control: 60 bp DNA with inosine in the middle; A, ABE-RNP in TAS; D, ABE-RNP in TAS with 10% DMSO. Numbers indicate molar excess of the CBB compound *versus* ABE. (g) Fluorescence quantification of substrate and products of deamination and EndoV cleavage. Two independent samples per data point were assayed in parallel.

With these revisions in Figure 4, my points on this topic have been fully addressed.

Reviewer Comment 1.11: In all formulations DMSO is present at ~10%. Dialysis could have easily removed it... Why not done?

Original Response 1.11: We removed the solvent by dialyzing the complex of ABE-RNP and CBB11 in TAS buffer with 10% DMSO through a 7000 MWCO membrane. The CBB11 amount in the resulting solution was quantified by LC-MS. The results indicated that most of the CBB11 (95.3%) was lost, suggesting that the CBB binding to protein is reversible, which helps support our stated goal of identifying molecules that would possess transient non-specific protein binding to assist in delivery. It should also be noted that DMSO is tolerated by the eye, and DMSO is present in some formulations.

The solvent is quickly removed through vitreous exchange, thus not presenting an issue even for clinical applications. We have added a sentence to the Results conveying more clearly the utility and safety of including DMSO in the formulation.

Reviewer Comment 4.11: Once more, this fully addresses the point raised by Reviewer 1. However, as other readers may have similar questions, and it appears that not removing DMSO is of pivotal importance for the stability and downstream applications of these formulations, this information seems of high importance to the readers as well. Especially as it may not be directly clear that currently DMSO is therefore a component of the final formulation, which is of high relevance for potential applications.

Whereas additional experiments / data that shows stability of specific CBBs are not required to discuss this issue, it is a potential limitation that should be discussed. Please add a section to the discussion section that underlines the importance of DMSO in the discussion. (indeed, this is not problematic for a variety of clinical formulations, but this could of course also be added to the discussion if the authors would desire to do so).

Response and Revision 4.11: Thank you for the suggestion. We have added the following text to the discussion:

“The CBB-RNP complex contained 7.25% DMSO, which was well tolerated by ABE and as such may be further explored as a vehicle for protein delivery.”

Thank you, this fully addresses my previous comment.

Reviewer Comment 1.13: What are the size of these CRE/CBB complexes?

Original Response and Revision 1.13. Dynamic light scattering indicated that the size of the Cre/CBB11 complex is 1400 nm and the size of the Cre/CBB14 complex is 1500 nm. These values are now included in the Result section.

Reviewer 4.13: Thank you for including this new data. As these particles are this large, this has implications for potential suitability for specific *in vivo* applications, which should be discussed.

Response and Revision 1.13: Thank you for the comment. We cannot draw conclusions about implications of charge and hydrophobicity of the Cre-CBB complexes on the *in vivo* efficacy, as both CBB11 and CBB14 effectively delivered Cre *in vivo* despite the higher hydrophobicity of CBB11. We rephrased this part of the Results section for clarity and added the following section to the discussion:

“An additional practical determinant is particle size. Dynamic light scattering measurements indicated that CBB11/Cre and CBB14/Cre form ~1.4–1.5 μm assemblies, with a significant fraction of CBB14-Cre forming smaller, 160-nm particles, whereas CBB11 alone forms ~400–500 nm colloids; importantly,

CBB11_doped liposomes (Lipo1–6) are ~140–160 nm and highly monodisperse. Micrometer_scale complexes are unlikely to diffuse broadly within the neural retina or traverse ocular barriers from systemic delivery, whereas systemic delivery of the smaller, narrowly distributed liposomal formulations would be effective. Notably, despite their large hydrodynamic diameters, CBB11 and CBB14 still delivered Cre via subretinal injection.”

Thank you, this fully addresses my previous comment.

Original Reviewer Comment 1.15: Please show experimental evidence that CBB11 is in fact incorporated into the liposomes.

Original Response 1.15: We have performed experiments related to the solubility of CBB11 and the liposome (Lipo 1). Unconjugated CBB11 is mostly insoluble, but when mixed with lipids it is retained in the suspension (see below). Please also see relevant response to **Comment 3.2**.

Response Figure 2: CBB11 (40 μ M) in TAS with 10% DMSO or Lipo1 (containing 40 μ M CBB11) in TAS was dialyzed overnight against TAS buffer, followed by centrifugation at 5000 rpm for 5 min. The CBB11 in the supernatant was quantified *via* LC-MS. The data suggest that formulation with the liposome prevented CBB11 aggregation. Revisions in Discussion part:

Particularly striking was the high efficacy observed when these compounds were formulated into ionizable cationic liposomal particles to improve the colloidal stabilities of the CBB-protein complexes.

These facts indicated that the protein binding on the surface of Lipo1 was intensified by CBB11. As expected, this enhanced liposome-protein interaction boosted intracellular protein delivery.

Comments Reviewer 4.15 Whereas the comparison of a control lipid to CBB11 in context of liposomes in Fig 5D is an interesting experiment, it does not fully answer the question of reviewer 1. The question specifically states “: Please show experimental evidence that CBB11 is in fact incorporated into the liposomes”. However, the data shown in Figure 5d could also be explained by direct nanocomplexation of ABE8e-RNP with CBB11. Response Figure 2 is a more conclusive / direct read-out for liposomal incorporation (especially since it’s LC-MS based analysis, which is a more direct read-out).

To answer this question as the authors appear to have access to LC-MS for formulation analysis, I would request that the authors incorporate data to the manuscript that directly confirms CBB11 integration to answer this comment.

General comment, based on these replies: please avoid the use of RPM in methods section, as the rcf (g) can differ dramatically based on the rotor diameter. Please check throughout the manuscript that all mentions of centrifugation speeds are shown in rcf (g), and not in RPM.

Response and Revision 4.15: Thank you for your comment. To directly answer the question of incorporation of CBB11 into the liposomes, we decided to perform gel filtration. We used PD10 desalting columns with a Sephadex G-25 resin. We traced the elution of CBB11 by fluorescence. We found fluorescence signals originating from CBB at elution volume (3-3.5 ml) corresponding to Lipo1 (**Fig. R1a**). As expected, we did not observe CBB fluorescence at the elution volume for Lipo2 (negative control without CBB11) or in the eluate for the column containing uncomplexed CBB11. Visually, we found that free CBB11 was retained in the column (**Fig. R1b-3**), and fluorescence scans of the columns confirmed the retention of the free CBB11 in the column (**Fig. R1c-3**).

Figure R1: Gel filtration of CBB-liposomes. (a) Coomassie fluorescence scan of a 96-well plate containing samples collected from the PD10 desalting column. V_E = elution volume. (b) A photograph of the PD10 desalting columns after the gel filtration. (c) Coomassie fluorescence scan of the PD10 desalting columns after the gel filtration.

With this experiment, I feel that the initial concern of Reviewer 1 has indeed been addressed. However, this proof is currently not included in the manuscript. Given the value of these data to support the general conclusions of this manuscript, and the clear results of this assay, I feel that these data belong, at the very least, in the Supplementary Info of the manuscript. Given how clear the results are, I feel that the current figure would suffice. However, if the authors wish to include a graph showing fluorescence per fraction instead of an image of the 96-well plate, this is equally acceptable.

Response to General comment: We corrected the revolutions per minute to g throughout the manuscript, where applicable. In some instances (sample dissolution in bench top thermomixer, microbial shakers) we kept rpm, a customary measure of mixing speed.

Thank you for these changes.

Reviewer Comment 1.18: P9L299: 'encapsulation within the particle...' if you refer to a new method, clarify or remove.

Original Response and Revision 1.18: We now clarify that ABE-RNP predominantly adsorbs to, rather than being fully encapsulated by, the lipid corona; the text has been reworded accordingly.

Reviewer Comment 4.18: Thank you for explaining this. This is now in line with the TOC graphic. However, if possible, please make sure to include a bit more clarity to the TOC graphical abstract: right now it is implied that the H⁺ is present in the cytosol, rather than the endocytosis pathway.

Response and Revision 1.18: Thank you for the advice. We expanded the TOC graphic to present the concept of endosomal escape triggered by acidification of the endosome. The TOC graphic has been modified accordingly, and the new version is included in the revised manuscript.

Figure 1. Construction of complexes, and proposed mechanism(s) underlying intracellular delivery of genome-editing ribonucleoproteins (RNPs) mediated by Coomassie Brilliant Blue (CBB) liposomes. The fundamental design principles for a CBB-containing delivery agent include: 1) the CBB protein-binding domain; 2) an ionizable-linker domain; and 3) a lipid or lipidoid domain. The CBB compounds are formulated into liposomes comprising ionizable lipid (orange), phospholipid (grey), cholesterol (red) and PEG lipid (light blue); and bind RNP on the liposome surface in aqueous, neutral environment. The RNP-liposome complex is absorbed by the cells *via* an endocytosis pathway. Subsequent protonation of the ionizable CBB-delivery agents and ionizable lipids releases the tagged proteins from the particles, enabling their intracellular transport to the nucleus to edit the targeted DNA sequence.

With these changes to the figure, my comment has been fully addressed.